# Marcus kinetics control singlet and triplet oxygen evolving from superoxide

Soumyadip Mondal[1], Huyen T. K. Nguyen[1,2], Robert Hauschild[1] & Stefan A. Freunberger[1✉]

Oxygen redox chemistry is central to life[1] and many human-made technologies, such as in energy storage[2–4]. The large energy gain from oxygen redox reactions is often connected with the occurrence of harmful reactive oxygen species[3,5,6]. Key species are superoxide and the highly reactive singlet oxygen[3–7], which may evolve from superoxide. However, the factors determining the formation of singlet oxygen, rather than the relatively unreactive triplet oxygen, are unknown. Here we report that the release of triplet or singlet oxygen is governed by individual Marcus normal and inverted region behaviour. We found that as the driving force for the reaction increases, the initially dominant evolution of triplet oxygen slows down, and singlet oxygen evolution becomes predominant with higher maximum kinetics. This behaviour also applies to the widely observed superoxide disproportionation, in which one superoxide is oxidized by another, in both non-aqueous and aqueous systems, with Lewis and Brønsted acidity controlling the driving forces. Singlet oxygen yields governed by these conditions are relevant, for example, in batteries or cellular organelles in which superoxide forms. Our findings suggest ways to understand and control spin states and kinetics in oxygen redox chemistry, with implications for fields, including life sciences, pure chemistry and energy storage.

More than half a century after the discovery that singlet oxygen forms from superoxide[8], what governs the evolution of singlet or triplet oxygen under many conditions relevant to life and human-made oxygen redox systems remains unknown. Oxygen redox chemistry is crucial to life[1] and encompasses some of the most fundamental and widespread chemical reactions, including those in batteries[2–4,9], fuel cells and electrocatalysis[10], and organic chemistry[11]. The oxidation states that are accessed in these reactions range from 0 (dioxygen) to −2 (oxide), with intermediate redox states of −½ and −1, pertaining to superoxide ($O_2^-$) and peroxide ($O_2^{2-}$) or the oxide radical ($O^{·-}$). Superoxide is pivotal in oxygen reduction and evolution reactions because it is the closest oxidation state to dioxygen. Dioxygen appears either in its ground state as triplet oxygen ($^3O_2$ or $^3\Sigma_g^-$) or as singlet oxygen ($^1O_2$ or $^1\Delta_g$) in its first electronically excited state. Whereas $^3O_2$ is relatively unreactive, $^1O_2$ is highly reactive with most organic matter[11,12]. This is readily evidenced in metal-ion batteries[6,7] and metal-$O_2$ batteries[3,4], in which $^1O_2$ is the primary source of degradation, causing decomposition of organic electrolytes and conductive carbon additives, which ultimately degrades the overall device function. Moreover, $^1O_2$ and (su)peroxide are well-known reactive oxygen species in biological systems and are involved in several processes from signalling to cell damage[1,5].

Superoxide liberates oxygen under a broad range of oxidizing conditions. A prevalent process is disproportionation due to the instability of superoxides in most of the environments[3,9,13]. This reaction occurs in both protic (aqueous)[14,15] and aprotic environments with relatively strong Lewis acids such as Li⁺ and Na⁺ (refs. 6,9,13,16). During disproportionation ($2O_2^- \rightarrow O_2 + O_2^{2-}$), one superoxide molecule is reduced

to peroxide, whereas the other is oxidized to form either $^3O_2$ or $^1O_2$. Examples can be found in both cellular respiration[1,5] and batteries[13,16–20]. Relative $^3O_2$ and $^1O_2$ yields and kinetics of superoxide disproportionation, and superoxide oxidation more generally, are therefore fundamental to these systems, but underlying mechanisms are still unknown.

Here we examined $^1O_2$ evolution from the oxidation of superoxide through both chemical reaction and disproportionation over a wide range of driving forces. On the basis of this, we observe distinct Marcus normal and inverted region behaviour for $^3O_2$ and $^1O_2$ evolution, which explains $^1O_2$ formation over a broad range of scenarios in which superoxide oxidation occurs, for example, in biological systems and energy storage.

## Excited species through electron transfer

Superoxide oxidation may form both $^3O_2$ and $^1O_2$ according to Wigner–Witmer spin conservation rules[21]. This applies to disproportionation, in which two doublet superoxides can produce triplet or singlet products[17], and to oxidation by a redox mediator (RM$^{ox}$) to form one excited ($^1O_2$) and one ground state molecule (the reduced mediator, RM$^{red}$) (ref. 22).

We previously investigated the mediated oxidation of superoxide through the reaction KO$_2$ + RM$^{ox}$ → O$_2$ + RM$^{red}$ + K⁺ in the search for possible $^1O_2$ evolution[18]. The potential $E^\circ_{RM^{ox/red}}$ was limited to moderate values because of the stability of the ether electrolyte used. We measured $^3O_2$ and $^1O_2$ yields using mass spectrometry and a chemical trap, respectively, and tracked the reaction kinetics using ultraviolet–visible

[1]Institute of Science and Technology Austria (ISTA), Klosterneuburg, Austria. [2]Present address: Texas Materials Institute, The University of Texas at Austin, Austin TX, USA. ✉e-mail: stefan.freunberger@ist.ac.at

(UV–Vis) spectroscopy. The data shown in Extended Data Fig. 1 supported two main conclusions: first, $^1O_2$ was observed with RMs with $E^\circ_{RM^{ox/red}} \gtrsim E^\circ_{3_{O_2/KO_2}} + 1$ V, which could be in accordance with the frequently quoted threshold[3,23–26] for $^1O_2$ evolution based on the energy difference of 0.97 eV between $^3O_2$ and $^1O_2$; and second, the kinetics of superoxide oxidation over the measured range of driving forces exhibited Marcus normal and inverted region behaviour. This behaviour is characterized by a parabolic relationship between the logarithm of the kinetic constant $k$ and the driving force (free energy change) $-\Delta G^\circ = \left(E^\circ_{RM^{ox/red}} - E^\circ_{3_{O_2/KO_2}}\right)F$:

$$k = Z_{el} \cdot e^{\frac{-(\Delta G^\circ + \lambda)^2}{4RT\lambda}} \quad (1)$$

where $Z_{el}$ is the collision factor, $\lambda$ is the reorganization energy and $F$ is the Faraday constant[22,27,28]. The kinetic constant $k$ reaches a peak when $-\Delta G^\circ = \lambda$ and electron transfer becomes barrierless. The inverted region results from growing barriers at even higher driving forces. The data suggested that superoxide oxidation kinetics follow a single parabola and that some $^1O_2$ is generated for $-\Delta G^\circ \gtrsim 0.97$ eV. However, what controls the extent to which $^1O_2$ or $^3O_2$ evolve is yet unknown.

$^1O_2$ evolution should be a distinct elementary step both thermodynamically and kinetically. We, therefore, propose that the observed parabola in Extended Data Fig. 1 describes oxidation to $^3O_2$, and a second parabola should appear at sufficiently large driving forces, representing oxidation to $^1O_2$ (Fig. 1a). The intersection of these two parabolas would indicate the transition from $^3O_2$ to $^1O_2$. This follows classical work by Marcus on the electrogeneration of electronically excited species[22,27]. However, this hypothesis has not yet been experimentally evaluated for $^1O_2$ generation.

Based on equation (1), a kinetic expression with separate terms for $^3O_2$ and $^1O_2$ evolution can be given as

$$k_{1+3} = k_3 + k_1 = Z_{el,3} \cdot e^{\frac{-(\Delta G^\circ + \lambda_3)^2}{4RT\lambda_3}} + Z_{el,1} \cdot e^{\frac{-(\Delta G^\circ + \Delta G^\circ_{1\leftarrow3} + \lambda_1)^2}{4RT\lambda_1}}. \quad (2)$$

The subscripts 3 and 1 on $k$, $Z_{el}$ and $\lambda$ denote values for $^3O_2$ and $^1O_2$, respectively. The blue- and red-dashed parabolas in Fig. 1a represent the terms for $^3O_2$ and $^1O_2$, whereas the solid line shows their sum. The $k_3$ parabola results from the transition from the black to the blue potential energy surfaces in Fig. 1b, which are shown for $-\Delta G^\circ \approx \lambda_3$ for case (i). The vibrational ground states of $^3O_2$ and $^1O_2$ differ by $\Delta H^\circ_{1\leftarrow3} = \Delta H^\circ\left(^1\Delta_g \leftarrow ^3\Sigma^-_g\right) = 0.97$ eV, where $\Delta H$ is the enthalpy change[12]. Assuming, for now, a vanishing entropy change $\Delta S$ as typically done in electrochemiluminescence literature[28], $\Delta G^\circ_{1\leftarrow3} = \Delta G^\circ\left(^1\Delta_g \leftarrow ^3\Sigma^-_g\right) \approx \Delta H^\circ_{1\leftarrow3}$, by which the singlet potential energy surface shifts vertically (Fig. 1b, red). For sufficiently large driving forces, the barriers $\Delta G^\ddagger$ and kinetics for crossing to $^3O_2$ and $^1O_2$ become equal (Fig. 1, case (ii)) or even barrierless to $^1O_2$ (case (iii)). Hence, individual kinetic parabolas for superoxide oxidation to $^3O_2$ and $^1O_2$ can be constructed. The hypothetical parabola for $^1O_2$ evolution can be drawn if we further assume, for now, equal collision factors $Z_{el,1} = Z_{el,1}$ and equal reorganization energies $\lambda_1 = \lambda_3$ as fitted to the data in Extended Data Fig. 1. The theory developed initially for homogeneous electron transfer can still be used for semiconductors or insulators, as noted by Marcus[22]. Therefore, the considerations in Fig. 1 apply to the oxidation of solid superoxides and superoxide solutions.

## Kinetics over extended driving forces

To test the hypothesis of individual kinetic parabolas for $^3O_2$ and $^1O_2$ evolution, we selected acetonitrile (MeCN) as the solvent. Acetonitrile is highly stable against oxidation and, therefore, allows for large driving forces (Extended Data Fig. 2a) to be tested. We used a wide range

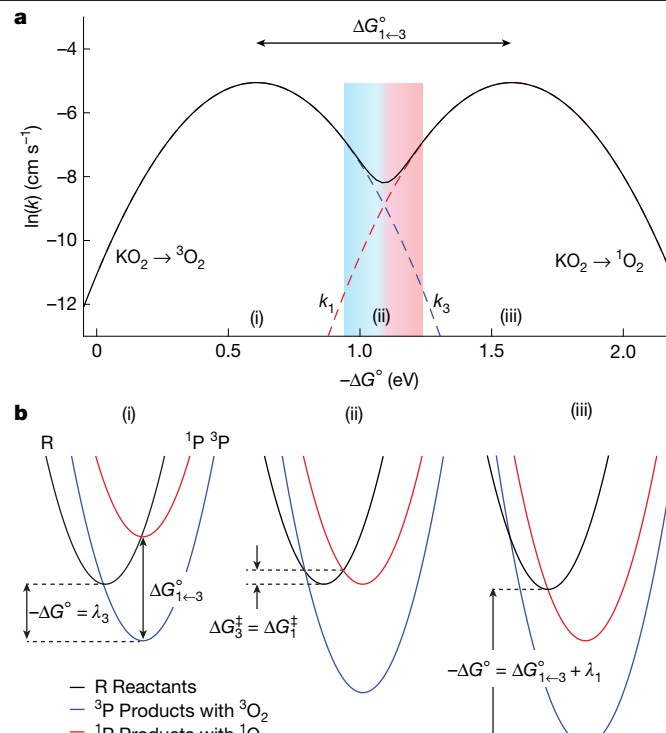

**Fig. 1 | Marcus theory suggests separate kinetics of superoxide oxidation to $^3O_2$ and $^1O_2$. a**, Hypothesis for how the driving force could govern $^3O_2$ and $^1O_2$ formation kinetics from superoxide oxidation. The left parabola results from previously measured rate constants for mediated superoxide oxidation with driving forces up to $-\Delta G^\circ \approx 1.2$ eV as shown in Extended Data Fig. 1. However, a single kinetic parabola could not conclusively explain why $^1O_2$ formed for $-\Delta G^\circ \gtrsim 1$ eV. Based on the considerations in **b**, individual kinetic parabolas ($k_3$ and $k_1$) for the reactions yielding $^3O_2$ and $^1O_2$ can be constructed with the full line showing their sum. The maxima are shifted by $\Delta G^\circ_{1\leftarrow3} \approx 0.97$ eV (see text), and equal prefactors and reorganization energies are assumed. The blue- and red-shaded area shows the transition from $k_3/(k_{1+3}) = 0.99$ to $k_1/(k_{1+3}) = 0.99$. **b**, Potential energy surfaces for mediated superoxide oxidation for different driving forces. Black, blue and red parabolas denote reactants ($KO_2 + RM^{ox}$) and $^3O_2$ or $^1O_2$ in the products ($^3O_2 + RM^{red}$ or $^1O_2 + RM^{red}$), respectively. The cases shown are for barrierless reactions to $^3O_2$ (i) and $^1O_2$ (iii) and equal barriers (ii). The subscripts 3 and 1 denote triplet and singlet states, respectively.

of mediators (Extended Data Fig. 2b) with redox potentials that exceed the expected maximum of the $^1O_2$ parabola. Figure 2a shows the measured kinetics as a function of driving force relative to $E^\circ_{3_{O_2/KO_2}}$ (Methods). The measured kinetics can be adequately fitted to equation (2). The blue- and red-dashed parabolas represent the terms for $^3O_2$ and $^1O_2$ evolution, whereas the solid line shows their sum. Evidence for these parabolas corresponding to $^3O_2$ and $^1O_2$ is provided by measured yields (Fig. 2b,c). The $^3O_2$ yields were measured by mass spectrometry and quantified as the molar ratio $^3O_2/RM^{ox}$. $^1O_2$ formation was measured by its specific 1,270 nm near-infrared (NIR) radiation (Methods). Absolute quantification of $^1O_2$ is difficult, and values were, therefore, normalized to the maximum observed.

Observable yields of $^3O_2/RM^{ox}$ and NIR intensities do not simply resemble the relative kinetics of $^3O_2$ and $^1O_2$ formation as obtained in Fig. 2a. Instead, the formed $^1O_2$ undergoes multiple decay pathways, of which some yield $^3O_2$ and only a small fraction emits the NIR radiation[12]. To rationalize the measured values, we simulated them based on formation rates and $^1O_2$ decay processes (Methods and Extended Data Fig. 3): (1) $^3O_2$ and $^1O_2$ formation rates as given by the parabolas in Fig. 2a; (2) physical and reactive $^1O_2$ quenching by the solvent; (3) physical and reactive $^1O_2$ quenching by the mediators; and (4) $^3O_2$ losses resulting from reactive quenching of $^1O_2$ with solvent or mediator. Simulation

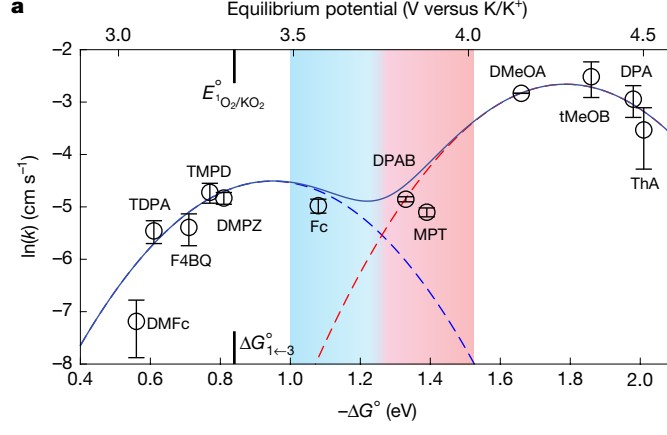

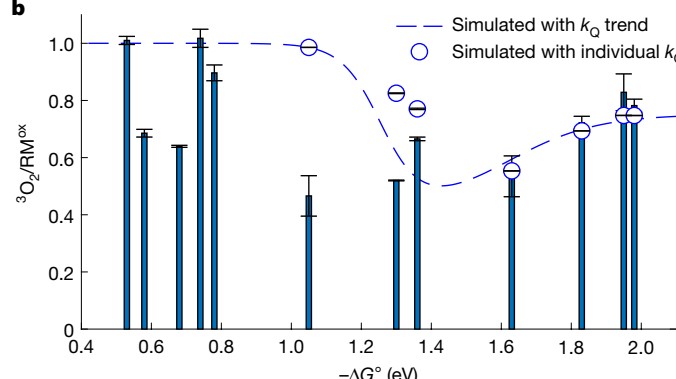

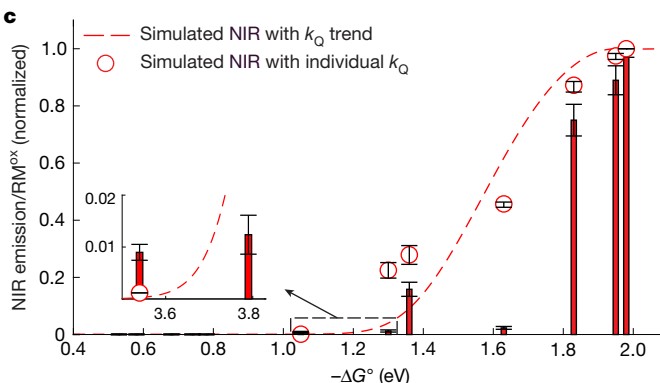

**Fig. 2 | Free energy dependence of superoxide oxidation kinetics to $^3O_2$ and $^1O_2$ and their yields. a**, We measured the kinetic constants $k$ for mediated $KO_2$ oxidation in MeCN electrolyte with mediators covering a large range of redox potentials. Plot of $\ln(k)$ compared with the mediator potential ($E^\circ_{RM^{ox/red}}$, top axis) and driving force ($-\Delta G^\circ$, bottom axis). $-\Delta G^\circ = \left( E^\circ_{RM^{ox/red}} - E^\circ_{3_{O_2/KO_2}} \right) F$, where $E^\circ_{3_{O_2/KO_2}} = 2.48$ V on the K/K$^+$ scale. The mediators are shown in Extended Data Fig. 2. The full line best fits equation (2); the broken line parabolas represent the first and second terms in equation (2). The fitted values are $Z_{el,3} = 1.10 \times 10^{-2}$ cm s$^{-1}$, $Z_{el,1} = 7.00 \times 10^{-2}$ cm s$^{-1}$, $\lambda_3 = \lambda_1 = 0.95$ eV, $\Delta G^\circ_{1\leftarrow3} = 0.84$ eV and $R^2 = 0.998$. Based on these fits, the standard potential $E^\circ_{1_{O_2/KO_2}}$ and associated driving force $\Delta G^\circ_{1\leftarrow3}$ are marked. They are linked by $E^\circ_{1_{O_2/KO_2}} = E^\circ_{3_{O_2/KO_2}} + \Delta G^\circ_{1\leftarrow3}/F$. The blue- and red-shaded area indicates the transition from $k_3/(k_{1+3}) = 0.99$ to $k_1/(k_{1+3}) = 0.99$. **b**, $^3O_2$ yield per mole of RM$^{ox}$ (bars) during $KO_2$ oxidation as measured by mass spectrometry. The dashed line and the circular markers show simulated $^3O_2$ yields considering $^1O_2$ quenching by solvent and redox mediator (Methods and Extended Data Fig. 3). The dashed line used the trend line for the mediator quenching rate constant $k_Q$, whereas the markers use the individually measured values (Extended Data Fig. 3c). **c**, Normalized 1,270 nm NIR emission (bars) during $KO_2$ oxidation. The dashed line and the circular markers show the simulated NIR emission considering $^1O_2$ formation with the kinetics $k_1$ (the right parabola in **a**) and $^1O_2$ quenching by solvent and redox mediator (Methods and Extended Data Fig. 3). Data are presented as mean ± s.d. ($n \geq 3$).

results are shown as dashed lines and markers in Fig. 2b,c and resemble the measured data. Deviations result from simplifications of the model, such as not accounting for specific reactivities of the chemically diverse mediators (Methods). Overall, a proper fit of the kinetics to the double parabola in equation (2) (Fig. 2a), the deficiencies in $^3O_2$ yields and the growing NIR signal beyond the predicted onset potential (Fig. 2b,c) all support the proposed hypothesis that individual Marcus parabolas govern $^3O_2$ and $^1O_2$ formation from superoxide.

The adequate fit supports several important conclusions: (1) The prefactors $Z_{el,1} \approx 6.3 \times Z_{el,3}$ result in substantially higher maximum kinetics for $^1O_2$ evolution. (2) Driving forces for maximum kinetics are solvent-dependent, as shown by the comparison between values in ether solvent (Fig. 1a; $\lambda_3 = 0.61$ eV) and acetonitrile (Fig. 2a; $\lambda_3 = 0.95$ eV). This aligns with the prediction[27] of Marcus and work by Miller[29]. The crossing point of the two parabolas is not at a constant driving force, which benefits $^1O_2$ for solvents with lower $\lambda$ (Extended Data Fig. 4a). (3) $E^\circ_{1_{O_2/superoxide}}$ should not be considered the threshold above which $^1O_2$ rather than $^3O_2$ forms[18,24,26] as suggested from the electrochemiluminescence literature (Methods) but as a possible onset of $^1O_2$. Reaching $E^\circ_{1_{O_2/superoxide}}$ per se tells little about the relative $^3O_2$ and $^1O_2$ formation kinetics because of the solvent-dependent reorganization energy (Extended Data Fig. 4). Instead, driving forces $-\Delta G^\circ > \lambda_3$ are required to slow $^3O_2$ formation to benefit $^1O_2$. (4) In the electrochemiluminescence literature, the entropy change is unknown and typically neglected[28]. Having data that show the peaks for $^3O_2$ and $^1O_2$ permits us to determine $\Delta G^\circ_{1\leftarrow3} = 0.84$ eV and with $\Delta H^\circ_{1\leftarrow3} = \Delta G^\circ_{1\leftarrow3} + T\Delta S^\circ_{1\leftarrow3}$ we obtain $T\Delta S^\circ_{1\leftarrow3} = -0.13$ eV. Consequently, $E^\circ_{1_{O_2/superoxide}} = E^\circ_{3_{O_2/superoxide}} + 0.84$ V can now be given more precisely, rather than with the typically used difference of 0.97 V.

Figure 2a establishes a working curve for an extensive range of $\Delta G^\circ$, facilitating an understanding of the behaviour of various important systems. Lewis-acid- and Brønsted-acid-driven superoxide disproportionation are two widely relevant cases, examined in the following sections. More examples of superoxide oxidation, in which an explanation of whether, or not, $^1O_2$ forms has been unknown, but which can now be explained, are examined in Extended Data Fig. 4 and the Methods. These examples include superoxide in contact with $CO_2$ and organic peroxides, with relevance for energy storage and biological systems[1,4,30].

## Disproportionation in non-aqueous systems

$^1O_2$ is known to cause degradation in non-aqueous alkali metal-$O_2$ batteries[3,13,16–19]. The Li–$O_2$ battery, for example, operates by reversibly forming lithium peroxide at the positive electrode, $O_2 + 2e^- + 2Li^+ \rightleftarrows Li_2O_2$. Initially formed $LiO_2$ disproportionates, which always results in some $^1O_2$ according to $2LiO_2 \rightarrow Li_2O_2 + x^3O_2 + (1-x)^1O_2$ (refs. 13,16–19). Weak Lewis acids such as tetrabutylammonium (TBA$^+$), or other similar cations from ionic liquid electrolytes, are often present in these cells. Despite not driving disproportionation themselves, weak Lewis acids were found to raise the $^1O_2$ fraction from about 2% in pure Li$^+$ electrolyte to around 20% for 1/1 Li$^+$/TBA$^+$ electrolyte[16]. However, to date, it has been unclear why pure Li$^+$ generates $^1O_2$ at all and why weak Lewis acids should increase the $^1O_2$ fraction.

Figure 3a shows the thermodynamics of the relevant redox couples as a function of the Li$^+$ and TBA$^+$ salt fractions in glyme electrolyte. Figure 3b shows the NIR emission intensities on adding $KO_2$ as a superoxide source into these electrolytes. The thermodynamics are explored in further detail in the Methods and Extended Data Fig. 5 and are summarized here. $Li_2O_2$ is insoluble, and the potential $E_{3_{O_2/Li_2O_2(s)}}$, therefore, fixed (Fig. 3a, black line). However, superoxide is appreciably soluble in non-aqueous electrolytes[31–33]: species include $(Li^+O_2^-)_{n\geq1,(sln)}$ clusters or ion pairs, where (sln) denotes solvate species. Solid $LiO_{2(s)}$ has the lowest Gibbs free energy, and increasingly solvated species are

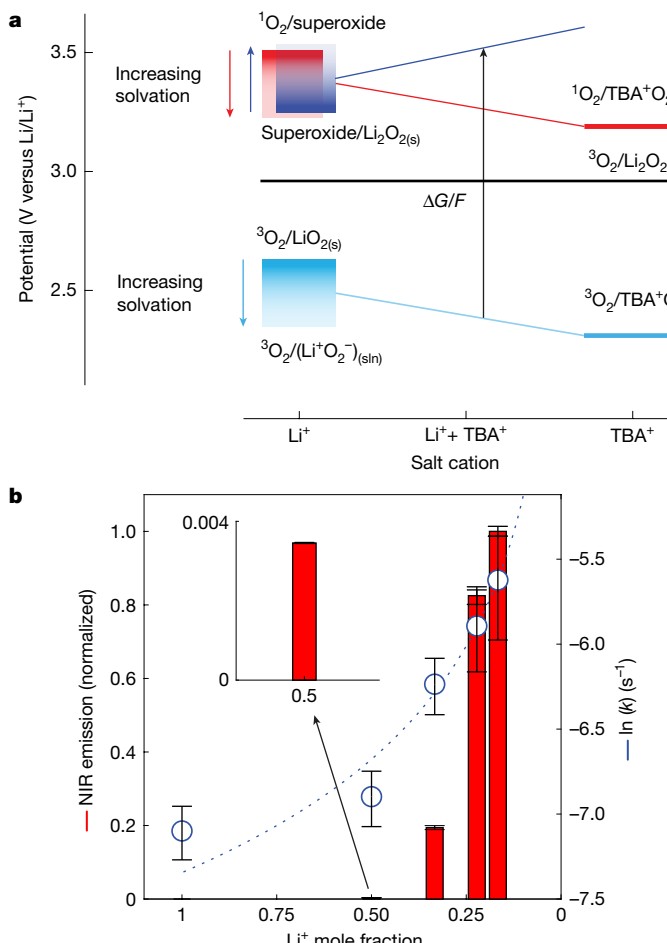

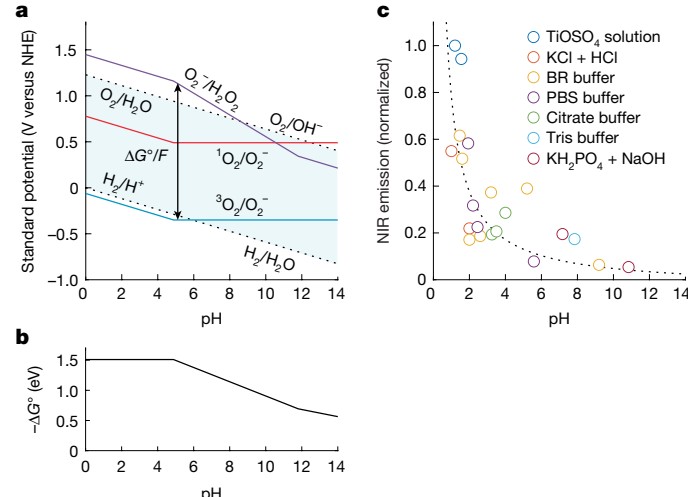

**Fig. 3 | Driving forces, kinetics and $^1O_2$ formation during Lewis-acid-induced superoxide disproportionation. a**, Thermodynamics of relevant redox couples for $Li^+$-induced superoxide disproportionation as a function of the fractions of $Li^+$ and $TBA^+$ salt. The gradient boxes and short arrows indicate increasing superoxide solvation with potentials between values relevant for solid $LiO_{2(s)}$ (dark colour), solvated $(Li^+O_2^-)_{n \geq 1,(sln)}$ clusters, and solvated $(Li^+O_2^-)_{(sln)}$ (faded colour). $LiO_2$ is in all relevant electrolytes at least somewhat soluble as $(Li^+O_2^-)_{n \geq 1,(sln)}$ clusters. The nature of superoxide shifts from $(Li^+O_2^-)_{n \geq 1,(sln)}$ towards $(TBA^+O_2^-)_{(sln)}$ as the cation changes from pure $Li^+$ towards pure $TBA^+$. The inclined lines indicate the associated shift of the potentials. $E_{1O_2/\text{superoxide}} = E_{3O_2/\text{superoxide}} + 0.84$ V. The driving force $\Delta G = -(E_{\text{superoxide}/Li_2O_2} - E_{3O_2/\text{superoxide}})F$ grows with the shift of the $O_2$/superoxide and superoxide/$Li_2O_{2(s)}$ potentials. Note that the $O_2$ reduction potential changes nonlinearly with the $Li^+:TBA^+$ ratio[32]. **b**, Normalized 1,270 nm NIR emission as a function of $Li^+$ mole fraction, which determines the driving force. The markers show superoxide disproportionation kinetics as measured by pressure evolution. The dotted line is a guide to the eye. Glyme served as the electrolyte solvent. Data are presented as mean ± s.d. ($n \geq 3$).

**Fig. 4 | Driving forces and NIR emission during proton-induced superoxide disproportionation. a**, Pourbaix diagram showing the pH-dependent stability range of aqueous electrolyte and the standard potentials $E°$ of the indicated redox couples on the normal hydrogen electrode (NHE) scale[15]. The kinks in the curves arise from the $pK_a$ values of $H_2O_2$ ($pK_a = 11.7$) and $HO_2$ ($pK_a = 4.8$). For simplicity, $O_2^-/H_2O_2$ is written while the pH-dependent (de)protonated species are meant. $E_{1O_2/O_2^-}° = E_{3O_2/O_2^-}° + 0.84$ V. **b**, The pH-dependent driving force for superoxide disproportionation is $-\Delta G° = (E_{O_2^-/H_2O_2}° - E_{3O_2/O_2^-}°)F$ as indicated by the vertical arrows in **a**. **c**, $^1O_2$-specific NIR emission at 1,270 nm as a function of pH. Each marker represents a single measurement. The dotted line is a guide to the eye. PBS, phosphate-buffered saline.

## Proton-induced disproportionation

We next investigated how pH-related changes in driving force affect the $^1O_2$ formation from proton-induced superoxide disproportionation. This is particularly relevant for living organisms, in which superoxide is found in organelles with pH levels ranging between 4.7 and 8 (ref. 35), as well as in aqueous electrocatalysis[10]. Figure 4a shows the Pourbaix diagram with the potentials of relevant redox couples as a function of pH. Again, the driving force for disproportionation is the difference between the $^3O_2$/superoxide and superoxide/peroxide couple, marked with the vertical arrows. It grows from about 0.5 eV at pH 14 to about 1.5 eV below pH 4.8 (Fig. 4b). The oxidant potential exceeds $E_{1O_2/O_2^-}°$ at pH about 10.5, in which an onset of $^1O_2$ formation can be expected.

Figure 4c shows the $^1O_2$-specific NIR signal at 1,270 nm on proton-induced superoxide disproportionation over a pH range from about 1 to 10.8. We exposed $KO_2$ to various buffer solutions under vigorous stirring and recorded the NIR signal (Methods). A strongly increasing NIR signal with decreasing pH is in accord with the increasing driving force. A pH of around 11 giving low but non-negligible $^1O_2$ yields is consistent with reliable theoretical[17] and experimental works[14] on the reaction $HO_2 + O_2^- \rightarrow O_2 + O_2^-$, which found $^1O_2$ evolution only about 0.3 eV endoergic and around 0.2% $^1O_2$ production on exposing $KO_2$ to $H_2O_2$ as the proton source.

## Conclusions

We found that the driving force for superoxide oxidation to $^3O_2$ and $^1O_2$ is the common descriptor that determines the spin state, following individual Marcus normal and inverted region behaviour. $^1O_2$ can become significant only because the kinetics for $^3O_2$ evolution slows down in its inverted region. The results help clarify previously inconclusive findings about $^1O_2$ formation from superoxide, including through interaction with chemical oxidants, and proton and Lewis-acid-driven disproportionation. For disproportionation, the results explain the

increasingly less stable[32–34]. The $^3O_2$/superoxide potentials are, therefore, within the upper limit of $E_{3O_2/LiO_{2(s)}}$ and the lower limit of $E_{3O_2/(Li^+O_2^-)_{(sln)}}$ (Fig. 3a, light-blue gradient-coloured box). When $TBA^+$ is present, the lower limit extends to $E_{3O_2/(TBA^+O_2^-)_{(sln)}}$.

The superoxide/$Li_2O_{2(s)}$ couple acts as the oxidant during superoxide disproportionation. $E_{\text{superoxide}/Li_2O_{2(s)}}$ exceeds $E_{1O_2/\text{superoxide}}$ already in pure $Li^+$ electrolyte, in which the superoxide species are solvated, and therefore explains why an onset of $^1O_2$ is observed. $^1O_2$ is detected in pure $Li^+$ electrolyte using more sensitive chemical trapping[16]. Addition of $TBA^+$ favours more solvated species, which also explains why $TBA^+$ increases the fraction of $^1O_2$. In line with this, Fig. 3b shows steeply increasing $^1O_2$ evolution as the $TBA^+$ mole fraction increases, along with faster kinetics.

increasing $^1O_2$ formation with stronger Brønsted and weaker Lewis acidity, respectively, because of their impact on driving forces. Increasing $^1O_2$ yield with lower pH aligns with higher driving forces as the pH decreases. This corresponds with higher and lower pH in respiratory (mitochondria) and degenerative organelles (lysosomes)[35], respectively, in which $^1O_2$ must be avoided or may even be beneficial. This connection between pH and $^1O_2$ formation may well have been a so far unrecognized evolutionary driver for the pH found in organelles. In human-made redox systems, in which $^1O_2$ is, in most cases, detrimental and damaging, strategies to suppress $^1O_2$ should aim at reducing the driving forces for superoxide oxidation, increasing the reorganization energy, or avoiding situations in which superoxide disproportionates. We discuss the wider relevance for oxygen redox systems in life sciences and energy, and for the electrogeneration of excited species more generally in the Methods. The findings offer insights into understanding and controlling spin states and kinetics in oxygen redox chemistry, with implications for fields such as the life sciences and energy storage.

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

## Methods

### Materials

All chemicals were from Sigma Aldrich unless indicated differently. Potassium superoxide ($KO_2$), potassium perchlorate ($KClO_4$, ≥99.99%), lithium bis(trifluoromethanesulfonyl)imide (LiTFSI, 99.99%) and tetrabutylammonium bis(trifluoromethanesulfonyl) imide (TBATFSI, ≥99.0%) were dried under reduced pressure for 24 h at 100 °C. Decamethylferrocene (DMFc), tris[4-(diethylamino) phenyl]amine (TDPA), $N,N,N',N'$-tetramethyl-$p$-phenylenediamine (TMPD), ferrocene (Fc), ferrocenium tetrafluoroborate ($FcBF_4$), 9,10-diphenylanthracene (DPA), 1,4-bis(diphenylamino)benzene (DPAB), thianthrene (ThA), 1,4-dimethoxyanthracene (DMeOA, Acros Organics), 1,4-di-tert-butyl-2,5-dimethoxybenzene (tMeOB, BLDpharm) and 5,10-dihydro-5,10-dimethylphenazine (DMPZ, TCI chemicals) were used as received. $N$-methyl phenothiazine (MPT), 1,4-diazabicyclo[2.2.2]octane (DABCO) were sublimed. 18-Crown-6 and tetrafluorobenzoquinone ($F_4BQ$) were recrystallized from ethanol. Tetraethylene glycol dimethyl ether (TEGDME, ≥99%) was dried over lithium, distilled under vacuum and further stored over activated 3 Å molecular sieves. Acetonitrile (MeCN, 99.8% anhydrous) was stored over activated molecular sieves. Both solvents had a water content below 5 ppm as determined by Karl–Fischer titration (Mettler Toledo). All non-aqueous experiments were performed in an Ar-filled glovebox (Vigor) or hermetically sealed setups without air exposure. The structures of the mediators and their redox potentials are shown in Extended Data Fig. 2.

### Electrochemistry and mediator oxidation

Electrochemistry experiments were performed using a BioLogic potentiostat (SP-300 and MPG-2). Cyclic voltammetry was performed with a glassy carbon disk as the working electrode and a glassy carbon rod as the counter electrode in a one-compartment glass cell. Partially delithiated $Li_{(1-x)}FePO_4$ (LFP, MTI), separated by a Vycor glass frit, was used as the reference electrode. $DMFc/DMFc^+$ was used as the internal standard and converted using $E^\circ_{DMFc/DMFc^+} = E^\circ_{Li/Li^+} + 3.16\ V = E^\circ_{K/K^+} + 3.02\ V$. DMeOA, tMeOB, DPA and ThA were electrochemically oxidized in an H-cell. The oxidation compartment contained a Pt working electrode, the reference electrode and 5 ml MeCN containing 2 mM RM and 10 mM $KClO_4$. The reduction compartment contained Ni foam as a counterelectrode with 5 ml of 100 mM 1,4-benzoquinone (sublimed) in MeCN. A $K^+$ selective ion-exchange membrane separated the two compartments. RMs were oxidized galvanostatically to 80% of their total capacity. DMFc, TDPA, TMPD, DMPZ, MPT and DPAB were oxidized using 1 equiv. $NOBF_4$ in MeCN. After 3 h stirring, they were precipitated with cold diethyl ether, filtered and dried under a vacuum at 30 °C for 12 h. $F_4BQ$ is the oxidized form itself.

### Measurements of kinetics

Kinetics of mediated superoxide oxidation were measured using UV–Vis spectroscopy using an Avantes AvaSpec-HSC spectrometer with AVALIGHT-DH-S-BAL light source and fibre optics to perform measurements inside the glove box. Pure $KO_2$ powder was pressed into about 0.5 mm thick pellets using a 7 mm die set and a hand press (PIKE). In a 10-mm quartz cuvette (Hellma), a $KO_2$ pellet was placed in a polytetrafluoroethylene frame for alignment, followed by a magnetic stirring bar and then the cuvette was sealed with a gas-tight injection lid. $RM^{ox}$ solution containing 10 mM $KClO_4$ was then injected using a gas-tight syringe (Hamilton). Consumption of $RM^{ox}$ was followed except for $F_4BQ$, in which formation of $RM^{red}$ was followed (Extended Data Fig. 7). Data and error bars are presented as mean ± s.d. ($n \geq 3$). Error bars appear asymmetric on an $\ln(k)$ scale. Repetitions mean that each time a new portion of $RM^{ox}$ was produced by electrochemically oxidizing a portion of $RM^{red}$ or by dissolving the chemically produced oxidized form. Differences in the magnitude of the error bars among mediators arise from

their large chemical diversity and specific reactivities. For example, $Fc^+$ and reduced quinones react with $O_2$, or high-voltage $RM^{ox}$ shows limited long-term stability in the electrolyte. For the latter, it was checked that degradation was at least several times slower than the oxidation of $KO_2$.

Kinetics of $Li^+$-induced superoxide disproportionation in TEGDME were measured by placing $KO_2$ powder in a closed reaction vessel equipped with a pressure sensor (Omega, PAA35X) and injecting the $Li^+$ electrolyte using a syringe through a septum (Extended Data Fig. 8).

### $^3O_2$ yields and 1,270 nm emission measurements

$^3O_2$ yields on mediated superoxide oxidation were measured using mass spectrometry, as detailed previously[30]. The $RM^{ox}$ solutions were injected using a gas-tight syringe, and the measurement continued until the $O_2$ signal ceased. We used the $^1O_2$-specific NIR emission at 1,270 nm from the decay of $^1O_2$ to $^3O_2$ to determine $^1O_2$ yields and lifetimes as detailed previously[30]. The signal was recorded from the detector using an oscilloscope (Pico Technology) and at a gain of 820 V (control voltage). Extended Data Fig. 6 shows examples of the recorded signal during mediated oxidation, as well as $Li^+$- and $H^+$-induced disproportionation.

### From oxidation rates to $^3O_2$ yields and NIR emission intensities

We use the yields of $^3O_2/RM^{ox}$ and the normalized NIR intensities in Fig. 2b,c to prove that the two kinetic parabolas correspond to $^3O_2$ and $^1O_2$ evolution. To rationalize their assignment, we use a minimal model to calculate expected $^3O_2/RM^{ox}$ and NIR emission intensities based on formation rates and $^1O_2$ decay processes. We considered the following processes: (1) $^3O_2$ and $^1O_2$ formation rates ($k_3$ and $k_1$) as given by the two kinetic parabolas in Fig. 2a; (2) physical and reactive $^1O_2$ quenching by the solvent; (3) physical and reactive $^1O_2$ quenching by the reduced mediators $RM^{red}$. Note that the same processes with $RM^{ox}$ are typically negligible in comparison because of the electron demand of these processes[36]; and (4) $^3O_2$ losses ($^3O_2/RM^{ox} < 1$) resulting from reactive quenching of $^1O_2$ with solvent or mediator.

Losses in $^3O_2/RM^{ox}$ in Fig. 2b result from incomplete physical quenching of $^1O_2$ to $^3O_2$ due to reactions of $^1O_2$ with electrolyte or RM. Hence, we start by considering the decay routes. There are multiple decay routes, as shown in Extended Data Fig. 3f, and only a small fraction of the total $^1O_2$ can be detected by the NIR emission at 1,270 nm (refs. 12,37). First, interactions of $^1O_2$ with solvent will result in electronic-to-vibrational (e–v) deactivation and reactions with the solvent. The first-order $^1O_2$ decay rate constant $k_D = k_{d,r} + k_d$ is composed of a reactive fraction $k_{d,r}$ and a non-reactive fraction $k_d$. Second, electron-rich species, such as reduced mediators, exert charge transfer quenching; however, they may also react with the electrophilic $^1O_2$ along pathways depending on the particular chemistry[36]. The rate constant $k_Q = k_{q,r} + k_q$ is equally composed of a reactive fraction $k_{q,r}$ and a non-reactive fraction $k_q$. Third, radiative decay of $^1O_2$ to $^3O_2$ with the emission of a 1,270 nm photon.

We measured the rate constants $k_D$ and $k_Q$ using the luminescence lifetime. An $O_2$-saturated solution containing Rose Bengal (absorbance of about 0.1 in a 1 cm fluorescence cuvette) was illuminated using a pulsed Coherent OBIS 561 nm laser. The lifetime ($\tau$) and the decay constant ($k = 1/\tau$) of $^1O_2$ were measured using the NIR detector. $\tau$ was obtained by fitting the decay profile with $\exp(-t/\tau)$ (Extended Data Fig. 3a). $\tau$ in the absence of a quencher relates to the solvent quenching rate constant $k_D$. $k_Q$ is obtained by plotting $1/\tau$ against the $RM^{ox}$ concentration $c_{RM^{ox}}$ (Extended Data Fig. 3b,c). Similar to previously reported quenchers[12], $k_Q$ decreases with $E^\circ_{RM^{ox/red}}$ following a trendline with a slope of about $-10^{3.5}\ V^{-1}$. The logarithmic dependence of the non-radiative fraction $k_q$ is explicable by the required partial charge transfer from the $e^-$-rich quencher to the $^1O_2$, which makes quenchers with lower redox potentials more efficient[12]. We consider that the reactive fraction $k_{q,r}$ depends similarly on redox potential, given that the addition reactions equally require charge transfer from the substrate to $^1O_2$ (refs. 12,36). The scattering of $k_Q$ around the trendline arises from the large chemical diversity of the used mediators. We used both the

trendline and the individual values of $k_Q$ for the prediction of NIR intensities in Fig. 2c. As for the fraction of reactive deactivation, $k_{d,r}/k_D$ and $k_{q,r}/k_Q$, these can be determined from $^3O_2$ consumption[12], which we do below when simulating values of $^3O_2/RM^{ox}$.

The surface area $A$ of the $KO_2$ powder, needed for the calculations, was determined by analysing optical images (Extended Data Fig. 3d). $KO_2$ powder was dispersed in TEGDME, sonicated, a drop was placed between a microscope slide and a cover slip and sealed air-tight. Images were acquired with transmitted light on a Nikon Ti2E-01 inverted microscope using a Plan Apo λ 40×/0.95 DIC (Differential Interference Contrast) air PFS (Perfect Focus System) objective lens, resulting in a pixel size of 0.183 μm. The images were analysed using the ilastik pixel classification and object classification workflows (https://www.ilastik.org/), resulting in a histogram of particle sizes (Extended Data Fig. 3e). The surface area was calculated by assuming spherical particles, yielding $A = 0.23 \pm 0.04$ m$^2$ g$^{-1}$.

As quantification of absolute $^1O_2$ yields from NIR emission is not straightforward, we normalized the values. The NIR intensity at time $t$ after bringing $KO_2$ in contact with $RM^{ox}$ will be proportional to the $^1O_2$ concentration, $I_{1,270nm} \propto c_{1_{O_2}}(t)$. The $^1O_2$ formation rate by mediated $KO_2$ oxidation is $v_{1_{O_2}}(t) = A \cdot k_1 \cdot c_{RM^{ox}}(t)$, with $k_1$ being the rate constant as given in Fig. 2a. $A$ is constant because the experiments were performed with a large excess of $KO_2$ over $RM^{ox}$. $c_{RM^{ox}}$ and $c_{RM^{red}}$ are given by $c_{RM^{ox}}(t) = c_{RM^{ox}}(0) \cdot e^{-k_{tot} \cdot A \cdot t}$ and $c_{RM^{red}}(t) = c_{RM^{ox}}(0) - c_{RM^{ox}}(t) + c_{RM^{red}}(0) - v_{1_{O_2},q,r}(t) \cdot t$, where $k_{tot} = k_1 + k_3$. The term $v_{1_{O_2},q,r}(t)$ is the rate at which $^1O_2$ reacts with $RM^{red}$ as detailed below. $^1O_2$ formation and decay will balance[37], which results in the presence of solvent and mediator quenching in $c_{1_{O_2}}(t) = v_{1_{O_2}}(t)/(k_d + k_Q \cdot c_{RM^{red}}(t))$. $k_Q$ is either the measured value for the particular mediator or the trendline in Extended Data Fig. 3c. Finally, analogous to the experiment where we integrate the NIR signal until it ceases, we arrive at the expected NIR emission:

$$I_{1270nm,cum} \propto \int_0^\infty \frac{A \cdot k_1 \cdot c_{RM^{ox}}(t)}{k_D + k_Q \cdot c_{RM^{red}}(t)} dt. \tag{3}$$

To simulate losses of $^3O_2$ ($^3O_2/RM^{ox} < 1$ in Fig. 2b), we used the rate of reactive $^1O_2$ decay $v_{1_{O_2},r}(t) = (k_{d,r} + k_{q,r} \cdot c_{RM^{red}}(t)) \cdot c_{1_{O_2}}(t)$ to define $f_r$ as the reactive fraction of the total $^1O_2$ decay rate

$$f_r = \frac{k_{d,r} + k_{q,r} \cdot c_{RM^{red}}(t)}{k_D + k_Q \cdot c_{RM^{red}}(t)}. \tag{4}$$

$v_{1_{O_2},r}(t)$ equals the total mediated $^1O_2$ formation rate times $f_r$: $v_{1_{O_2},r}(t) = f_r \cdot A \cdot k_1 \cdot c_{RM^{ox}}(t)$. As some of the $RM^{red}$ reacts, its loss is accounted for using $v_{1_{O_2},q,r}(t) = k_{q,r} \cdot c_{RM^{red}}(t) \cdot c_{1_{O_2}}(t) = f_{q,r} \cdot A \cdot k_1 \cdot c_{RM^{ox}}(t)$, where $f_{q,r} = k_{q,r} \cdot c_{RM^{red}}(t)/(k_D + k_Q \cdot c_{RM^{red}}(t))$. Calculations of $f_r$, $f_{q,r}$ and $c_{RM^{red}}(t)$ were iterated until convergence was achieved. Finally, $^3O_2$ yields are

$$^3O_2/RM^{ox} = 1 - \int_0^\infty f_r \cdot A \cdot k_1 \cdot c_{RM^{ox}}(t) dt / \int_0^\infty A \cdot k_{tot} \cdot c_{RM^{ox}}(t) dt. \tag{5}$$

The fractions of reactive deactivation, $k_{d,r}/k_D$ and $k_{q,r}/k_Q$ were obtained by fitting the measured values of $^3O_2/RM^{ox}$ in Fig. 2b with the simulated ones from equation (5). The simulated $^3O_2/RM^{ox}$ are plotted as the dashed line in Fig. 2b.

Deviations between simulated and measured values may result from the simplicity of the model. The main simplification is as follows: (1) $k_Q$ was measured in homogeneous solutions of $RM^{red}$ and the model calculates bulk concentrations $c_{RM^{ox}}$ and $c_{RM^{red}}$, but during the heterogenous $KO_2$ oxidation, bulk and surface concentrations of $RM^{ox}$ and $RM^{red}$ differ. (2) We fitted a common fraction of mediator reactivity ($k_{q,r}/k_Q$) for all mediators, causing $k_{q,r}$ and $k_Q$ to decrease exponentially with growing $E^\circ_{RM^{ox/red}}$. As a trend, this is justified given that both reaction and charge transfer quenching require e$^-$ transfer, but individual

reactivity will vary because of the large chemical diversity of mediators[36]. (3) We did not account for quenching by $O_2$ and $O_2^-$. (4) Some RMs show further reactivities, which can cause $^3O_2$ loss: Fc$^+$ reacts with $O_2$ (ref. 38), TDPA gets in contact with $O_2$ spontaneously oxidized to TDPA$^+$ and TDPA$^{2+}$ (ref. 18), and reduced quinones bind to $O_2$ (ref. 39).

## Proton-induced disproportionation
Six types of buffers were used for proton-induced superoxide disproportionation. Britton–Robinson (BR) buffers were prepared with 0.1 M acetic acid, 0.1 M boric acid and 0.1 M phosphoric acid. NaOH (1 M) and HCl (1 M) were used to adjust the pH. Phosphate-buffered saline (PBS) solution and aqueous PBS powder solution (for pH 7.4) were added and the pH was adjusted using 1 M NaOH or 1 M HCl. Citrate buffers were prepared using citric acid and trisodium citrate dihydrate. To adjust the pH, we varied the concentrations of citric acid and trisodium citrate dihydrate. Acidic buffers of KCl and HCl (pH 1 and 2) were prepared by mixing 0.2 M KCl with 0.2 M HCl and adjusting the volume. Neutral to basic buffers of $KH_2PO_4$/NaOH (pH 7.1 and 10.8) were prepared by mixing 0.1 M $KH_2PO_4$ with 0.1 M NaOH and adjusting the volume accordingly. To trap the hydrogen peroxide ($H_2O_2$) produced, to eliminate the possibility of forming $^1O_2$ by peroxoacids[40], titanium(IV)oxysulfate solution ($TiOSO_4$, 15 wt% in dilute sulfuric acid), 1 M NaOH was used to adjust the pH of $TiOSO_4$ solutions to pH 1.2 and 1.55.

Figure 4c shows $^1O_2$ yields still increasing when the pH is below 4.8, the p$K_a$ of $O_2^-$. $KO_2$ hydrolysis increases the pH in aqueous media according to $KO_2 + H_2O \rightarrow HO_2 + K^+ + OH^-$. Buffer capacities were selected so that the amount of $KO_2$ did not significantly affect the resulting pH after the reaction. However, despite the buffers, the local pH at the reaction site will be higher than the average. The importance of buffering for a local pH close to the average is evident in control experiments without a buffer. Even at a pH of 1.5, we could not detect $^1O_2$ in an unbuffered $H_2SO_4$ solution, whereas we could in the buffered one (Extended Data Fig. 6e). Therefore, the $^1O_2$ yields in Fig. 4c result from a higher local pH.

## Driving forces on superoxide oxidation
Superoxide experiences a broad range of oxidizing conditions to liberate oxygen, but explanations for why and to what extent certain oxidizing redox couples evolve $^1O_2$ have been unknown. Extended Data Fig. 4 shows the driving forces for superoxide oxidation with various redox couples. The driving forces are shown in comparison with the Marcus kinetic parabola in ether and acetonitrile solvent from Figs. 1 and 2. Li$^+$- and H$^+$-induced disproportionation are shown in Figs. 3 and 4. The other examples we discuss in Extended Data Fig. 4 arise from superoxide in contact with $CO_2$ or organic peroxides, with relevance for energy storage and biology[1,30,41].

Considering $CO_2$ first, we have previously shown that $CO_2$ in contact with $O_2^-$ yields $^1O_2$, but the energetics were unknown[30]. $CO_2$ in contact with $O_2^-$ is known to form peroxomonocarbonates and peroxodicarbonates by repeated uptake of $CO_2$ by $O_2^-$. Intermediate peroxocarbonate species may be reduced by $O_2^-$, which releases $O_2$ (refs. 41–44). However, the $O_2$ spin state has previously not been considered. Extended Data Fig. 4c shows likely redox couples of oxidized/reduced peroxocarbonate species, but their redox potentials are not established experimentally. A previous study has shown, using density functional theory (DFT), that depending on the cation present and the solvent, the particular peroxocarbonate redox couples that oxidize $O_2^-$ to $O_2$ differ[4], but likely involve $CO_4^{\bullet-}/CO_4^{2-}$, $LiCO_4^\bullet/LiCO_4^-$, $Li_2C_2O_6/Li_2C_2O_6^-$ or $Li_3C_2O_6/Li_3C_2O_6^-$. The calculations have shown reaction energies relative to the $O_2^-/^3O_2$ between about 1 and 1.4 eV. These driving forces, hence, explain $^1O_2$ formation from $CO_2$ in contact with $O_2^-$ (Extended Data Fig. 4b).

$^1O_2$ formation has been examined for exposure of superoxide to organic peroxides, given their occurrence in biological systems[45]. No reaction was observed with alkyl peroxides, whereas acyl peroxides yielded $^1O_2$ (refs. 45,46); however, this has not been connected with

driving forces for superoxide oxidation. Nucleophilic attack of $O_2^-$ on acyl peroxides forms an acyl radical and carboxylate (Extended Data Fig. 4d). The potential for the acyl radical/carboxylate redox couple has been reported between 1.5 V and 1.7 V compared with SHE (ref. 23) (1.95–2.05 V compared with $O_2/O_2^-$), which explains $^1O_2$ formation according to Extended Data Fig. 4a. Alkyl peroxides (ROOR) such as di-tert-butyl, dicumyl and di-$n$-butyl peroxide have been reported not to form $^1O_2$ (ref. 45). This can be understood by ROOR only breaking down to $RO^-$ and $RO^{\cdot}$ at low potentials (−1.31 V to −1.15 V compared with SHE, that is, −0.86 V to −0.70 V compared with $O_2/O_2^-$) (ref. 47). The $RO^{\cdot}$/$RO^-$ couple (−0.06 V to 0.04 V compared with SHE, that is, 0.39–0.49 V compared with $O_2/O_2^-$) could, if formed, oxidize $O_2^-$, but not to $^1O_2$.

Proton-induced disproportionation that yields low but non-negligible $^1O_2$ yields at a high pH of about 11 is consistent with reliable theoretical and experimental works. Using a high-level ab initio method, a previous study[17] found that the reaction $HO_2 + O_2^- \rightarrow O_2 + HO_2^-$ can proceed by both the singlet and triplet pathways, with the singlet pathway being only about 0.3 eV endoergic. Equivalent experiments exposing $KO_2$ to $H_2O_2$ in toluene found about 0.2% $^1O_2$ production, as measured using a chemical trap[14]. In this experiment, $H_2O_2$ was the proton source to first form $HO_2$ ($H_2O_2 + O_2^- \rightarrow HO_2 + HO_2^-$). The conditions in these theoretical and experimental studies correspond to a pH of about 11 ($pK_a = 11.7$ for $H_2O_2$), in which the reaction to $^1O_2$ is only weakly driven. Conversely, $^1O_2$ was not detected with alkyl hydroperoxides ROOH as the proton source ($pK_a \approx 12.6$) (ref. 45,48), a result that insufficient driving forces can now explain.

## Relation between $E^\circ_{^1O_2/O_2^-}$ and the $^1O_2$ fraction

The electrochemiluminescence literature refers to energy-sufficient processes to form the electronically excited species[23,28,49]. For example, consider the generic redox couples $R/R^{\cdot-}$ and $M^{+}/M$ (note that these could be, for example, $^3O_2/O_2^-$ and $RM^{ox}/RM^{red}$). The process $R^{\cdot-} + M^{+} \rightarrow {}^1R^* + M$ is considered energy-sufficient to form the excited species $^1R^*$, if $(E^\circ_{M^{+}/M} - E^\circ_{R/R^{\cdot-}})F \geq -\Delta H^\circ ({}^1R^* \leftarrow R)$. This condition is fulfilled if the potential of the oxidizing redox couple exceeds the redox potential of the excited species: $E^\circ_{M^{+}/M} \geq E^\circ_{^1R^*/R^{\cdot-}} = E^\circ_{R/R^{\cdot-}} + \Delta H^\circ ({}^1R^* \leftarrow R)/F$. The connotation of energy-sufficient processes led to the interpretation that $E^\circ_{^1R^*/R^{\cdot-}}$ or $E^\circ_{^1O_2/O_2^-}$ establishes a threshold above which the excited species rather than ground state species forms[18,24,26]. Extended Data Fig. 4a shows that reaching this threshold potential (or the driving force for which this is exceeded) gives no indication about the extent to which $^1O_2$ rather than $^3O_2$ forms. An onset of $^1O_2$ may be expected at $\Delta G^\circ_{1 \leftarrow 3}$, but its formation will become significant only for driving forces $-\Delta G^\circ > \lambda_3$, for which $^3O_2$ formation slows down to benefit $^1O_2$ formation.

## Thermodynamics in mixed alkali metal/TBA$^+$ electrolytes

Superoxide disproportionation in Li$^+$ and Na$^+$ containing glyme electrolytes was found to always yield some $^1O_2$ according to $2MO_2 \rightarrow M_2O_2 + x\,{}^3O_2 + (1 − x)\,{}^1O_2$ (refs. 13,16,18,19). Li$^+$ yielded small fractions (about 2%) at large kinetics, and Na$^+$ yielded larger fractions (around 12%) at slow kinetics[16]. Often, electrolytes for non-aqueous metal-$O_2$ batteries contain weakly Lewis acidic cations, such as tetrabutylammonium (TBA$^+$) or other cations from ionic liquid electrolytes. In mixed Li$^+$/TBA$^+$ and Na$^+$/TBA$^+$ (1/1) electrolytes, the $^1O_2$ yields increased to about 20 and 18%, respectively. The reasons for this behaviour must, hence, lie in (1) already sufficient driving forces for $^1O_2$ formation in pure Li$^+$ and Na$^+$ electrolytes; and (2) increasing driving forces on adding TBA$^+$.

$M_2O_2$ (M = Li$^+$, Na$^+$) are insoluble[50,51] and the potential $E^\circ_{^3O_2/M_2O_{2(s)}} = \Delta_f G^\circ(M_2O_{2(s)})/F = 2.96$ V compared with M$^+$/M, therefore, fixed to the value obtained using the formation energy $-\Delta_f G^\circ$ of solid Li$_2O_{2(s)}$. Given that the superoxide/$M_2O_{2(s)}$ couple acts as the oxidant during superoxide disproportionation, the driving force is given by $-\Delta G = (E_{superoxide/M_2O_{2(s)}} - E_{^3O_2/superoxide})F$. Note that here superoxide does not denote a particular species, but it could be anything, including solid $MO_{2(s)}$, solvated $(M^+O_2^-)_{n \geq 1,\,(sln)}$ clusters and ion pairs, or the

weakly coordinated $(TBA^+O_2^-)_{(sln)}$. $E_{superoxide/M_2O_{2(s)}}$ cannot be directly measured but can be inferred from $E_{^3O_2/M_2O_{2(s)}}$ and $E_{^3O_2/superoxide}$. Using $\Delta G = -zFE$, where $z$ is the number of transferred electrons, it can be derived that $E_{superoxide/M_2O_{2(s)}} = 2E_{^3O_2/M_2O_{2(s)}} - E_{^3O_2/superoxide}$. For the stable solid compounds (Li$_2O_2$, Na$_2O_2$, NaO$_2$, K$_2O_2$ and KO$_2$), tabulated formation energies $\Delta_f G^\circ$ can be found and the $E^\circ_{^3O_2/M_2O_{2(s)}}$ and $E^\circ_{^3O_2/MO_{2(s)}}$ be calculated as shown in Extended Data Fig. 5a. On the basis of this, KO$_2$ is not expected to disproportionate to K$_2O_2$ and the K$^+$-case, hence, not be further considered.

$E_{^3O_2/superoxide}$ require further consideration given the electrolyte-dependent solubilities of superoxide. Theoretical work shows that solvated $(Li^+O_2^-)_{n \geq 1,\,(sln)}$ species are less stable in terms of Gibbs free energy than the bulk solid LiO$_{2(s)}$, but as the cluster size $n$ grows, the structure approaches bulk $MO_{2(s)}$ and free energy approaches a constant value[16,50,51]. Aggregation into $(Li^+O_2^-)_{n > 1,\,(sln)}$ clusters stabilizes the solvated species relative to separated $(Li^+O_2^-)_{(sln)}$ species[16,34]. The Gibbs free energy grows, therefore, in the order of increasing solvation: $LiO_{2(s)} < (Li^+O_2^-)_{n > 1,(sln)} < (Li^+O_2^-)_{(sln)}$. Accordingly, the potentials in pure M$^+$ electrolyte are in the order and within the limits $E_{^3O_2/LiO_{2(s)}} > E_{^3O_2/(Li^+O_2^-)_{n > 1(sln)}} > E_{^3O_2/(Li^+O_2^-)_{(sln)}}$. If TBA$^+$ is present, the even weaker association in $(TBA^+O_2^-)_{(sln)}$ extends the lower potential limit to $E_{^3O_2/(TBA^+O_2^-)_{(sln)}}$. The values for these potential limits, as shown in Fig. 3a, Extended Data Fig. 5c,d, were estimated from cyclic voltammograms with the salt shifting from pure M$^+$ to pure TBA$^+$ (Extended Data Fig. 5b). The largely solvation-independent DMFc/DMFc$^+$ redox couple was used as internal standard. $\Delta_f G^\circ$ of solid LiO$_{2(s)}$ is not available, but $E_{^3O_2/LiO_{2(s)}}$ may be estimated from cyclic voltammograms in poorly solvating electrolytes, in which large $(Li^+O_2^-)_{n,\,(sln)}$ clusters approach the thermodynamics of LiO$_{2(s)}$. The shift in the onset of $O_2$ reduction in Li$^+$ compared with TBA$^+$ electrolyte at slow scan rates was taken as the difference between $E_{^3O_2/LiO_{2(s)}}$ and $E_{^3O_2/(TBA^+O_2^-)_{(sln)}}$. $E_{^3O_2/(Li^+O_2^-)_{(sln)}}$ as the lower limit of potentials in pure Li$^+$ electrolyte was estimated from the potential shift between pure Li$^+$ and pure TBA$^+$ electrolytes with the highly solvating solvent 1-methylimidazole. These were taken from ref. 32 and show a shift of 33 mV.

Extended Data Fig. 5d shows the analogous thermodynamics for Na$^+$/TBA$^+$ mixtures as shown in Fig. 3 for Li$^+$/TBA$^+$. Considerations as above for the relative stabilities of the Li superoxides apply analogously to the relative stability of Na superoxide clusters compared with the NaO$_{2(s)}$ bulk. Theoretical work has similarly shown the stabilization by forming $(Na^+O_2^-)_{n > 1,\,(sln)}$ clusters[52,53]. Extended Data Fig. 5e shows the NIR emission on adding KO$_2$ into glyme electrolyte containing various Na$^+$/TBA$^+$ ratios. The data show that, even in a pure Na$^+$ electrolyte, the driving force is sufficient for $^1O_2$ formation and that added TBA$^+$ increases the driving force and formation.

More generally, the electrolyte properties (solvent(s), salts(s) and their concentrations) will affect the reorganization energy and hence the maxima and crossing point of the two kinetic parabolas. The classical approach to accounting for this is the equation given by Marcus[54], which connects the reorganization energy with the effective dielectric properties of the electrolyte and the separation of the redox centres. A lower dielectric constant and smaller separation will result in a larger reorganization energy. A refined equation by Marcus[55] further takes into account the ionic environment. These considerations apply well, for example, to aqueous anionic redox couples and the series of alkali metal cations from Li$^+$ to Cs$^+$ as spectator cations, in which $\lambda$ decreases[56]. However, caution is required with non-aqueous, low dielectric constant media, in which strong ion pairing occurs. Some works suggest an inverse trend for $\lambda$ among the alkali metals[19,57]. Ion pairing and even clustering is particularly severe for (su)peroxide as the redox anions as discussed above. Superoxide forms in non-aqueous Li$^+$ and Na$^+$ electrolytes clusters[32,33] and the peroxides are practically insoluble[51]. The order and extent to which the reorganization energy changes for superoxide oxidation in non-aqueous media among the alkali cations

may, therefore, not be predicted straightforwardly and would merit further investigation. As we observe $^1O_2$ at low driving force for the $Na^+$ electrolyte, the reorganization energy appears sufficiently low therein.

### Wider relevance for life sciences and energy

Our study contributes to understanding how the pH affects the link between the four important reactive oxygen species (ROS) superoxide, peroxide, $^3O_2$ and $^1O_2$. Disproportionation is notably the pathway to maintain a low superoxide concentration. However, detoxification from superoxide produces the harmful $^1O_2$. Superoxide occurs in cells in several organelles with different pH levels between 4.7 and 8, but the superoxide-degrading enzyme superoxide dismutase occurs only in neutral to basic organelles[35]. In pathological situations, the pH balance is known to be affected (typically towards lower pH) and therefore signalling, redox regulation and defence[58,59]. Our study contributes to the understanding of how the redox chemistry of superoxide, pH and $^1O_2$ formation are linked. We noted that in non-aqueous media, superoxide in contact with $CO_2$ forms $^1O_2$. Given the ubiquity of $CO_2$ in organelles containing superoxide, further investigations into the aqueous chemistry of $CO_2$ and superoxide are warranted.

For energy applications, further relevance and future research directions emerge: (1) For suppressing $^1O_2$, generally, the driving force should be decreased, and the reorganization energy for the superoxide oxidation reaction should be increased. The classical equation by Marcus[54], which connects the reorganization energy with the effective dielectric properties of the electrolyte and the separation of the redox centres, applies well to aqueous anionic redox couples[56]. For non-aqueous, low dielectric constant media, in which strong ion pairing occurs, particularly so with superoxide, the change in reorganization energy among different cations and electrolytes may not be predicted straightforwardly and would merit further investigation. (2) Oxygen evolution catalysis from water: metal-superoxo species have been identified as preceding the $O_2$ evolution on, for example, the extensively studied Ni(Fe)OOH or CoOOH catalysts[10]. The metal $M^{n-1/n}$ redox couple is considered to oxidize the superoxo moiety to $O_2$ (refs. 10,60). Some of the most active $M^{n-1/n}$ metal redox couples are typically a few 100 mV above the $^1O_2$/superoxide potential shown in Fig. 4a and provide, in principle, enough driving force for $^1O_2$ evolution. For example, at pH 14 $E^\circ_{^1O_2/O_2^-} = 1.32$ V, $E^\circ_{Co^{II/III}} \approx 1.25$ V, $E^\circ_{Co^{III/IV}} \approx 1.5$ V, $E^\circ_{Ni^{II/III}} \approx 1.52$ V and $E^\circ_{Ni^{III/IV}} \approx 1.6$ V on the RHE scale (refs. 10,60). Further investigations specifically on $^1O_2$ evolution in oxygen evolution catalysis are therefore warranted. (3) Both Li-stoichiometric[6] and Li-rich transition metal (TM, for example, Ni, Mn and Co) oxide[2,7,61,62] intercalation materials used for positive electrodes in Li- or Na-ion batteries are known to undergo parasitic lattice oxygen loss at high states of charge. Both the intercalation material and the electrolyte degrade, hampering long-term cyclability. However, non-matching patterns of $O_2$ and $CO_2/CO$ release from electrolyte decomposition (all containing lattice O as shown by isotopic labelling[7,62]) and enhanced lattice O loss with surface carbonates present[7] remain unexplained and highlight the need for a deeper understanding of the prevailing ROS and decomposition pathways. For LiNiO$_2$, $^1O_2$ evolution has been suggested to result from the disproportionation of oxide radicals[6]. More generally, $^1O_2$ may evolve from superoxo species (at the lattice surface, in (su)peroxocarbonates[4,30]) at the available driving forces. The oxidants could be a combination of (su)peroxides (for example, coordinated by TMs or carbonates) that get stabilized by further reduction and TM redox, such as $Co^{III/IV}$ or $Ni^{III/IV}$. Further investigations into the involvement of $^1O_2$ evolution in TM oxide outgassing and surface reactions are therefore warranted.

The results expand the current knowledge on the electrogeneration of excited species more generally and pose open questions about the origin. Specifically, the process is more effective than typically assumed, given that $\Delta G^\circ_{1\leftarrow3} < \Delta H^\circ_{1\leftarrow3}$ and that $Z_{el,1} \gg Z_{el,3}$. Electrogeneration of excited ROS has significance ranging from biological systems to energy storage. Reactive excited species in life are very broadly associated with pathogenic events[1,63]. Recombination reactions in redox flow batteries are recognized to cause self-discharge, but this has so far not been recognized to potentially form extremely energetic excited species. Soluble parasitic oxidized and reduced species at the cathode and anode of Li- and Na-ion batteries may recombine to form energetic excited species.

## Data availability

The data that support the plots in this paper and other findings of this study are available from the corresponding author upon reasonable request. Source data are provided with this paper.

## Code availability

Figures were made using MATLAB and the built-in toolboxes. Simulations in Fig. 2 were done following the model and equations given in the Methods. The scripts used in this study are available from the corresponding author upon reasonable request.

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

**Acknowledgements** S.A.F. thanks the Institute of Science and Technology Austria (ISTA) for the support. The Scientific Service Units of ISTA supported this research through resources provided by the Imaging and Optics Facility, the Lab Support Facility, the Miba Machine Shop and Scientific Computing. This research was partly funded by the Austrian Science Fund (FWF) (10.55776/P37169 and 10.55776/COE5). For open access purposes, the author has applied for a CC BY public copyright licence to any author-accepted manuscript version arising from this submission. R.H. acknowledges funding through CZI grant DAF2020-225401 (10.37921/120055ratwvi) from the Chan Zuckerberg Initiative DAF, an advised fund of Silicon Valley Community Foundation (10.13039/100014989). H.T.K.N. acknowledges funding by the European Commission Erasmus Mundus Joint Masters programme. We thank M. Sixt and M. Chinon for the discussions about O-redox in life and R. Jethwa for proofreading. Open access funding was provided by ISTA.

**Author contributions** S.A.F. and S.M. conceptualized the project and developed the methodology for all experiments, developed the model and code for data evaluation and performed simulations. S.M. and H.T.K.N. carried out electrochemistry and kinetics measurements. R.H., S.M. and S.A.F. developed the NIR setup, and S.M. carried out the chemiluminescence and time decay study. S.M. carried out the mass spectrometry. S.A.F. conceived and supervised the project, secured funding and wrote the paper. S.A.F. and S.M. evaluated and interpreted the results and revised the paper.

**Funding** Open access funding provided by Institute of Science and Technology (IST Austria).

**Competing interests** The authors declare no competing interests.

**Additional information**
**Correspondence and requests for materials** should be addressed to Stefan A. Freunberger.

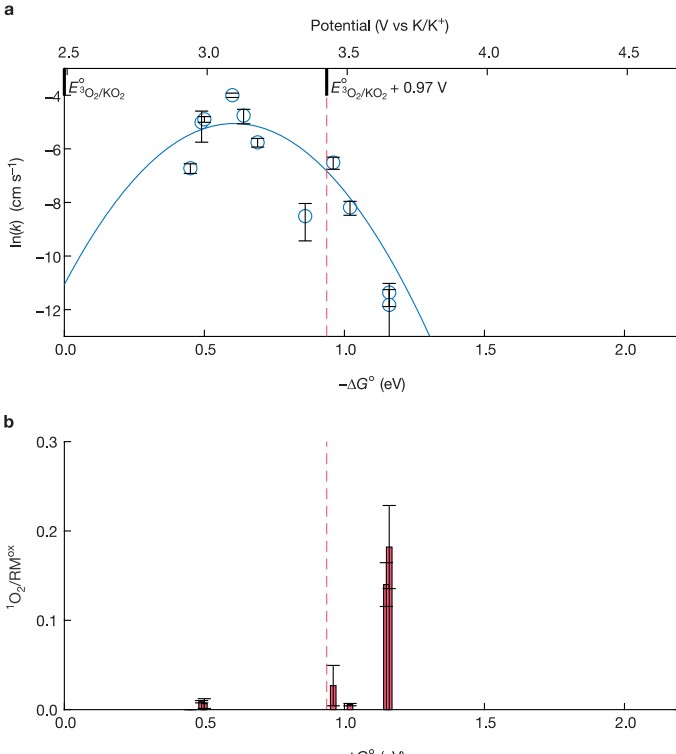

**Extended Data Fig. 1 | Kinetics and $^1O_2$ yields during mediated superoxide oxidation in TEGDME electrolyte. a**, We measured the kinetic constants $k$ for mediated $KO_2$ oxidation with mediators using UV-Vis spectroscopy up to moderate mediator redox potentials. Higher potential mediators could not be used in the tetraglyme electrolyte, and errors were already significant at the upper end of driving forces. Plot of $\ln(k)$ versus mediator potential ($E^\circ_{RM^{ox/red}}$, top axis) and driving force ($-\Delta G^\circ$, bottom axis). $-\Delta G^\circ = (E^\circ_{RM^{ox/red}} - E^\circ_{^3O_2/KO_2})F$, where $E^\circ_{^3O_2/KO_2} = 2.48$ V vs K/K$^+$. The blue curve is a fit to Marcus kinetics in Eq. (1), which gives $Z_{el} = 6.4 \times 10^{-3}$ cm s$^{-1}$ and $\lambda = 0.61$ eV. The red dashed line indicates the driving force at $E^\circ_{^1O_2/KO_2} = E^\circ_{^3O_2/KO_2} + 0.97$V, the commonly expected "threshold" for $^1O_2$ formation. **b**, The formed $^1O_2$ per mole of RM$^{ox}$ as measured using 9,10-dimethylanthracene (DMA) conversion to DMA-$O_2$ using HPLC. Data are presented as mean ± s.d. ($n \geq 3$). Panel **a** and **b** adapted from ref. 18, Springer Nature Limited.

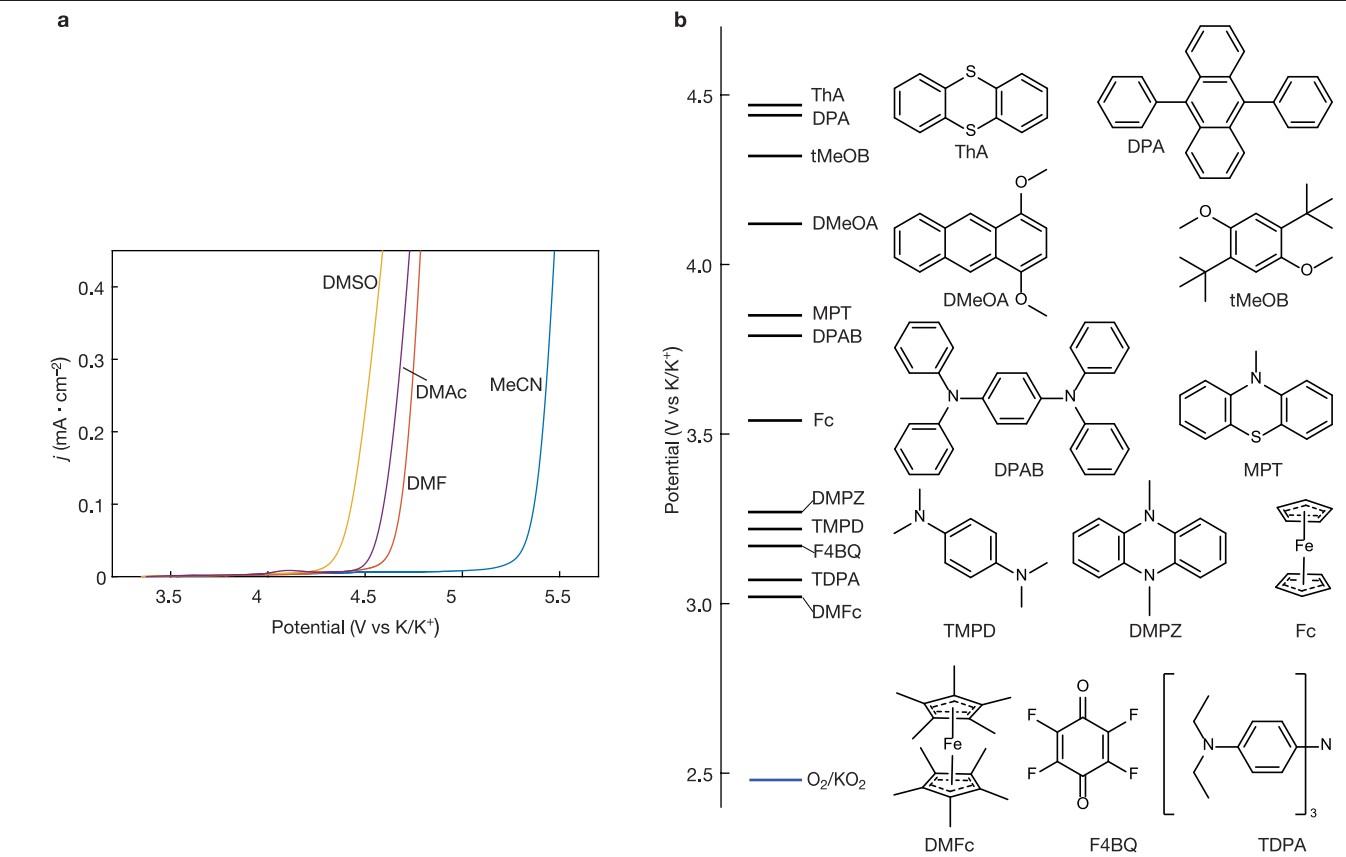

**Extended Data Fig. 2 | Oxidation stability of different solvents and potentials of the redox mediators used here. a**, Forward scan of cyclic voltammetry of dimethyl sulfoxide (DMSO), N,N-dimethylacetamide (DMAc), N,N-dimethylformamide (DMF), and acetonitrile (MeCN) containing 10 mM KClO$_4$ The scan rate was 20 mVs$^{-1}$ and the working electrode was a glassy carbon disc. **b**, The selected RMs, their abbreviations and measured redox potentials on the K/K$^+$ scale as well as $E°$ of the O$_2$/KO$_2$ couple.

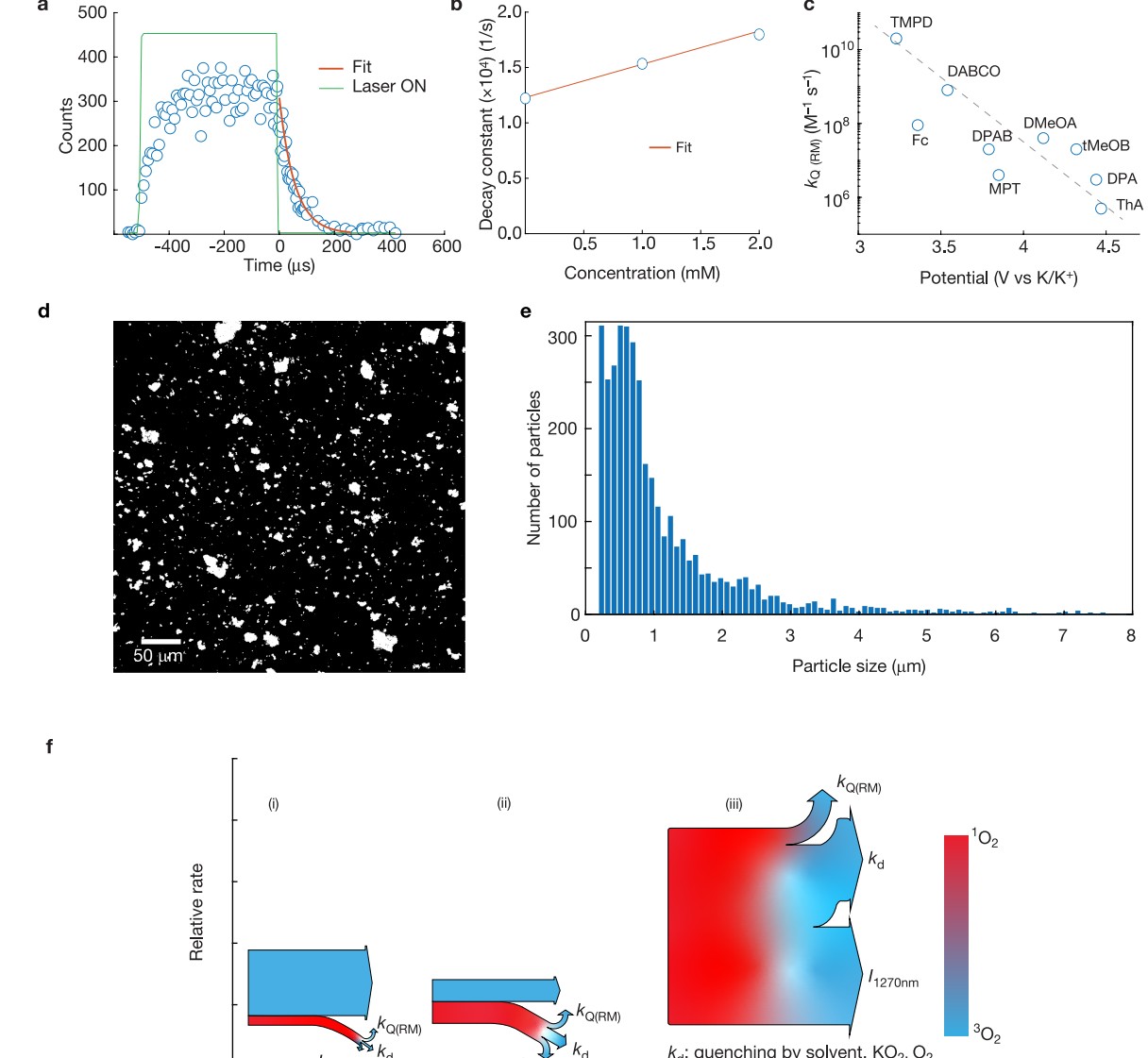

**Extended Data Fig. 3 | From $^1O_2$ formation rates to measurable NIR emission intensities. a**, Example for one mediator of the decay of the $^1O_2$ phosphorescence at 1270 nm in MeCN after pulsed irradiation. The pulse frequency was 1 kHz and the signal was accumulated for 2 min. **b**, Example for one RM of the decay constant of $^1O_2$ versus $RM^{red}$ concentration to get $k_{Q(RM)}$. Every value at any concentration is the mean of $n \geq 3$ decay measurements. Error bars are too small to be visible. **c**, Measured $^1O_2$ quenching rate constants $k_{Q(RM)}$ of the used mediators in their reduced form $RM^{red}$ as a function of their redox potential. The trend line with a slope of $10^{3.5}$/V has been added to guide the eye. Each marker represents one set of measurements as shown in **b**. **d**, Segmented optical microscopy image of dispersed $KO_2$ powder. **e**, Extracted particle size distribution from **d**, from

which a surface area of $A = 0.23 \pm 0.04$ m²g⁻¹ (mean ± s.d., $n = 3$) was determined. **f**, Schematic Sankey plots of relative rates of the involved processes for the examples of Fc, MPT and ThA. The widths of the arrows are proportional to the rates. Relative rates at the left end are those given by the Marcus parabola for superoxide oxidation to $^3O_2$ and $^1O_2$ in Fig. 2a. Any $^1O_2$ then undergoes various decay pathways. First, a combination of physical and reactive quenching by the solvent, $^3O_2$ and superoxide, as denoted as $k_d$. Second, strongly potential-dependent charge transfer quenching by the RM denoted as $k_{q(RM)}$. The fraction emitting NIR radiation is strongly exacerbated as it would otherwise be invisible relative to the others. Data are presented in **b** and **c** are from single experiment.

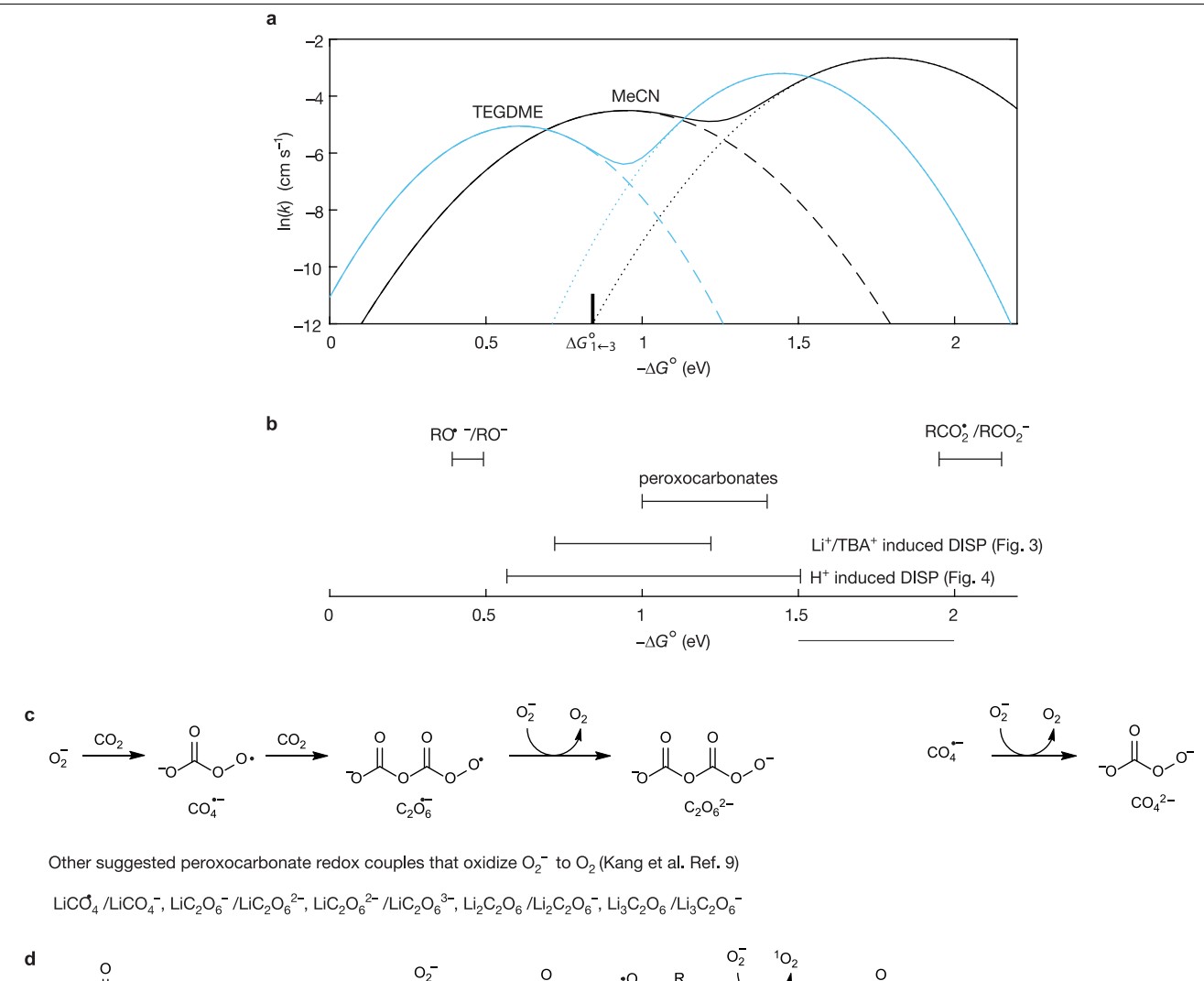

**Extended Data Fig. 4 | Ranges of driving forces for various systems, where superoxide oxidation occurs. a**, Comparison of individual kinetic parabolas as determined for mediated superoxide oxidation. Curves for MeCN are the Marcus fits in Fig. 2. For TEGDME the $^3O_2$ parabola is the Marcus fit to data in Extended Data Fig. 1. The parabola for $^1O_2$ evolution could not be measured in TEGDME, and is hence constructed using $\Delta G^{\circ}_{1\leftarrow3} = 0.84$ eV and $Z_{el,1} = 6.3 \times Z_{el,3}$ as determined with MeCN (Fig. 2). **b**, Ranges of driving forces for cation-induced $O_2^-$ disproportionation (DISP) and for the indicated redox couples as defined in

c and d. **c**, Commonly suggested peroxocarbonate species that form from $O_2^-$ in contact with $CO_2$ (refs. 41–44). For example, the superoxo species $CO_4^{\bullet-}$ or $C_2O_6^{\bullet-}$ were suggested to be reduced by $O_2^-$, liberating $O_2$. Kang's group[4] suggested using DFT further peroxocarbonate redox couples and their redox potentials relative to $O_2^-/^3O_2$ (shown in b). **d**, Reactions of acyl and alkyl peroxides in contact with $O_2^-$ which were found to form or not to form $^1O_2$ (refs. 45,46) as explicable by their reduction potentials (shown in b).

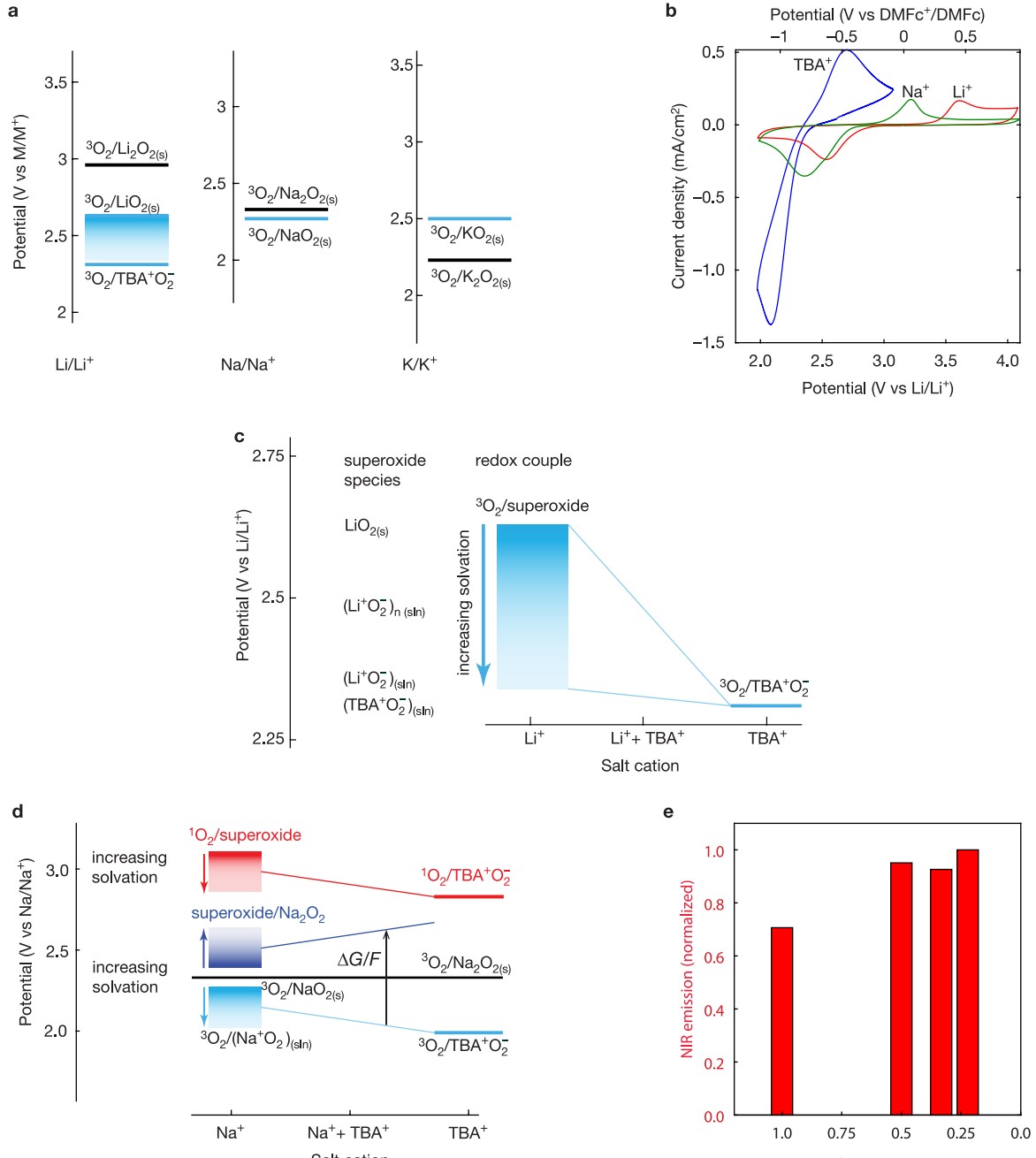

**Extended Data Fig. 5 | Redox potentials in mixed Li⁺/TBA⁺ and Na⁺/TBA⁺ electrolytes. a**, Standard potentials of the $O_2/MO_{2(s)}$ and $O_2/M_2O_{2(s)}$ redox couples with M = Li, Na, K as calculated from tabulated formation energies. The common scale is based on their $E°_{M/M^+}$. **b**, Cyclic voltammograms (scan rate 100 mVs⁻¹) in $O_2$-saturated tetraglyme containing 100 mM of either TBAClO₄, LiClO₄, or NaClO₄. **c**, Detailed thermodynamics of the $^3O_2$/superoxide couple as a function of the fractions of Li⁺ and TBA⁺ salt. **d**, Thermodynamics of relevant redox couples for Na⁺-induced $O_2^-$ disproportionation as a function of the fractions of Na⁺ and TBA⁺ salt. **e**, Normalised 1270 nm NIR emission signal as a function of Na⁺ mole fraction, which determines the driving force. Each bar represents a single measurement. Tetraglyme served as the electrolyte solvent.

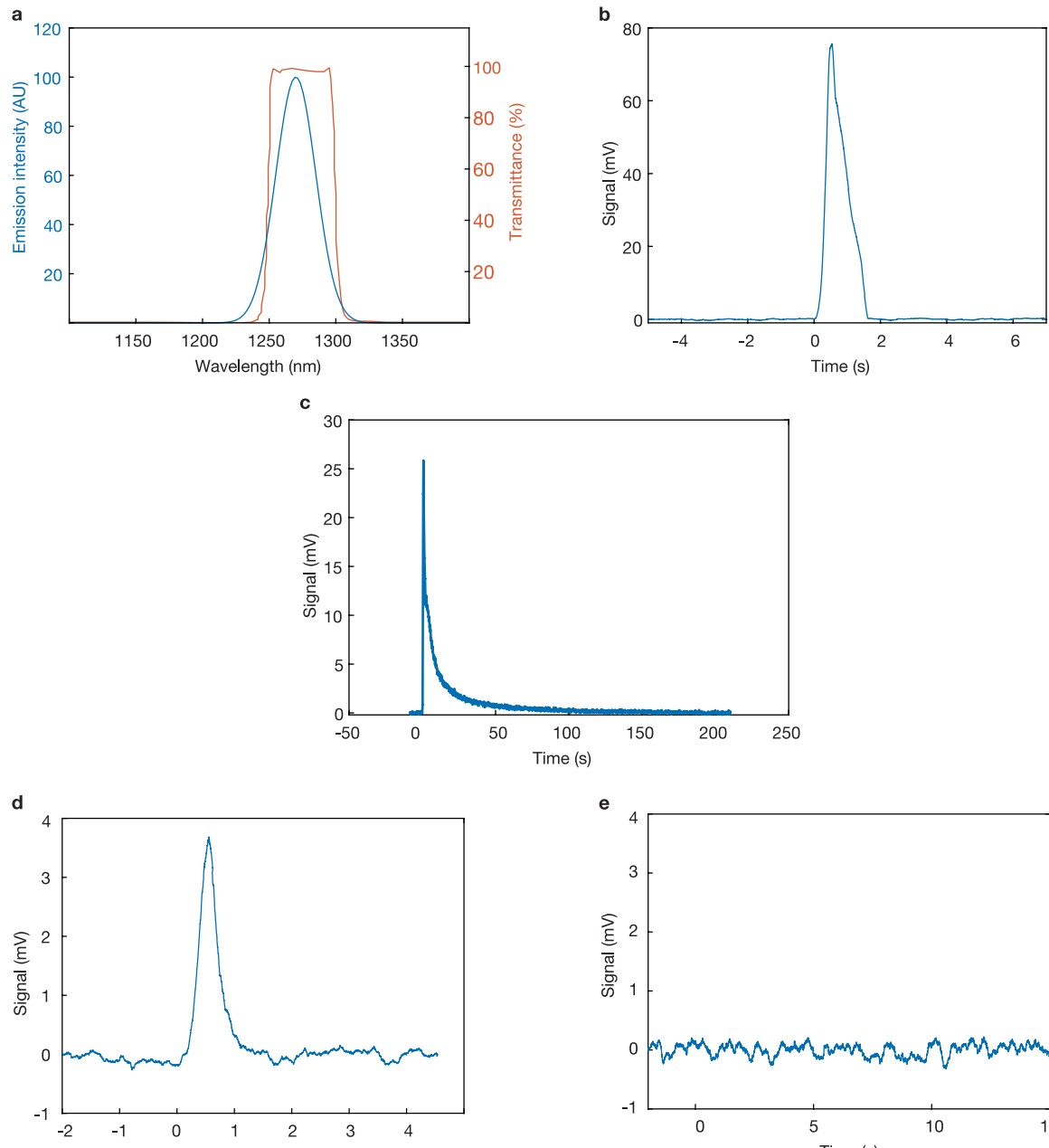

**Extended Data Fig. 6 | NIR emission to measure $^1O_2$ yields. a**, $^1O_2$ NIR emission spectrum as reported in the literature[12] and transmission of the used bandpass filter. **b**, Representative output signal upon injecting RM$^{ox}$ solution onto KO$_2$ powder. The integral of the recorded signal was used to determine the $^1O_2$ yields. The example shown is with MPT. **c**, Representative output signal of the NIR detector upon injecting Li$^+$ electrolyte solution onto KO$_2$ powder. The setup was similar to the one shown in Extended Data Fig. 4, except that a cuvette replaced the glass vial and the NIR detector was placed next to it. The example shown is with a Li$^+$ mole fraction of 0.166. **d**, Representative NIR signal upon injecting H$^+$ solution containing buffer onto KO$_2$ powder. The example shown is with BR buffer at pH 2.2. **e**, Output signal of the NIR detector upon injecting H$_2$SO$_4$ (pH 1.58) solution without any buffer onto KO$_2$ powder.

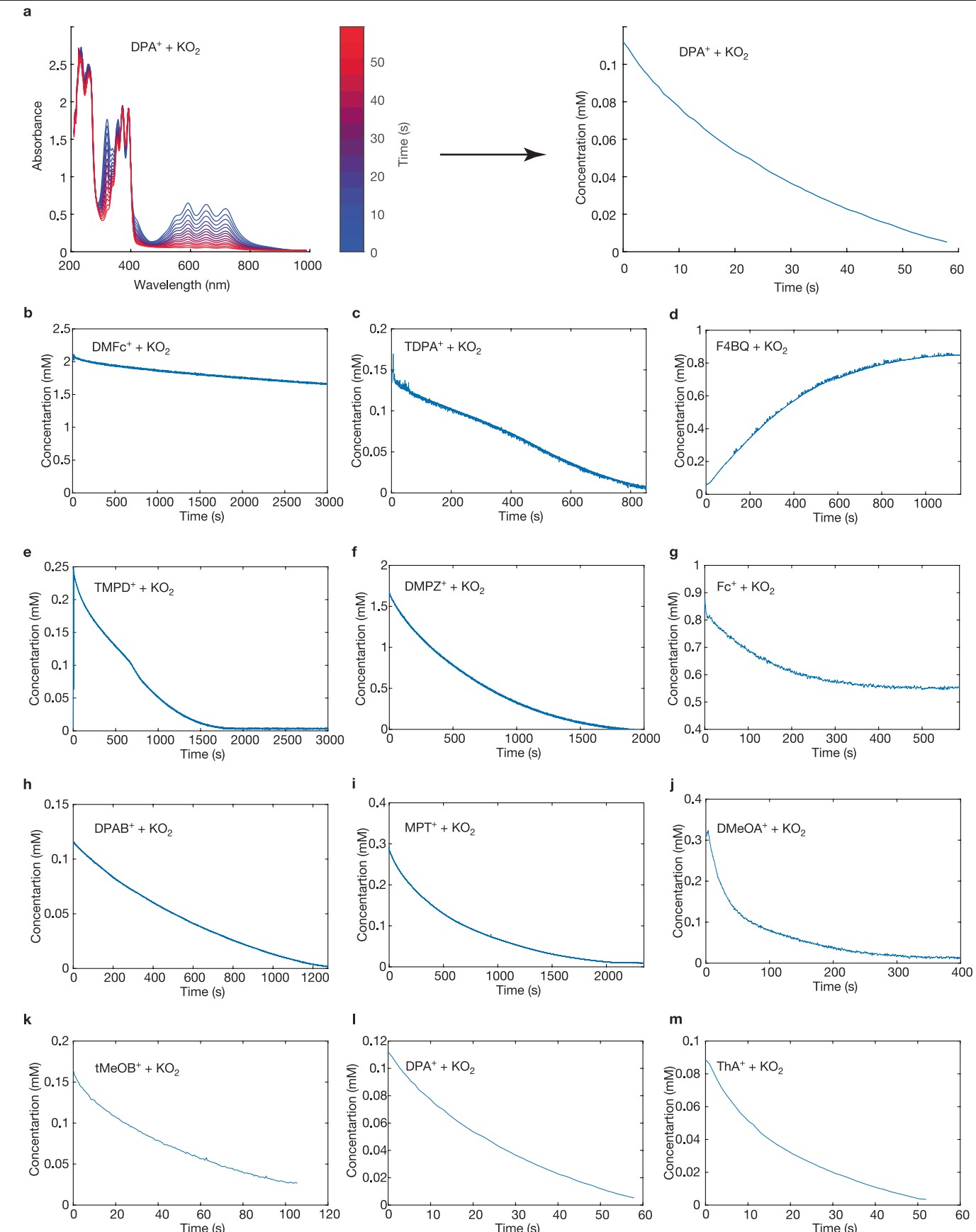

**Extended Data Fig. 7 | Measuring KO₂ oxidation kinetics using UV-Vis absorbance. a**, Exemplary evolution UV-Vis spectra of DPA and ThA. Concentration versus time for the indicated RM^ox in contact with KO₂. **b,c,e–m**, RM^ox concentration versus time. For F4BQ (**d**), concentration versus time for RM^red.

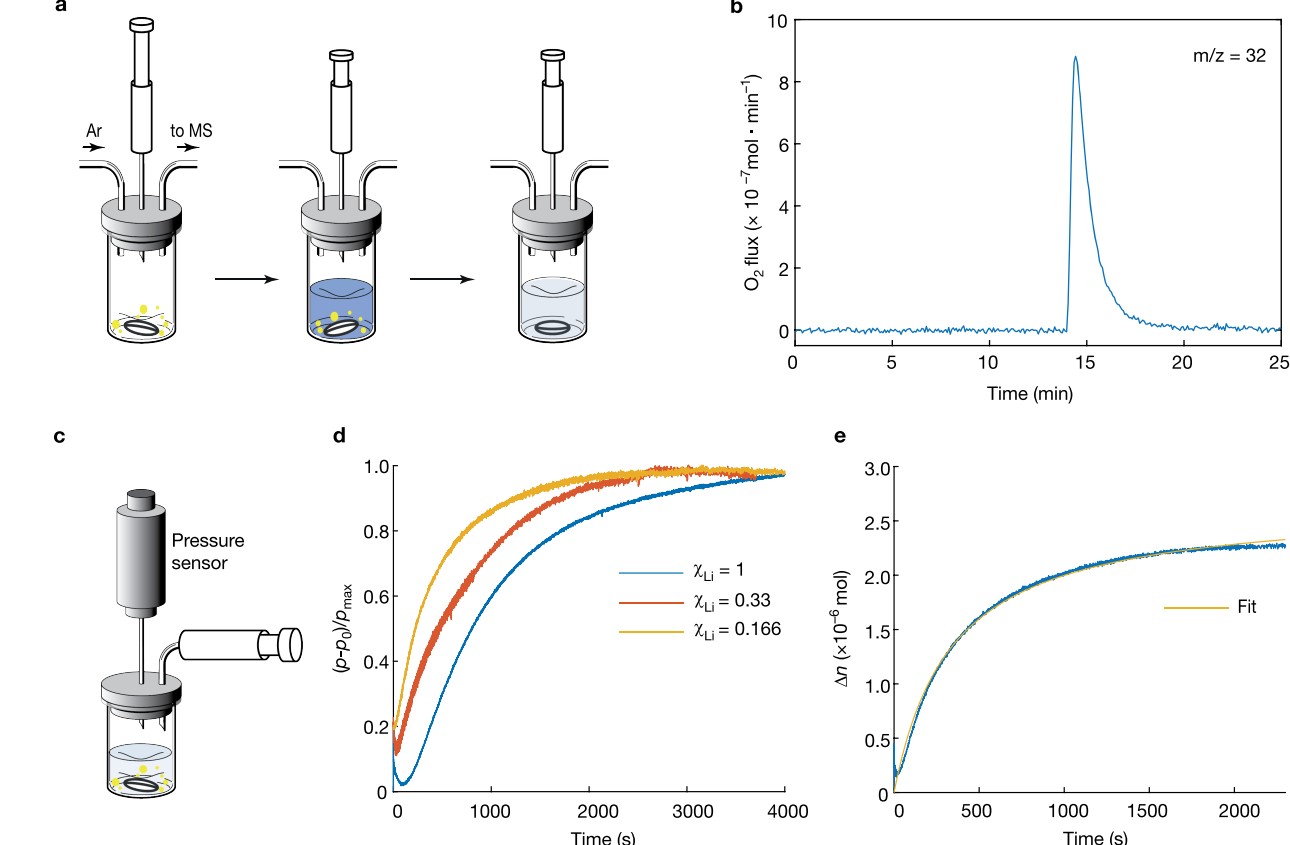

**Extended Data Fig. 8 | Measuring $^3O_2$ yields and disproportionation kinetics.** **a**, Schematic set-up of the sample vial used for mass spectrometry and sequence of the solution addition during the $^3O_2$ yield measurement. **b**, $O_2$ evolution versus time upon immersing $KO_2$ powder in $RM^{ox}$ solution. The example shown is with DMPZ. **c**, Schematic of the set-up to measure disproportionation kinetics by pressure changes. **d**, Pressure evolution from three different cases where $Li^+/(Li^++TBA^+)$ mole fractions are 1, 0.33 and 0.166. **e**, Evolved moles versus time for superoxide disproportionation as measured by the pressure $p$. Fitting is done with $p(t) = p_0 + \Delta p(1-\exp(-kt))$. The example is shown for the case with the mole fraction $Li^+/(Li^++TBA^+) = 0.166$. Panel **a** adapted with permission from ref. 64, American Chemical Society.