## [Peer Review File · Nature]

Marcus kinetics control singlet and triplet oxygen evolving from superoxide

Corresponding Author: Professor Stefan Freunberger

Version 1:

Reviewer comments:

Referee #1

(Remarks to the Author)

Individual Marcus-type kinetics controls singlet and triplet oxygen evolution from superoxide

Freunberger and coworkers have submitted a paper to Nature in which they shed light on the formation of highly reactive singlet oxygen, which is an important scientific question for the life science and energy communities. Their hypothesis is that the driving force (dG) governs the formation of singlet oxygen, which is explained by a Marcus-type kinetics. Using the superoxide KO_2 , the authors demonstrate that for low anodic potentials (w. r. t. K/K^+), triplet oxygen is formed. With increasing anodic potential, the rate constant for the formation of triplet oxygen drops down due to a Marcus-inverted region. On the other hand, larger anodic potentials allow for the formation of the thermodynamically less stable singlet oxygen, which has a higher equilibrium potential (w. r. t. K/K^+) compared to triplet oxygen. Therefore, the rate constant for the formation of singlet oxygen rises with increasing anodic potential so that the formation of singlet oxygen is superior to that of triplet oxygen. The implication of this finding is that the driving force (dG) must be controlled to produce either triplet oxygen or singlet oxygen.

Without further ado, the results reported by the authors are groundbreaking and fall within the scope of the Nature journal. I generally support the publication of the authors' work, although I ask them to address the following questions and comments before publication can potentially take place:

i) I must admit that it was hard to grasp figure 1 based on the calculated $\ln k$ values at first glance, considering that equation (1) was introduced after the figure was presented. However, the authors did not discuss which experimental technique was used to quantify the rate constants k in their analysis, at least not in the context of this figure. This requires further clarification and discussion in the manuscript. It is also unclear for the reader which technique was used to detect 3O_2 or 1O_2 to distinguish between these chemical species. After reading the manuscript further, I understood that mass spectrometry and NIR emission was used, respectively (lines 147-149). In any case, the didactics can be further improved in the discussion of the results. In connection with Figure 1, several questions arise which are addressed later in the text.

The error bars of the $\ln k$ values differ to quite some extent. Although I believe that not all of this information needs to be included in the main text, an explanation of how these error bars are determined and the reason for their different sizes should be part of the supplementary notes. I am also not sure whether the data shown in figure 1 was measured for the current manuscript or whether it is taken from a previous work of the authors in Nature Chemistry. The data shown in Figure 1 and Supplementary Figure 1 are essentially the same, with the only difference of the red bars quantifying the formation of 1O_2 . Therefore, in the presentation and discussion it remains somewhat hidden which data are new and which come from previous work by the authors. To be eligible for publication in Nature, a significant amount of new data should be available compared to the authors' previous publications in Nature Chemistry.

Further confusion arises because the authors state in lines 69-70: "As we reached the higher end of the investigated driving forces, 1O_2 evolution became significant." This indicates that the authors quantified 1O_2 evolution by their experimental setup (as shown in Supplementary Figure 1), although they present a hypothesis in lines 90-91. I would have expected to discuss the hypothesis before providing any experimental proof.

In short, the intermingling of hypothesis and experimental quantification as well as old and new data needs to be resolved.

ii) While the developed theory in lines 96-113 is beautiful, what is the scientific argument for using the same reorganization energy for the formation of $1O_2$ as for the formation of $3O_2$? ($\lambda \equiv \lambda_3$) This question also arises in connection with lines 131-132, where it becomes clear that this assumption is not entirely fulfilled.

iii) Supplementary Figure 2: Given that the discussion of the manuscript relates to U vs. K/K+, it would be helpful for the reader if the redox potentials of the redox mediators are also shown on the K/K+ scale.

iv) Lines 190-195: this part is hard to grasp. To which calculations do the authors refer in line 193?

v) In the field of batteries, potassium is not the main ion used for battery applications. To date, most batteries are still based on lithium-ion or sodium-ion technologies. This is touched in the section "Disproportionation in non-aqueous systems". While the authors already discussed that the crossing point of the two parabolas is not at a constant driving force (Extended Data Fig. 2), I am wondering whether a change in the ion has also an effect on the crossing point of the two parabolas. How do the thermodynamic correlations as discussed in Figure 3a look like for Na- or K-based systems? Further discussions could lead to a generalization of the results to other battery technologies.

vi) The statement in line 246-247 is unclear because the authors compare the driving force (dG) to an equilibrium potential. Do the authors intend to say that in the oxygen evolution reaction (OER), $1O_2$ rather than $3O_2$ is formed for $pH < 10$? Considering that the OER requires significant overpotentials of at least 300 mV to reach current densities in the order of 10 mA/cm², which part of the two parabolas does this overpotential regime represent? I think that the implications for electrocatalysis should be further discussed.

vii) A general question is, how can the driving force in life sciences or energy applications be controlled to render the formation of either $3O_2$ or $1O_2$? Can the authors provide recommendations or guidelines?

Referee #2

(Remarks to the Author)

In this paper Freunberger et al. explore the reactivity of superoxides in forming oxygen molecules both in their ground state and in their first excited state (singlet oxygen). While the thermodynamic of the superoxide \rightarrow oxygen reactions is easily understood, their kinetic proved to be much harder to explore. Its study is however crucial to understand the behaviour of superoxide in electrochemical systems. The knowledge of the conditions for the formation of highly reactive oxygen species in battery chemistries can hardly be underestimated. In this respect this paper is a timely and compelling report of new and well devised experiments that systematically evaluate the behaviour of superoxides in a variety of chemical conditions. The authors show, with what appears to me a coherent set of unambiguous data, how singlet oxygen emerges from superoxides as a consequence of the strengthening of the oxidation driving force under electrochemical conditions. This paper does this in a very systematic way, thus (finally, I would add) providing a clear picture of the rates of the transformations of superoxides into oxygen (triplet and singlet). My opinion is that, given the relevance of the subject and the importance of such a systematic study, the paper is suitable for Nature. There are few minor points that the authors may want to consider:

- The introduction does a good job in introducing the subject of reactive oxygen production, but there's no explanation as to why it represents a problem in electrochemical devices. I think the authors should provide a brief summary of what they recall in refs. 6 and 7.

- Figure 1 is very dense and the caption does not explain it properly. For example the color code for the potential energy surfaces in (c) is not reported. The blue and red vertical shaded areas are also without explanation. Finally, the origin of the data points is not reported.

- In Figure 2 the y-axes of panels (b) and (c) are two different things and this should be indicated by a change in the axis text labels.

- Near line 185: "They only cover smaller" is not clear who is "they".

- Please check that the text label of y-axis in Figure 3(b) is correct with respect to the caption.

- The conclusions section should summarise also the results for the disproportionation.

Referee #3

(Remarks to the Author)

Opinion: Reject

In this manuscript, the authors investigated the evolution of singlet oxygen (1O_2) and triplet oxygen (3O_2) through superoxide disproportionation. The authors present their findings on reaction kinetics, which are governed by driving force-controlled behavior in accordance with Marcus theory. The study examines the 1O_2 and 3O_2 evolution kinetics through the different formation energies of the oxygen species and proposes that their dual-parabola model provides clear explanations for discrepancies in oxygen redox systems and thus offers insights for controlling reactive oxygen species in both energy storage and biological contexts.

While the investigation of singlet and triplet oxygen dimer states is undoubtedly very important for understanding degradation behaviors in oxygen redox chemistry, the manuscript's primary arguments, regarding disproportionation kinetics and thermodynamics of superoxide into singlet/triplet oxygen molecules with peroxide species, do not present sufficient novelty. The relationship between singlet oxygen formation and superoxide disproportionation has been previously documented (Energy & Environmental Science, 2019, 12, 2559), as has the competition between singlet and triplet oxygen formation and its Marcus theory explanation (ACS Energy Letters, 2020, 5, 1893). Furthermore, the analysis in Figure 3 regarding bulk cation effects on driving force relies heavily on speculation rather than rigorous experimental evidence, largely reiterating previous work without substantial new insights (Energy & Environmental Science, 2019, 12, 2559).

I am also afraid that the manuscript does not adequately address a general topic suitable for a broad readership of Nature. While the authors suggest potential implications for life sciences, these connections need stronger development and the discussion of practical approaches to mitigate reactive singlet oxygen species should be expanded to encompass more broad aspects. Given these considerations, the paper may not currently align with the exceptional quality expected by Nature.

Below are the comments for the enhancement of the manuscript:

1. The manuscript provides several experimental points about superoxide disproportionation into the triplet molecule products in Fig. 1a, but it lacks the data points correlated with the formation of singlet oxygen molecule products.
2. The triplet and singlet oxygen molecule yields depending on the mediators suggested in Fig. 2b-c does not follow the expected values. Despite the possible reasons the manuscript suggests including the technical issues and overlooked decay pathways, the authors then should provide more accurate consensus by adopting other theoretical models beyond the renowned Marcus theory since the huge deviations of data significantly undermines the credibility of the manuscript.
3. The figures in the manuscript are not sufficiently self-descriptive and do not adequately explain the content. For instance, in Figure 1b, it is challenging to discern what each parabola represents and how elements (i), (ii), and (iii) influence them. Similarly, Figure 4 fails to intuitively convey the correlation between the Pourbaix diagram and singlet oxygen formation. Enhancing the figures with clearer labels and explanatory legends is necessary to improve comprehension.
4. The manuscript requires overall clarity improvements. For instance, the authors frequently interchange terms such as potassium superoxide, lithium superoxide, and superoxide ion without clear distinction. It is essential to consistently and clearly differentiate these species within both the figures and the text to avoid confusion.
5. The manuscript would benefit from a thorough revision to enhance the quality of English writing and adopt a more academic tone throughout the text. For example, on page 8, line 203, the authors state: "They also enhance kinetics. However, these findings lacked a conclusive explanation, which the insights from Fig. 2 can provide." It is unclear what "they" refers to and what specific insights Figure 2 offers. The authors should provide explicit explanations of such references to improve clarity.
6. On page 9, line 214, the authors claim that dissolved LiO_2 is less stable compared to solid LiO_2 , leading to energy inversion phenomena in pure Li^+ electrolytes. However, it is generally observed that when solid species dissolve and become solvated, they are typically more stabilized. Furthermore, considering the discharge pathway of lithium-air batteries, intermediate dissolved LiO_2 states precede the formation of Li_2O_2 without precipitation of solid LiO_2 . This suggests that dissolved LiO_2 may, in fact, be more stable than its solid counterpart. The authors should reconcile these statements with existing evidence to ensure consistency and accuracy.
7. Although the authors address factors such as pH and solvent properties, more concrete solutions for achieving stable oxygen redox behavior would strengthen the manuscript's impact.

Version 2:

Reviewer comments:

Referee #1

(Remarks to the Author)

Freunberger and coworkers have submitted a revised manuscript, in which they have addressed the criticism from the first round of review. I am pleased to read the authors' response and their efforts to clearly communicate the results and implications of their work to the community. Overall, I conclude that the authors did a good job in rewriting the introduction and revising figure 1. The didactics and differentiation of this work compared to the previous Nat Chemistry article becomes clear and the hypothesis as well as the subsequent analyses are clearly elaborated. I also agree that the additional paragraph "Greater relevance to life sciences and energy" in the Methods section is an enrichment to the manuscript. Based on the above assessment and the authors' revisions, I recommend publication of their work in Nature.

Referee #2

(Remarks to the Author)

I have read the revised version of the paper and the letter with the answers to the referees. The latter significantly strengthens the case for publication. I suggest publication of the paper in its revised form without further revisions on my side. The authors have addressed all the referees' comments in great details and have substantially improved both the clarity and the generality of the discussion. The revisions have also expanded the scope of the work and now its general relevance is more apparent than in the previous version.

Referee #3

(Remarks to the Author)

We appreciate the authors' detailed explanations and the revised manuscript that thoughtfully addresses our previous comments. Our primary concern pertained to the novelty of the main arguments, particularly regarding the kinetics and thermodynamics of superoxide disproportionation into singlet and triplet oxygen species in the presence of peroxide. The authors have now provided sufficient clarification concerning the novelty of their work in relation to their prior publications and other relevant studies. In particular, the strength of this manuscript lies in its rigorous elucidation of phenomena that have been experimentally observed yet previously lacked clear mechanistic interpretation. Moreover, the authors convincingly demonstrate the broad applicability of their findings across diverse disciplines, including the life sciences and energy storage, as articulated in the Conclusion and Methods sections. We find this interdisciplinary scope to be well aligned with Nature's broad readership, extending the relevance of the work beyond the battery field.

To further improve the clarity and perceived novelty of the manuscript prior to final publication in Nature, we offer the following suggestion:

As noted in the Introduction, oxygen redox chemistry plays a critical role in battery materials. In addition to metal–air batteries, layered transition metal oxides have recently gained attention as cathode materials that exploit oxygen redox reactions (see reference 2 in the manuscript). While the discussion of oxygen spin states in this context remains limited, the evolution and reactivity of various oxygen species are highly pertinent. We believe it would greatly benefit the community if the authors could briefly share their perspective on the relevance of their findings to the oxygen-redox cathode field as well.

Response to reviewers on the paper entitled

“Individual Marcus-type kinetics controls singlet and triplet oxygen evolution from superoxide oxygen formation”

Manuscript: 2024-09-18358A-Z

We thank the Reviewers for their helpful comments, to which we respond below point by point. The Reviewers' comments are reproduced in italics. In the manuscript, we highlighted the changes in yellow. The comments helped us greatly in improving the manuscript, and we are confident that we have satisfactorily responded to all of them.

Reviewer #1:

Freunberger and coworkers have submitted a paper to Nature in which they shed light on the formation of highly reactive singlet oxygen, which is an important scientific question for the life science and energy communities. Their hypothesis is that the driving force (dG) governs the formation of singlet oxygen, which is explained by a Marcus-type kinetics. Using the superoxide KO_2 , the authors demonstrate that for low anodic potentials (w. r. t. K/K^+), triplet oxygen is formed. With increasing anodic potential, the rate constant for the formation of triplet oxygen drops down due to a Marcus-inverted region. On the other hand, larger anodic potentials allow for the formation of the thermodynamically less stable singlet oxygen, which has a higher equilibrium potential (w. r. t. K/K^+) compared to triplet oxygen. Therefore, the rate constant for the formation of singlet oxygen rises with increasing anodic potential so that the formation of singlet oxygen is superior to that of triplet oxygen. The implication of this finding is that the driving force (dG) must be controlled to produce either triplet oxygen or singlet oxygen. Without further ado, the results reported by the authors are groundbreaking and fall within the scope of the Nature journal. I generally support the publication of the authors' work, although I ask them to address the following questions and comments before publication can potentially take place:

We sincerely thank the Reviewer for the enthusiastic support of our work. Equally, we are very grateful for taking the time to highlight in detail where we should be clearer in our writing. In response to your comments, we revised the text profoundly as detailed below.

- i) I must admit that it was hard to grasp figure 1 based on the calculated $\ln k$ values at first glance, considering that equation (1) was introduced after the figure was presented. However, the authors did not discuss which experimental technique was used to quantify the rate constants k in their analysis, at least not in the context of this figure. This requires further clarification and discussion in the manuscript. It is also unclear for the reader which technique was used to detect 3O_2 or 1O_2 to distinguish between these chemical species. After reading the manuscript further, I understood that mass spectrometry and NIR emission was used, respectively (lines 147-149). In any case, the didactics can be further improved in the discussion of the results. In connection with Figure 1, several questions arise which are addressed later in the text.*

We sincerely apologise for the unclear presentation and thank you very much for pointing out where the writing was not clear. In response, we rewrote the text quite profoundly and hope that the didactics becomes now clear. Please see below our response and changes in response to your more detailed comments.

The error bars of the $\ln k$ values differ to quite some extent. Although I believe that not all of this information needs to be included in the main text, an explanation of how these error bars are determined and the reason for their different sizes should be part of the supplementary notes.

For one, error bars appear asymmetric on an $\ln(k)$ scale if they are symmetric on a linear scale. For the other, differences in the magnitude of the error bars amongst mediators arise from their large chemical diversity and specific reactivities. For example, Fe^+ , reduced quinones, and TDPA react with O_2 , or high-voltage RM^{ox} show limited long-term stability in the electrolyte. For repetitions, we always made new portions of RM^{ox} . While we made every effort to conduct the experiments as reproducibly as possible, the mentioned reactivities and manual handling resulted in some scattering. Yet, we believe the scattering does not impact our conclusions. We have included this and the description of the errors in the Methods section “Kinetics measurements”.

Please note that, for editorial reasons, supplementary notes are now integrated into the Methods, and all previous Supplementary Figures are now designated as Extended Data Figs.

I am also not sure whether the data shown in figure 1 was measured for the current manuscript or whether it is taken from a previous work of the authors in Nature Chemistry. The data shown in Figure 1 and Supplementary Figure 1 are essentially the same, with the only difference of the red bars quantifying the formation of 1O_2 . Therefore, in the presentation and discussion it remains somewhat hidden which data are new and which come from previous work by the authors. To be eligible for publication in Nature, a significant amount of new data should be available compared to the authors' previous publications in Nature Chemistry.

Further confusion arises because the authors state in lines 69-70: “As we reached the higher end of the investigated driving forces, 1O_2 evolution became significant.” This indicates that the authors quantified 1O_2 evolution by their experimental setup (as shown in Supplementary Figure 1), although they present a hypothesis in lines 90-91. I would have expected to discuss the hypothesis before providing any experimental proof.

In short, the intermingling of hypothesis and experimental quantification as well as old and new data needs to be resolved.

We apologise for any confusion related to the data in Fig. 1 and Supplementary Fig. 1, and how we came up with the hypothesis to test – two rather than one parabola.

The measured data in Fig. 1 were taken from a previous publication, which suggested that there is a single kinetic parabola and $^1\text{O}_2$ forming at the upper end. However, precisely the inability of these data to explain what truly controls $^3\text{O}_2$ vs $^1\text{O}_2$ evolution triggered us to the present study. The value of these data for this

paper, therefore, lies *not in being new*, but – after giving them sufficient thought – in *highlighting a fundamental knowledge gap* and a *common misperception*.

Previous works on $^1\text{O}_2$ evolution from superoxide oxidation *only considered “thresholds”* for $^1\text{O}_2$ evolution. Hence, they only considered the thermodynamic *possibility*, but could not make any statement about the extent to which $^1\text{O}_2$ will form since *nothing was known about relative kinetics*. This is true for the classical work of, e.g., Bard¹ and for works in the context of Li-O₂ cells². Considering a “threshold” for $^1\text{O}_2$ evolution was, after all, also true for our previous paper in Nat. Chem. Therein, we first introduced the aspect of kinetics of electron extraction, but didn’t distinguish between the kinetics of forming the individual products.

The shown data points (now in Extended Data Fig. 1, previously Supplementary Fig. 1) suggested two major conclusions in the previous publication:

- 1) Superoxide oxidation kinetics as a function of driving force follows *a single* Marcus-type kinetics parabola.
- 2) Driving forces above 0.97 eV (redox potentials $>E_{\text{O}_2/\text{KO}_2}^\circ + 0.97\text{V}$) do produce $^1\text{O}_2$. $^1\text{O}_2$ yields were measured using a chemical trap.

The second conclusion ties in with the views of the previously mentioned works, which only considered “thresholds” for $^1\text{O}_2$ evolution. Note that none of these previous works considered either absolute or relative kinetics of the formation of $^3\text{O}_2$ or $^1\text{O}_2$. The frequently used connotation of a “threshold” arises from the literature on electrochemiluminescence, which speaks of “*energy-sufficient*” processes³. Remarkably, reviews on electrochemiluminescence do not even mention the word “kinetics” or “rate” and only refer to “*energy-deficient*” or “*energy-sufficient*” processes. What controls the extent to which $^1\text{O}_2$ or $^3\text{O}_2$ evolve remained unclear.

The single kinetic parabola obtained from measurements at low to moderate driving forces (now in Extended Data Fig. 1) led us now to the hypothesis that this was the parabola for the ground state (and not overall kinetics) and that a second parabola for the excited state formation should occur as described from Fig. 1 onwards.

To avoid confusion, we rewrote the section “Electronically excited species from electron transfer reactions” and revised Fig. 1. We describe how the previous data suggest the above conclusions (and introduce equation (1)), but how they fail to conclusively explain what controls the extent to which $^1\text{O}_2$ or $^3\text{O}_2$ evolve. Then we formulate the hypothesis of individual parabola, introduce and explain equation (2) using the potential energy surfaces and kinetic parabola in Fig. 1. We moved the data points from Fig. 1 into Extended Data Fig. 1 and state that the left parabola results from the experimental data there. Also, in response to other Reviewers’ requests, we simplified Fig. 1 and revised the legend as follows.

¹ E.g., Bard, *JACS* 95, 6223 (1973); Ando, *JACS* 102, 4526 (1980); Bard, *Anal. Chem.* 85, 292 (2013);

² Hassoun, *Angew. Chem. Int. Ed.* 50, 2999 (2011); McCloskey, *J. Phys. Chem. Lett.* 3, 3043 (2012); Wandt, *Angew. Chem. Int. Ed.* 55, 6892 (2016). Based on these works also our previous paper *Nat. Chem.* 13, 465 (2021) considered $E_{\text{O}_2/\text{KO}_2}^\circ + 0.97\text{V}$ as the threshold for $^1\text{O}_2$ formation.

³ Bard & Faulkner, *Electrochemical Methods: Fundamentals and Applications*; Miao, *Chem. Rev.* 108, 2506 (2008); Richter, *Chem. Rev.* 104, 3003 (2004). Knight & Greenway. *Analyst* 119, 879 (1994).

Fig. R1 (revised Fig. 1) | Marcus theory suggests separate kinetics of superoxide oxidation to 3O_2 and 1O_2 . **a**, Hypothesis for how the driving force could govern 3O_2 and 1O_2 formation kinetics from superoxide oxidation. The left parabola results from previously measured rate constants for mediated superoxide oxidation with driving forces up to $-\Delta G^\circ \approx 1.2$ eV as shown in Extended Data Fig. 1. However, a single kinetic parabola could not conclusively explain why 1O_2 formed for $-\Delta G^\circ \geq 1$ eV. Based on the considerations in **b**, individual kinetic parabolas (k_3 and k_1) for the reactions yielding 3O_2 and 1O_2 can be constructed with the full line showing their sum. The maxima are shifted by $\Delta G_{1\leftarrow 3}^\circ \sim 0.97$ eV (see text), and equal prefactors are assumed. The blue and red shaded area shows the transition from $k_3/(k_{1+3}) = 0.99$ to $k_1/(k_{1+3}) = 0.99$. **b**, Potential energy surfaces for mediated superoxide oxidation for different driving forces. Black, blue, and red parabolas denote reactants ($KO_2 + RM^{ox}$), and 3O_2 or 1O_2 in the products (${}^3O_2 + RM^{red}$ or ${}^1O_2 + RM^{red}$), respectively. The cases shown are for barrier-less reactions to 3O_2 (i) and 1O_2 (iii) and equal barriers (ii). The subscripts 3 and 1 denote triplet and singlet states, respectively.

We thank you again for pointing out the unclear writing, and we hope it is now clear. We hope to convince you that the value of the previously measured data for this manuscript lies in their new interpretation (the kinetics of forming 3O_2) and in provoking the hypothesis of individual parabolas for 1O_2 or 3O_2 , where the data fit perfectly.

ii) While the developed theory in lines 96-113 is beautiful, what is the scientific argument for using the same reorganization energy for the formation of $^1\text{O}_2$ as for the formation of $^3\text{O}_2$? ($\lambda \equiv \lambda_3$) This question also arises in connection with lines 131-132, where it becomes clear that this assumption is not entirely fulfilled.

This question stems from a similar issue to the previous one, which arose from our unclear writing in this section. We apologise for this and appreciate you bringing it to our attention. It should now be fixed with the revision. We think the question is best answered in two parts.

First, the assignment $\lambda \equiv \lambda_3$ in the previous text was misleading. With this, we did not want to say that λ_1 and λ_3 have to be equal. We wanted to say that for λ_3 in equation (2) we used the value of λ that was fitted to the initially available data (now in Extended Data Fig. 1). This data could, because of the stability limit of the used glyme electrolyte, only cover a relatively limited range of driving forces where only a *single parabola* became apparent. Therefore, the standard Marcus expression in equation (1) $k = Z_{\text{el}} \exp(-(\Delta G^\circ + \lambda)^2/4\lambda RT)$ with a *single λ value* seemed appropriate. However, as described above, a single parabola could not explain what really controls $^3\text{O}_2$ vs $^1\text{O}_2$ evolution, which led us to the hypothesis that the previously seen parabola (characterised by λ) was the one for $^3\text{O}_2$ (characterised by λ_3) and that there should be a second parabola for $^1\text{O}_2$ (characterised by λ_1). This is now formulated in the revised section “Excited species through electron transfer”.

Second, the values of the reorganisation energies λ_1 versus λ_3 . We needed to assume a value for λ_1 to draw the hypothetical k_1 parabola in Fig. 1; thereafter, the values are obtained from fits to the data. The reorganisation energy arises from solvent reorganisation between reactants ($\text{RM}^{\text{ox}} + \text{O}_2^-$) and products ($\text{RM}^{\text{red}} + ^3\text{O}_2$ or $^1\text{O}_2$) and their change in nuclear coordinates. The most significant reorganisations arise from the reactants to any O_2 product. E.g., the bond length changes by ~12% from O_2^- to $^3\text{O}_2$, while $^3\text{O}_2$ and $^1\text{O}_2$ differ by only 0.6%. λ_1 and λ_3 and should therefore anyway be rather similar.

Following the IUPAC recommendation, we adapted the notation slightly to use the shorthand $\Delta G_{1\leftarrow 3}^\circ = \Delta G^\circ(^1\Delta_g \leftarrow ^3\Sigma_g^-)$ and revised equation (2) to

$$k_{1+3} = k_3 + k_1 = Z_{\text{el},3} \cdot e^{-\frac{-(\Delta G^\circ + \lambda_3)^2}{4RT\lambda_3}} + Z_{\text{el},1} \cdot e^{-\frac{-(\Delta G^\circ + \Delta G_{1\leftarrow 3}^\circ + \lambda_1)^2}{4RT\lambda_1}},$$

where the first and second terms denote the parabola for $^3\text{O}_2$ vs $^1\text{O}_2$ evolution, respectively. There was an initial typo in this equation, and we apologise for it. The first parabola has its peak at λ_3 , the second one at $\Delta G_{1\leftarrow 3}^\circ + \lambda_1$. The k_3 parabola for $^3\text{O}_2$ in Fig. 1 is given by the fitted values of $Z_{\text{el},3}$ and λ_3 to the data in Extended Data Fig. 1.

For the sake of drawing the hypothetical k_1 parabola for $^1\text{O}_2$ evolution in Fig. 1, we needed to assume values for $Z_{\text{el},1}$, λ_1 , and $\Delta G_{1\leftarrow 3}^\circ$. We assumed $Z_{\text{el},1} = Z_{\text{el},3}$, $\lambda_1 = \lambda_3$, and $\Delta G_{1\leftarrow 3}^\circ \approx \Delta H_{1\leftarrow 3}^\circ = 0.97$ eV. Only the measured kinetics over the extended range of driving forces in Fig. 2 allowed us to test those assumptions. Fitting the data allowed us to determine $Z_{\text{el},1} \sim 6.4 \cdot Z_{\text{el},3}$. Within the accuracy of the fit, λ_1 and λ_3 are equal. This is still reasonable since differences between λ_3 and λ_1 should only result from different solvation and O–O distances of $^3\text{O}_2$ and $^1\text{O}_2$. The shift between the maxima is not the typically assumed 0.97 eV, but $\Delta G_{1\leftarrow 3}^\circ = 0.84$ eV, which allowed us to determine the previously unknown entropy change as $T\Delta S_{1\leftarrow 3}^\circ = -0.13$ eV.

iii) *Supplementary Figure 2: Given that the discussion of the manuscript relates to U vs. K/K+, it would be helpful for the reader if the redox potentials of the redox mediators are also shown on the K/K+ scale.*

We changed the scale to the K/K⁺ scale. The figure is now in Extended Data Fig. 2.

iv) *Lines 190-195: this part is hard to grasp. To which calculations do the authors refer in line 193?*

Thank you for pointing this out. For brevity, we had discussed this very (too) briefly in the main text and in more detail only in the supplementary information. We have now shifted this discussion to the section “Driving forces upon superoxide oxidation” (line 490 onwards) and Extended Data Fig. 7, where this is discussed in more detail.

As for the calculations, we refer to the calculations of Kisuk Kang’s group⁴. CO₂ in contact with O₂⁻ forms peroxocarbonates, which can be reduced by O₂⁻, liberating O₂. The group calculated the reaction energy of various redox couples that liberate O₂ from O₂⁻. For example, LiCO₄[•] + O₂⁻ → LiCO₄⁻ + O₂ (-ΔG° ~ 1 eV).

We condensed the section in the main text to

Figure 2a establishes a working curve for an extensive range of ΔG°, allowing us to understand the behaviour of various important systems. Lewis and Brønsted acid-driven O₂⁻ disproportionation are two widely relevant cases, examined in the following sections. Additional examples of O₂⁻ oxidation, where the explanation for whether ¹O₂ forms has been elusive, are examined in **Extended Data Fig. 4 and the Methods**. These examples include O₂⁻ in contact with CO₂ and organic peroxides, with relevance including for energy storage and biological systems^{1,9,38-40}.

v) *In the field of batteries, potassium is not the main ion used for battery applications. To date, most batteries are still based on lithium-ion or sodium-ion technologies. This is touched in the section “Disproportionation in non-aqueous systems”. While the authors already discussed that the crossing point of the two parabolas is not at a constant driving force (Extended Data Fig. 2), I am wondering whether a change in the ion has also an effect on the crossing point of the two parabolas. How do the thermodynamic correlations as discussed in Figure 3a look like for Na- or K-based systems? Further discussions could lead to a generalization of the results to other battery technologies.*

Please note that we have also revised Fig. 3a and the associated text to clarify how solvation affects the relevant Li-oxygen redox couples. We added a new Methods section “Thermodynamics in mixed alkali metal/TBA⁺ electrolytes” (line 541) and the new Extended Data Fig. 5a-c discusses this in more detail. The key to understanding is the solubility of LiO₂.

We expanded to other cations in non-aqueous cases also in this Methods section and Extended Data Fig. 5. The thermodynamic correlations for Li, Na and K-based systems are shown in Extended Data Fig. 5a where the redox potentials of the O₂/MO₂ and O₂/M₂O₂ couples are calculated from their formation energies (Δ_fG°) as far as literature values are known (for Li₂O₂, Na₂O₂, NaO₂, K₂O₂, KO₂, but not for LiO₂). KO₂ is

⁴ Kang, *et al.* Toward a Lithium–“Air” Battery: The Effect of CO₂ on the Chemistry of a Lithium–Oxygen Cell. *J. Am. Chem. Soc.* 135, 9733 (2013). <https://doi.org/10.1021/ja4016765>.

thermodynamically more stable than K_2O_2 , which makes the disproportionation of KO_2 to K_2O_2 thermodynamically unfavourable⁵.

NaO_2 disproportionation to Na_2O_2 is thermodynamically favourable, albeit only slightly based on potentials involving solid $\text{NaO}_{2(s)}$ and $\text{Na}_2\text{O}_{2(s)}$. Analogous to the Li-based system, NaO_2 disproportionation to Na_2O_2 yields $^1\text{O}_2$, and TBA^+ , when present next to Na^+ , enhances the amount of $^1\text{O}_2$. Extended Data Fig. 5d presents the thermodynamics of the relevant redox couples with an emphasis on the effect of solvated NaO_2 species. Extended Data Fig. 5e shows newly added NIR emission data for Na^+ and mixed Na^+/TBA^+ electrolytes, which confirm $^1\text{O}_2$ formation in a pure Na^+ electrolyte and an increase with TBA^+ . The driving force for NaO_2 disproportionation is smaller compared to that of LiO_2 disproportionation. However, since we observe $^1\text{O}_2$ at a lower driving force for Na^+ system, the reorganisation energy appears sufficiently low therein.

The change of the parabola with changing electrolytes is an important topic to generalise the results. We added the following discussion to the new Methods section “*Thermodynamics in mixed alkali metal/TBA⁺ electrolytes*”:

More generally, the electrolyte properties (solvent(s), salts(s), and their concentrations) will impact the reorganisation energy and hence the maxima and crossing point of the two kinetic parabolas. The classical approach to accounting for this is the equation given by Marcus⁵⁴, which connects the reorganisation energy with the effective dielectric properties of the electrolyte and the separation of the redox centres. A lower dielectric constant and smaller separation will result in a larger reorganisation energy. A refined equation by Marcus⁵⁵ further takes into account the ionic environment. Such considerations apply well, for example, to aqueous anionic redox couples and the series of alkali metal cations from Li^+ to Cs^+ as spectator cations, where λ decreases⁵⁶. However, caution is required with nonaqueous, low dielectric constant media, where strong ion-pairing occurs. Some works suggest an inverse trend for λ amongst the alkali metals^{19,57}. Ion pairing and even clustering is particularly severe for (su)peroxide as the redox anions as discussed above. Superoxide forms in nonaqueous Li^+ and Na^+ electrolytes clusters^{32,33} and the peroxides are practically insoluble⁵¹. The order and extent to which the reorganisation energy changes for superoxide oxidation in nonaqueous media among the alkali cations may therefore not be predicted straightforwardly and would merit further investigation. As we observe $^1\text{O}_2$ at low driving force for the Na^+ electrolyte, the reorganisation energy appears sufficiently low therein.

Let us note that cation effects have been extensively studied in aqueous OER electrocatalysis for well over 60 years, and yet they are not entirely understood. While the mentioned trend from Li^+ to Cs^+ is often observed, this is not always the case, and we may quote Eikerling: “It is certainly fair to say that the understanding of cation effects in the OER is still incomplete.”⁶

⁵ In accord with wide literature, e.g., Lu et al. *Acc. Mater. Res.* 2, 515 (2021).

⁶ *JACS Au* 1, 1752-1765 (2021).

vi) The statement in line 246-247 is unclear because the authors compare the driving force (dG) to an equilibrium potential. Do the authors intend to say that in the oxygen evolution reaction (OER), ${}^1\text{O}_2$ rather than ${}^3\text{O}_2$ is formed for $\text{pH} < 10$? Considering that the OER requires significant overpotentials of at least 300 mV to reach current densities in the order of 10 mA/cm², which part of the two parabolas does this overpotential regime represent? I think that the implications for electrocatalysis should be further discussed.

For proton-induced superoxide disproportionation, the oxidant is the $\text{O}_2^-/\text{H}_2\text{O}_2$ redox couple⁷. The driving force should therefore be $-\Delta G^\circ = (E_{\text{superoxide}/\text{H}_2\text{O}_2}^\circ - E_{{}^3\text{O}_2/\text{superoxide}}^\circ)F$.⁸ We also mark now in the figure with the potential scale correctly the difference in redox potentials as $\Delta G^\circ/F$.

Thank you for pointing out the imprecise statement that ${}^1\text{O}_2$ rather than ${}^3\text{O}_2$ could be expected below $\text{pH} < \approx 10$. An important outcome of our work is that one should not think of $E_{{}^1\text{O}_2/\text{O}_2^-}^\circ$ (or the driving force where this is exceeded) as a “threshold” potential above which ${}^1\text{O}_2$ rather than ${}^3\text{O}_2$ forms. Previous works⁹ on ${}^1\text{O}_2$ evolution from superoxide oxidation considered “thresholds” for ${}^1\text{O}_2$ evolution at $E_{{}^1\text{O}_2/\text{O}_2^-}^\circ = E_{{}^3\text{O}_2/\text{O}_2^-}^\circ + 0.97$ V. The frequently used connotation of a “threshold” arises from the electrochemiluminescence literature, which speaks of “energy-sufficient” processes¹⁰. For example, the process $\text{R}^{\bullet-} + \text{M}^{*+} \rightarrow {}^1\text{R}^* + \text{M}$ is considered “energy-sufficient” to form the excited ${}^1\text{R}^*$, if $(E_{\text{M}/\text{M}^{*+}}^\circ - E_{\text{R}/\text{R}^{\bullet-}}^\circ)F \geq -\Delta H^\circ({}^1\text{R}^* \leftarrow \text{R})$. This condition gives, however, no indication of the extent to which ${}^1\text{R}^*$ rather than R forms. More precisely, one should consider relative kinetics to form ground and excited species as a function of driving force. As Extended Data Fig. 4 illustrates, an onset of ${}^1\text{O}_2$ may be expected at $\Delta G_{1\leftarrow 3}^\circ$, but its formation will only become significant for driving forces $-\Delta G^\circ > \lambda_3$ where ${}^3\text{O}_2$ formation slows down to benefit ${}^1\text{O}_2$ formation.

In response to your comment, we made several changes throughout the text.

- 1) We added a supplementary note (the section “Relation between $E_{{}^1\text{O}_2/\text{O}_2^-}^\circ$ and the ${}^1\text{O}_2$ fraction” in the Methods, line 527), which explains the above considerations.
- 2) Together with the other conclusions from the double parabola in Fig. 2, we now explicitly state that $E_{{}^1\text{O}_2/\text{O}_2^-}^\circ$ should not be considered a threshold as frequently done (line 160).
- 3) We now mark $\Delta G_{1\leftarrow 3}^\circ$, which is the driving force at $E_{{}^1\text{O}_2/\text{O}_2^-}^\circ$, in Fig. 2 and Extended Data Fig. 4. This illustrates that reaching $E_{{}^1\text{O}_2/\text{O}_2^-}^\circ$ (or $\Delta G_{1\leftarrow 3}^\circ$) *per se* does not indicate the extent to which ${}^1\text{O}_2$ forms, but requires considering the relative kinetics as given by the parabolas.

⁷ Depending on pH, the protonated or deprotonated species are to be considered.

⁸ Please note that in response to a Reviewer request, we updated the Pourbaix diagram in Fig. 4a with the $E_{\text{O}_2^-/\text{H}_2\text{O}_2}^\circ$ and $E_{{}^3\text{O}_2/\text{O}_2^-}^\circ$ recommended by the IUPAC (Refs. below) rather than the previously used values from an older reference. The slight changes in driving force do not change the conclusions.

(a) Koppenol *et al.* Standard electrode potentials involving radicals in aqueous solution: inorganic radicals. *Biolnorg. React. Mech.* 9, 59 (2013). <https://doi.org/10.1515/irm-2013-0005>.

(b) Koppenol, *et al.* Electrode potentials of partially reduced oxygen species, from dioxygen to water. *Free Radic. Biol. Med.* 49, 317-322 (2010). <https://doi.org/https://doi.org/10.1016/j.freeradbiomed.2010.04.011>.

⁹ *Angew. Chem. Int. Ed.* 50, 2999 (2011); *J. Phys. Chem. Lett.* 3, 3043 (2012); *Angew. Chem. Int. Ed.* 55, 6892 (2016), based on this also our paper *Nat. Chem.* 13, 465 (2021).

¹⁰ Bard & Faulkner, *Electrochemical Methods: Fundamentals and Applications*; Miao, *Chem. Rev.* 108, 2506 (2008).

- 4) In the sections on “Disproportionation in non-aqueous systems” and “Proton-induced disproportionation”, we changed the formulations to make clear that driving forces $\gtrsim \Delta G_{1\leftarrow 3}^\circ$ establish a possible onset, but not a switch to $^1\text{O}_2$.

Considering **implications for electrocatalysis** and the overpotentials for OER. The mentioned >300 mV overpotential for significant electrocatalytic OER activity, e.g., via the adsorbate mechanism or lattice oxygen mechanism, are most often quoted to result from the O-O bond formation step¹¹. However, the reaction we are considering here is “only” the oxygen evolving step from superoxide species by chemical oxidants, such as another superoxide during DISP or the mediators we used in Fig. 2. Interestingly, the “active oxygen” species that precedes the O_2 evolution on, for example, the extensively studied Ni(Fe)OOH or CoOOH catalysts that are most active in basic media have more recently been identified as superoxo species. In terms of the mechanism by which O_2 evolves from these superoxo species, metal redox is the most considered mechanism (Fig. R2a-d). The $\text{M}^{n-1/n}$ redox couple has been suggested to oxidise the superoxo moiety to O_2 . Some works consider this to be the rate-limiting step (Fig. R2d)¹². Of note, using a model system, Nocera et al. suggested preferential singlet character of bridge Co-O-O-Co moieties (Fig. R2e), which may further get preferentially oxidised to $^1\text{O}_2$ ¹³. Figure R2f shows some of the most active metal redox couples for acidic and basic environments relative to the Pourbaix diagram shown in Fig. 4. Interestingly, the $\text{M}^{n-1/n}$ redox couples are typically a few 100 mV above the $^1\text{O}_2$ /superoxide potential and provide, in principle, enough driving force for $^1\text{O}_2$ evolution. For example, at pH 14 $E_{^1\text{O}_2/\text{O}_2^-}^\circ = 1.32$ V, $E_{\text{Co}^{\text{II/III}}}^\circ \approx 1.25$ V, $E_{\text{Co}^{\text{III/IV}}}^\circ \approx 1.5$ V, $E_{\text{Ni}^{\text{II/III}}}^\circ \approx 1.52$ V, $E_{\text{Ni}^{\text{III/IV}}}^\circ \approx 1.6$ V vs RHE^{11,12}. The $^1\text{O}_2$ /superoxide potential is at a driving force $\Delta G_{1\leftarrow 3}^\circ$ and, depending on the reorganisation energy, the system may already be operating in the regime between the two maxima of the kinetic parabolas. We are well aware that potentials involving free superoxide may not be the same as for the superoxo M-OO species. Additionally, pH and spectator cations in the electrolyte will impact the potential and reorganisation energy¹⁴, which ties in with the discussion in response to comment v). Overall, the above considerations suggest that $^1\text{O}_2$ evolution may be significant in the OER in the typical overpotential regime. If this is the case, this begs the question whether $^1\text{O}_2$ and $^3\text{O}_2$ evolution are similarly governed by individual Marcus kinetics. Further investigations specifically on OER catalysis are therefore warranted.

¹¹ The O-O forming step requires to highest overpotential for all reaction steps to become exothermic. The O_2 release is considered to be already before downhill. E.g., for metal oxides: Nørskov, *Chem. Phys.* 319, 178 (2005); Goddard, *PNAS* 115, 5872 (2018); For lattice oxygen cats: Shao-Horn et al. *Nat. Chem.* 9, 457 (2017).

¹² Hu, *JACS* 142, 11901 (2020).

¹³ Chen, *J. Phys. Energy* 3, 031004 (2021).

¹⁴ e.g., Koper, *Angew. Chem. Int. Ed.* 58, 12999 (2019); Grimaud, *Angew. Chem. Int. Ed.* 56, 8652 (2017).

REDACTED

Fig. R2a: Fig. 3B in Goddard, et al. Proc. Natl. Acad. Sci. USA 115, 5872-5877 (2018).

Fig. R2b: Fig. 9A in Hu, et al. Chem Catal. 3, 100475 (2023).

Fig. R2c: Fig. 7 in Messinger, et al. Energy Environ. Sci. 8, 2492-2503 (2015).

Fig. R2d: Fig. 8 in Hu et al., JACS 142, 11901 (2020).

Fig. R2e: Fig. 6 in Nocera, et al. Proc. Natl. Acad. Sci. USA 114, 3855-3860 (2017).

Fig. R2 |

vii) *A general question is, how can the driving force in life sciences or energy applications be controlled to render the formation of either $3O_2$ or $1O_2$? Can the authors provide recommendations or guidelines?*

We expanded the Conclusions section to provide more concrete implications for the life sciences and energy applications. Given that this had to be very concise, we added more detailed discussion in a new Methods section, “*Wider relevance for life sciences and energy*” (line 604), where we summarise the discussion just below and the discussion above on the OER reaction.

Our findings on proton-induced disproportionation are likely of primary interest to the life sciences. It is certainly out of the scope of the present paper to propose “solutions” to any adverse effect 1O_2 may have in cells. However, the paper contributes to understanding how the pH impacts the link between the four important reactive oxygen species (ROS) superoxide, peroxide, 3O_2 , and 1O_2 . These species have very

¹⁵ Nocera, Proc. Natl. Acad. Sci. USA 114, 3855 (2017).

¹⁶ Data from: Fabbri, Nat. Chem. 17, 856 (2025); Hu, Chem Catal. 3, 100475 (2023); JACS 142, 11901 (2020); Jones, Nature 587, 408 (2020).

diverse functions from metabolism to signalling and cell damage. Disproportionation is notably the pathway to maintain a low superoxide concentration; however, detoxification from superoxide produces the harmful $^1\text{O}_2$.

Superoxide occurs in healthy cells in several organelles with different pH levels between 4.7 and 8¹⁷. The superoxide degrading enzyme superoxide dismutase (SOD) is only found in neutral to basic organelles. Growing $^1\text{O}_2$ from proton-induced disproportionation as the pH decreases coincides with higher and lower pH in respiratory (mitochondria) and degenerative organelles (lysosomes), respectively, where $^1\text{O}_2$ must be avoided or may even be beneficial. This connection between pH and $^1\text{O}_2$ formation may well have been a so far unrecognised evolutionary driver for the pH found in organelles. It gives insight into how life works and how it deals with the adverse effects of its inherent chemistry. In pathological situations, the pH balance is known to be impacted (typically towards lower pH) and therewith signalling, redox regulation, and defence¹⁸. However, proposing recommendations is undoubtedly outside the scope of this paper. We may, however, quote Sies, one of the pioneers in oxidative stress in cells: “Better understanding of the molecular and cellular basis of redox biology will guide novel redox medicine approaches aimed at preventing and treating diseases associated with disturbed redox regulation.”¹⁹

Concerning energy applications, several recommendations emerge:

- A) We point to the discussion for the previous comment on OER electrocatalysis.
- B) We recall the discussion to comment v). For suppressing $^1\text{O}_2$, the reorganisation energy for the superoxide oxidation reaction should generally be increased. For non-aqueous, low dielectric constant media, where strong ion-pairing occurs, particularly with superoxide, the change in reorganisation energy among the alkali cations may not be predicted straightforwardly and would merit further investigation.
- C) Our results explain why superoxide disproportionation in non-aqueous Li^+ and Na^+ always yields some $^1\text{O}_2$. Therefore, metal- O_2 cells with peroxide discharge products must not undergo superoxide disproportionation during discharge and charge, as is currently the case. Peroxide should form and decompose via mediators capable of transferring two electrons and that bind to the O_2 moiety without releasing the superoxo state.

Let us note that the findings have implications for the electrogeneration of excited reactive species more generally, with significance ranging from cell biology to energy storage. These include: A) Reactive excited species in life, which are very broadly associated with pathogenic events²⁰. B) Recombination reactions in redox flow batteries are recognised to cause self-discharge, but they have so far not been recognised to potentially form extremely energetic excited species. C) Soluble parasitic oxidised and reduced species at the cathode and anode of Li- and Na-ion batteries may recombine to form energetic excited species.

¹⁷ Casey, et al. Sensors and regulators of intracellular pH. *Nat. Rev. Mol. Cell. Biol.* 11, 50 (2010).

¹⁸ Just a tiny glimpse on the vast literature: Gatenby. Why do cancers have high aerobic glycolysis? *Nat. Rev. Cancer* 4, 891 (2004); Sies, H. & Jones, D. P. Reactive oxygen species (ROS) as pleiotropic physiological signalling agents. *Nat. Rev. Mol. Cell. Biol.* 21, 363 (2020); Swietach et al. How protons pave the way to aggressive cancers. *Nat. Rev. Cancer* 23, 825 (2023).

¹⁹ Sies, Fundamentals of redox regulation in biology. *Nat. Rev. Mol. Cell. Biol.* 25, 701 (2024).

²⁰ Sies et al. *Nat. Rev. Mol. Cell. Biol.* 25, 701 (2024). Goncalves et al. Chemiexcitation: Mammalian Photochemistry in the Dark. *Photochem. Photobiol.* 99, 251-276 (2023).

Reviewer #2:

In this paper Freunberger et al. explore the reactivity of superoxides in forming oxygen molecules both in their ground state and in their first excited state (singlet oxygen). While the thermodynamic of the superoxide \rightarrow oxygen reactions is easily understood, their kinetic proved to be much harder to explore. Its study is however crucial to understand the behaviour of superoxide in electrochemical systems. The knowledge of the conditions for the formation of highly reactive oxygen species in battery chemistries can hardly be underestimated. In this respect this paper is a timely and compelling report of new and well devised experiments that systematically evaluate the behaviour of superoxides in a variety of chemical conditions. The authors show, with what appears to me a coherent set of unambiguous data, how singlet oxygen emerges from superoxides as a consequence of the strengthening of the oxidation driving force under electrochemical conditions. This paper does this in a very systematic way, thus (finally, I would add) providing a clear picture of the rates of the transformations of superoxides into oxygen (triplet and singlet). My opinion is that, given the relevance of the subject and the importance of such a systematic study, the paper is suitable for Nature. There are few minor points that the authors may want to consider:

Thank you very much for your very favourable overall appraisal of our work, as well as your detailed comments below, which have greatly helped us clarify the work for the reader.

- 1. The introduction does a good job in introducing the subject of reactive oxygen production, but there's no explanation as to why it represents a problem in electrochemical devices. I think the authors should provide a brief summary of what they recall in refs. 6 and 7.*

We now explicitly state that in electrochemical devices, $^1\text{O}_2$ causes decomposition of the organic electrolyte and carbon, and changed this sentence to “While $^3\text{O}_2$ is relatively unreactive, $^1\text{O}_2$ is highly reactive with most organic matter^{15,16}. This is readily evidenced in metal-ion batteries^{6,7} and metal- O_2 batteries^{3,4,8}, where $^1\text{O}_2$ is a primary source of degradation, causing decomposition of organic electrolytes and conductive carbon additives, which ultimately degrades the overall device function. ” We had to be brief in this discussion, given the demand to shorten the main text somewhat from the previous version. Still, we hope that this explanation is sufficient for the general reader who is less familiar with batteries.

- 2. Figure 1 is very dense and the caption does not explain it properly. For example, the color code for the potential energy surfaces in (c) is not reported. The blue and red vertical shaded areas are also without explanation. Finally, the origin of the data points is not reported.*

Thank you for pointing this out. We revised this Figure and the caption according to your requests. We added labels to explain the color code of the parabolas, which are black for reactants ($\text{KO}_2 + \text{RM}^{\text{ox}}$), blue for products with $^3\text{O}_2$ ($\text{K}^+ + ^3\text{O}_2 + \text{RM}^{\text{red}}$), and red for products with $^1\text{O}_2$ ($\text{K}^+ + ^1\text{O}_2 + \text{RM}^{\text{red}}$). As a shorthand, we use R and P for reactants and products following the common use in Marcus' papers. The shaded areas had no particular meaning and were removed.

Fig. R2.1 (revised Fig. 1) | Marcus theory suggests separate kinetics of superoxide oxidation to ${}^3\text{O}_2$ and ${}^1\text{O}_2$. **a**, Hypothesis for how the driving force could govern ${}^3\text{O}_2$ and ${}^1\text{O}_2$ formation kinetics from superoxide oxidation. The left parabola results from previously measured rate constants for mediated superoxide oxidation with driving forces up to $-\Delta G^\circ \approx 1.2$ eV as shown in Extended Data Fig. 1. However, a single kinetic parabola could not conclusively explain why ${}^1\text{O}_2$ formed for $-\Delta G^\circ \gtrsim 1$ eV. Based on the considerations in **b**, individual kinetic parabolas (k_3 and k_1) for the reactions yielding ${}^3\text{O}_2$ and ${}^1\text{O}_2$ can be constructed with the full line showing their sum. The maxima are shifted by $\Delta G_{1 \leftarrow 3}^\circ \sim 0.97$ eV (see text), and equal prefactors are assumed. The blue and red shaded area shows the transition from $k_3/(k_{1+3}) = 0.99$ to $k_1/(k_{1+3}) = 0.99$. **b**, Potential energy surfaces for mediated superoxide oxidation for different driving forces. Black, blue, and red parabolas denote reactants ($\text{KO}_2 + \text{RM}^{\text{ox}}$), and ${}^3\text{O}_2$ or ${}^1\text{O}_2$ in the products (${}^3\text{O}_2 + \text{RM}^{\text{red}}$ or ${}^1\text{O}_2 + \text{RM}^{\text{red}}$), respectively. The cases shown are for barrier-less reactions to ${}^3\text{O}_2$ (i) and ${}^1\text{O}_2$ (iii) and equal barriers (ii). The subscripts 3 and 1 denote triplet and singlet states, respectively.

Regarding the origin of the data points, these stem from our previous publication (previous Ref. 25, *Nat. Chem.* 13, 465 (2021)). The paper has been cited in the main text and the figure caption, but this may not have been clear enough. Also, in response to another Reviewer's comments, we revised the text quite profoundly in the section "*Excited species through electron transfer*".

To avoid any confusion about the origin of the data previously shown in Fig. 1, we have shifted it to Extended Data Fig. 1, where we present the kinetics of KO_2 oxidation in tetraglyme electrolyte and the ${}^1\text{O}_2$ yields (measured using the chemical trap dimethylantracene). These data points suggested in the previous paper two main conclusions: 1) Superoxide oxidation kinetics as a function of driving force follow *one single* Marcus-type kinetics parabola, and many commonly used mediators act already in the inverted

region. 2) Driving forces above 0.97 eV (redox potentials $>E_{3O_2/KO_2}^\circ + 0.97V$) produce 1O_2 . The second conclusion tied into the views of previous works²¹ on 1O_2 evolution from alkali peroxide oxidation, which considered “thresholds” for 1O_2 evolution at $E_{1O_2/superoxide}^\circ = E_{3O_2/superoxide}^\circ + 0.97 V$. The frequently used connotation of a “threshold” arises from the literature on electrochemiluminescence, which speaks of “energy-sufficient” processes²². However, this explanation proved inconclusive in the end and could not explain the extent of 1O_2 formation. Following Marcus’ classical work on the electrogeneration of electronically excited species²³, we therefore hypothesised that a second parabola should appear at sufficiently high driving forces. This hypothesis is illustrated in Fig. 1, where the left kinetic parabola represents the fit to the data presented in Extended Data Fig. 1.

3. In Figure 2 the y-axes of panels (b) and (c) are two different things and this should be indicated by a change in the axis text labels.

We changed the y-axis label as suggested. Panel (b) now only deals with 3O_2 yields, and panel (c) with NIR emission intensities, which result from 1O_2 formation. The revised Fig. 2 is shown below.

We further modified panels (b) and (c) in response to another Reviewer’s comment. We have previously shown relative kinetics of 3O_2 and 1O_2 formation ($k_3/(k_{1+3})$ and $k_1/(k_{1+3})$) to indicate where formation of the one or the other dominates. However, measured 3O_2 yields and NIR emission intensities results from the interplay of formation *and* quenching rates. We devised a model to simulate $^3O_2/RM^{ox}$ yields and NIR emission intensities as a function of mediator redox potential. We considered the following processes:

- 1) 3O_2 and 1O_2 formation rates (k_3 and k_1) as given by the two kinetic parabolas in Fig. 2a.
- 2) Physical and reactive 1O_2 quenching by the solvent.
- 3) Physical and reactive 1O_2 quenching by the reduced mediators RM^{red} . Note that the same processes with RM^{ox} are typically negligible in comparison because of the electron demand of these processes²⁴.
- 4) 3O_2 losses ($^3O_2/RM^{ox} < 1$) result from reactive quenching of 1O_2 with solvent or mediator.

Details about the model are given in the Methods Section “From oxidation rates to 3O_2 yields and NIR emission intensities” (line 395) and Extended Data Fig. 6. We also modified the associated main text from line 118.

The measured values follow the simulated ones, which provides evidence for assigning the two kinetic parabolas in Fig. 2a to 3O_2 and 1O_2 formation. Please note that any deviations between simulated and measured values result from simplifications in the model, which we intentionally restricted to capture the most important processes.

²¹ *Angew. Chem. Int. Ed.* 50, 2999 (2011); *J. Phys. Chem. Lett.* 3, 3043 (2012); *Angew. Chem. Int. Ed.* 55, 6892-6895 (2016), based on this also our paper *Nat. Chem.* 13, 465 (2021).

²² Bard & Faulkner, *Electrochemical Methods: Fundamentals and Applications*; Miao, *Chem. Rev.* 108, 2506 (2008).

²³ Marcus, R. A. On the Theory of Chemiluminescent Electron-Transfer Reactions. *J. Chem. Phys.* 43, 2654 (1965).

²⁴ Kwak et al. *Nat. Commun.* 10, 1380 (2019).

Fig. R2.2 (revised Fig. 2) | Free energy dependence of superoxide oxidation kinetics to 3O_2 and 1O_2 and their yields.

a, We measured the kinetic constants k for mediated KO_2 oxidation in MeCN electrolyte with mediators covering an extensive range of redox potentials. Plot of $\ln(k)$ versus mediator potential ($E_{RM^{ox/red}}^\circ$, top axis) and driving force ($-\Delta G^\circ$, bottom axis). $-\Delta G^\circ = (E_{RM^{ox/red}}^\circ - E_{O_2/KO_2}^\circ)F$, where $E_{O_2/KO_2}^\circ = 2.48$ V vs K/K⁺. The mediators are shown in Extended Data Fig. 2. The full line best fits equation (2); the broken line parabolas represent the first and second terms in equation (2). Fitted values are $Z_{el,3} = 1.10 \cdot 10^{-2}$ cm·s⁻¹, $Z_{el,1} = 7.00 \cdot 10^{-2}$ cm·s⁻¹, $\lambda_3 = \lambda_1 = 0.95$ eV, $\Delta G_{1\leftarrow 3}^\circ = 0.84$ eV, $R^2 = 0.998$. Based on these fits, the standard potential E_{1O_2/KO_2}° and associated driving force $\Delta G_{1\leftarrow 3}^\circ$ are marked. They are linked via $E_{1O_2/KO_2}^\circ = E_{O_2/KO_2}^\circ + \Delta G_{1\leftarrow 3}^\circ/F$. The blue and red shaded area indicates the transition from $k_3/(k_{1+3}) = 0.99$ to $k_1/(k_{1+3}) = 0.99$. **b**, 3O_2 yield per mole of RM^{ox} (bars) during KO_2 oxidation as measured by

mass spectrometry. The dashed line and the circular markers show simulated $^3\text{O}_2$ yields considering $^1\text{O}_2$ quenching by solvent and redox mediator (see Methods and Extended Data Fig. 3). The dashed line used the trend line for the mediator quenching rate constant k_Q while markers use the individually measured values (Extended Data Fig. 3c). **c**, Normalized 1270 nm NIR emission (bars) during KO_2 oxidation. The dashed line and the circular markers show the simulated NIR emission considering $^1\text{O}_2$ formation with the kinetics k_1 (the right parabola in **a**) and $^1\text{O}_2$ quenching by solvent and redox mediator (see Methods and Extended Data Fig. 3). Data are presented as mean \pm SD ($n \geq 3$).

4. Near line 185: "They only cover smaller" is not clear who is "they".

Thank you for pointing this out. "They" referred to the examples, for which an explanation for whether or not $^1\text{O}_2$ forms has been elusive. We now write explicitly "these examples". The concerned section has been updated accordingly.

Please note that for editorial reasons, the text has been shortened somewhat and supplementary notes are now integrated into the Methods and all previous supplementary figures are now Extended Data Figs. Therefore, these additional examples are now fully discussed in the Methods Section "Driving forces upon superoxide oxidation" (line 490) and Extended Data Fig. 4.

5. Please check that the text label of y-axis in Figure 3(b) is correct with respect to the caption.

Thank you for noting this. We corrected both the caption and the y-axis label.

6. The conclusions section should summarise also the results for the disproportionation.

We added a sentence in the conclusions, summarizing the results for disproportionation: "For disproportionation, the results explain increasing $^1\text{O}_2$ formation with stronger Brønsted and weaker Lewis acidity, respectively, via their impact on driving forces." Please note that we kept this very short to follow editorial requests that the discussion at the end should neither simply summarize the findings nor repeat statements from the abstract.

Reviewer #3:

In this manuscript, the authors investigated the evolution of singlet oxygen ($^1\text{O}_2$) and triplet oxygen ($^3\text{O}_2$) through superoxide disproportionation. The authors present their findings on reaction kinetics, which are governed by driving force-controlled behavior in accordance with Marcus theory. The study examines the $^1\text{O}_2$ and $^3\text{O}_2$ evolution kinetics through the different formation energies of the oxygen species and proposes that their dual-parabola model provides clear explanations for discrepancies in oxygen redox systems and thus offers insights for controlling reactive oxygen species in both energy storage and biological contexts.

Thank you for highlighting the importance of the topic of triplet and singlet oxygen evolution from superoxide in the broadly important field of oxygen redox chemistry. Thank you also for your detailed comments below, which greatly helped us improve the clarity of our data presentation and discussion. We took utmost care to respond to all of them with new data, analysis, and discussion and revised the manuscript comprehensively.

*While the investigation of singlet and triplet oxygen dimer states is undoubtedly very important for understanding degradation behaviors in oxygen redox chemistry, the manuscript's primary arguments, regarding disproportionation kinetics and thermodynamics of superoxide into singlet/triplet oxygen molecules with peroxide species, do not present sufficient novelty. The relationship between singlet oxygen formation and superoxide disproportionation has been previously documented (*Energy & Environmental Science*, 2019, 12, 2559), as has the competition between singlet and triplet oxygen formation and its Marcus theory explanation (*ACS Energy Letters*, 2020, 5, 1893). Furthermore, the analysis in Figure 3 regarding bulk cation effects on driving force relies heavily on speculation rather than rigorous experimental evidence, largely reiterating previous work without substantial new insights (*Energy & Environmental Science*, 2019, 12, 2559).*

Our paper is not a battery paper, but goes way beyond in terms of scope and implications. We most respectfully request the opportunity to explain how the current paper advances a fundamental open question in the realm of superoxide/ $^1\text{O}_2$ chemistry and how it contributes to the knowledge on the electrogeneration of excited species more generally. Hence, it goes well beyond the two mentioned battery papers.

Before alluding to why the current paper goes beyond the two mentioned battery papers (*EES*, 2019, 12, 2559; *ACS Energy Lett.*, 2020, 5, 1893), please let us briefly summarise the breadth of interest for the broad readership of *Nature*. We intentionally demonstrate relevance across various fields, rather than the narrow confines of batteries alone. We discuss these fields in more detail in response to your following comment.

- 1) Reactions of **superoxo species in general**. Explaining unresolved literature findings on reactions giving $^1\text{O}_2$ or not. For example, reactions with organic superoxo species or superoxo carbonates.
- 2) Superoxide **disproportionation in non-aqueous systems**. Relevant for batteries but likely also for hydrophobic environments in Life.
- 3) **Proton-induced superoxide disproportionation**. There has been a >50-year-old unresolved fierce debate on whether $^1\text{O}_2$ forms this way, but the all-important effect of pH has never been considered.
- 4) **Electrogeneration of excited species in general**. We show the first-ever example of individually resolved Marcus parabolas for ground and excited states.

The novelty beyond the mentioned papers:

In short, in the *EES* paper, we *observed* that Li⁺-driven superoxide disproportionation yields ¹O₂, but we were *unable to explain* this phenomenon. In the *ACS Energy Lett.* paper, to allow for any ¹O₂ formation, the assumed kinetic parabolas require extremely high driving forces (~2 to 3 eV!), which appear to contradict experimental backing (explained below). Neither paper could *explain* ¹O₂ during Li⁺-driven superoxide disproportionation.

Please note that the analysis in Fig. 3 is *not speculation*, but rather based on *quantitative arguments from experiments* to determine the kinetic parabola and the driving forces that occur. We explain this in detail in response to your comment 6. These quantitative arguments are supported by new and published experimental data (CVs, RRDE and others), backed by a wide theoretical literature. We revised Fig. 3, the main text, and added a supplementary section “Thermodynamics in mixed alkali metal/TBA⁺ electrolytes”.

Please allow us to consider the distinction in more detail in relation to the mentioned papers.

- 1) *Energy & Environmental Science*, 2019, 12, 2559: Yes, in this paper, we *observed* that Li⁺ driven superoxide disproportionation yields ¹O₂ and that the presence of weak Lewis acids increases the ¹O₂ yields. However, *observation is not understanding*. In an attempt to explain the observed behaviour, we performed DFT calculations of free energy profiles. However, the reason why a pure Li⁺ environment, where no free superoxide occurs, would result in ¹O₂ *remained unexplained* as prohibitively high barriers >1 eV were obtained. This highlights the known challenge of using DFT for excited states²⁵. Later, higher-level calculations could equally not explain why ¹O₂ could form in a pure Li⁺ environment²⁶.

You ask what the new insights are regarding superoxide disproportionation in Li⁺ electrolytes. We present this as one example among several to demonstrate that experimentally measured individual Marcus kinetic parabolas for ³O₂ and ¹O₂ evolution can explain relevant cases of O₂ evolution from superoxide, which were previously unclear. First and foremost, we can now base the explanation on *experimentally measured* double-parabolas with a free energy difference $\Delta G^\circ({}^1\Delta_g \leftarrow {}^3\Sigma_g^-) = 0.84$ eV and ~6-fold higher prefactor for ¹O₂ than for ³O₂. Both are very different from standard assumptions. Necessarily, e.g., the driving force for the onset of ¹O₂ must differ from standard assumptions (see Fig. R3.1a). We then consider the experimentally determined driving forces for the disproportionation of the relevant solvated LiO₂ clusters to solid Li₂O_{2(s)}. By doing so, we could now explain why ¹O₂ forms in a pure Li⁺ electrolyte.

- 2) *ACS Energy Lett.*, 2020, 5, 1893: This paper uses Marcus theory to consider relative rates of formation of the ground and excited states. However, the driving forces assumed for the disproportionation to ¹O₂ are extremely high, based on many assumptions, and appear to contradict

²⁵ González, L. & Lindh, R. *Quantum Chemistry and Dynamics of Excited States: Methods and Applications*. (Wiley, 2020).

²⁶ a) Bodo, et al, *Phys. Chem. Chem. Phys.* 23, 24487-24496 (2021). b) Bodo, et al. *ChemPhysChem* 21, 2060-2067 (2020).

They state „... we have found that the Li⁺ catalysis promotes only the release of ³O₂ whereas the ¹O₂ formation is energetically unfeasible at room temperature.“

We now can explain why the calculations couldn't explain ¹O₂ formation. The reason is the common assumption that dissolved Li₂O_{2(sol)} or LiO_{2(sol)} form. These are destabilized against Li₂O_{2(s)} by ~2-6 eV (Fig. R3.1b), hence driving forces are very low. The relevant reaction is the disproportionation of solvated LiO₂ to solid Li₂O₂.

reliable experimental and theoretical backing. Contrary to experiments, no $^1\text{O}_2$ is predicted for Li^+ electrolyte in the reaction $2 \text{LiO}_{2(\text{sol})} \rightarrow \text{Li}_2\text{O}_{2(\text{sol})} + \text{O}_{2(\text{sol})}$ ($^1\text{O}_2$ fraction of $\sim 10^{-13}$, Table 2). $^1\text{O}_2$ is only predicted for the reaction with fully dissociated (su)peroxide $2 \text{O}_2^-(\text{sol}) \rightarrow \text{O}_2^{2-}(\text{sol}) + \text{O}_{2(\text{sol})}$, which is assumed to be strongly exergonic ($\Delta G^\circ \leq -1.94$ eV, Table 1). However, this reaction is strongly endergonic and cannot occur, not even in minor shares. This is because in poorly solvating Li^+ electrolytes (such as the assumed glyme), LiO_2 and, even more so, Li_2O_2 do not dissociate in accordance with their low to extremely low solubility²⁷. Figure R3.1b shows calculated solvation energies by Kang's group for the mentioned species that support this argument. $\text{LiO}_{2(\text{sol})} \rightarrow \text{Li}_{(\text{sol})}^+ + \text{O}_{2(\text{sol})}^-$ dissociation is small. The assumed full dissociation $\text{Li}_2\text{O}_{2(\text{sol})} \rightarrow 2\text{Li}_{(\text{sol})}^+ + \text{O}_{2(\text{sol})}^{2-}$ is entirely negligible, given that already partial dissociation $\text{Li}_2\text{O}_{2(\text{sol})} \rightarrow \text{Li}_{(\text{sol})}^+ + \text{LiO}_{2(\text{sol})}^-$ is extremely unfavourable (≈ 6 eV above $\text{Li}_2\text{O}_{2(\text{sol})}$!) in accordance with the extremely low solubility of $\text{Li}_2\text{O}_{2(\text{s})}$. Free peroxide $\text{O}_2^{2-}(\text{sol})$, as assumed to be involved in this paper, hasn't even been calculated in Fig. R3.1b as it is exceedingly unstable, making the claimed reaction to form $^1\text{O}_2$ impossible. Overall, this paper cannot explain observed patterns of $^1\text{O}_2$ and $^3\text{O}_2$ evolution from Li^+ -driven superoxide disproportionation and, hence, cannot pre-empt our work.

REDACTED

Fig. R3.1b: Fig. 4d in Kang et al, Nat. Commun. 7, 10670 (2016).

Fig. R3.1 | Measured vs. commonly assumed kinetic parabola and the stability of dissociated Li and Na (su)peroxides. a, Kinetic parabola (top) and relative kinetics $k_1/(k_1+k_3)$ (bottom) for 3 combinations of ΔG° ($^1\Delta_g \leftarrow ^3\Sigma_g^-$), Z_1 and Z_3 . Blue: Standard assumption of 0.97 eV and $Z_1 = Z_3$, orange: 0.84 eV and $Z_1 = Z_3$. Red: measured values of 0.84 eV and $Z_1 = 6.3 Z_3$.

Overall, we present experimentally measured kinetic parabolas, which cover the normal and inverted region for ground and excited Z states, and show critical different features to previous assumptions (e.g.,

²⁷ Multiple papers support this as more detailed in response to comment 6. Data in Fig. 3.1b are only one example.

$\Delta G^\circ({}^1\Delta_g \leftarrow {}^3\Sigma_g^-) < 0.97 \text{ eV}$, $Z_1 \gg Z_3$). As such, this is already essential new knowledge for electrochemiluminescent reactions more generally. We apply this knowledge to explain a variety of longstanding, poorly understood, and important cases of superoxide oxidation, ranging from batteries and organic superoxides to peroxocarbonates and H^+ -induced superoxide disproportionation.

I am also afraid that the manuscript does not adequately address a general topic suitable for a broad readership of Nature. While the authors suggest potential implications for life sciences, these connections need stronger development and the discussion of practical approaches to mitigate reactive singlet oxygen species should be expanded to encompass more broad aspects. Given these considerations, the paper may not currently align with the exceptional quality expected by Nature.

How does the manuscript address a general topic of **interest for the broad readership of Nature**? On purpose, **we approach the topic from a general side**, *i.e.*, we use an extensive range of chemical oxidants to elucidate the oxidation of a radical anion into its ground and excited state products. We then **present several examples of superoxide oxidation and its interest in very different fields, rather than for the narrow confines of batteries alone**.

- 1) **Superoxide oxidation reactions in general.** Any reaction we found in the literature, producing ${}^1\text{O}_2$ or not, could be explained. Examples we considered are: 1) ${}^1\text{O}_2$ formation has been examined for exposure of superoxide to organic peroxides, given their occurrence in biological systems²⁸. No reaction was found with alkyl peroxides, while acyl peroxides yielded ${}^1\text{O}_2$. This behaviour has not been linked to driving force, but can now be understood in this way. 2) We have previously shown that CO_2 in contact with superoxide yields ${}^1\text{O}_2$, but the energetics were unclear²⁹. We describe these examples in more detail in the section “*Driving forces upon superoxide oxidation*” (for editorial reasons part of the Methods) and in Extended Data Fig. 4.
- 2) **Superoxide disproportionation in non-aqueous systems.** Relevant for batteries but likely also for hydrophobic environments in biological systems. Please note the counterintuitive driving force dependence shown in Figs. 3 and 4, which shows increasing ${}^1\text{O}_2$ formation with lower Lewis acidity and decreasing one with lower Brønsted acidity.
- 3) Explaining a half-century-old question: **${}^1\text{O}_2$ from proton-induced disproportionation.** As mentioned, there has been a fierce debate for over 50 years on whether ${}^1\text{O}_2$ forms in this way. The debate is almost as old as the knowledge about the existence of ${}^1\text{O}_2$ itself³⁰. This debate has been driven until now mainly by its relevance for the life sciences³¹. Conclusions undulated around positive (Khan *Science* 168, 476 (1970)), possible (Koppenol, *Nature* 262, 420 (1976)), negative (Aubry, *JACS* 103, 4965 (1981); Allen, *Photochem. Photobiol.* 39, 703 (1984)), positive (Khan, *PNAS* 91, 12365 (1994)), possible (Bodo, *ChemPhysChem* 21, 2060 (2020)). Our finding that ${}^1\text{O}_2$ from proton-induced disproportionation increases as the pH decreases coincides with higher and lower pH in respiratory (mitochondria) and degenerative organelles (lysosomes), respectively, where ${}^1\text{O}_2$ must be avoided or may even be beneficial. This connection between pH and ${}^1\text{O}_2$ formation may well have been an unrecognised evolutionary driver for the pH found in

²⁸ McNeill, K. J. *Org. Chem.* 71, 796 (2006). Khan, & Kasha, M. *Proc. Natl. Acad. Sci. USA* 91, 12365 (1994).

²⁹ Mondal et al. *Faraday Discuss.* 248, 175 (2024).

³⁰ Khan *Science* 168, 476 (1970).

³¹ E.g., Sies. Fundamentals of redox regulation in biology. *Nat. Rev. Mol. Cell. Biol.* 25, 701 (2024). Di Mascio. Singlet Molecular Oxygen Reactions with Nucleic Acids, Lipids, and Proteins. *Chem. Rev.* 119, 2043 (2019).

organelles. It gives insight into how life works and how it deals with the adverse effects of its inherent chemistry. Beyond this, the insight may have implications in pathology, considering that pathological situations typically result in tissue becoming more acidic³².

Following the Reviewer's suggestion, we now discuss the implications for the life sciences more broadly in the conclusions and the new supplementary discussion section "Wider relevance for life sciences and energy". Nevertheless, suggesting practical approaches for mitigating ¹O₂ in the life sciences is outside the scope of this paper. We hope and believe that the life sciences community will adopt the insights presented here.

- 4) Finally, the results are of interest for understanding **electrochemiluminescent reactions in general**. We show, to the best of our knowledge, the first-ever example of individually resolved Marcus parabolas for ground and excited states, even covering the inverted region of the excited state. Superoxide oxidation is merely the example by which we demonstrate it. We may not have emphasised sufficiently the importance of the findings for the electrogeneration of excited, reactive species in general, given the analogous underlying physics. The results expand the current knowledge on the electrogeneration of excited species more generally and pose open questions about the origin. Specifically, the process is more effective than typically assumed (see Fig. 3.1a), given that $\Delta G_{1\leftarrow 3}^{\circ} < \Delta H_{1\leftarrow 3}^{\circ}$ and that $Z_{el,1} \gg Z_{el,3}$. Typically, $\Delta G_{1\leftarrow 3}^{\circ} = \Delta H_{1\leftarrow 3}^{\circ}$ and $Z_{el,1} = Z_{el,3}$ would be assumed. The electrogeneration of excited species is of enormous significance for the generation of reactive and degrading species in various systems, ranging from cell biology to batteries. We list here some examples. Beyond the superoxide/¹O₂ system, the enormous breadth of relevance of electrochemiluminescent reactions includes these examples:
- Formation of reactive excited species in Life³³. They are very broadly associated with pathogenic events.
 - Recombination reactions in redox flow batteries³⁴. High-energy redox flow batteries aim at large voltage gaps between oxidised and reduced species at the cathode and anode, with voltage gaps sometimes exceeding 3 V. Cross-over and recombination of these charged species (typically radical anions and radical cations) is recognised to cause self-discharge, but so far, it has not been recognised that this may inherently form extremely energetic excited species through the mechanism of electrochemiluminescence (up to ~3-fold the energy of ¹O₂) [REDACTED]
 - Similarly, Li- and Na-ion batteries operate around 3 to 4 V. Soluble parasitic oxidised and reduced species at the cathode and anode may recombine to form energetic excited species, causing further degradation [REDACTED]

Our results are unexpected and surprising, as we find significantly different maximum kinetics (prefactors) for the ground and excited states. This unexpected difference exceeds by far current theoretical predictions³⁵ and, hence, poses new questions to theorists. Illuminating, as we now show how the previously inaccessible entropy change can be directly measured³⁶. Future work is warranted to investigate these quantities for other electrochemiluminescent reactions.

³² Gatenby. Why do cancers have high aerobic glycolysis? *Nature Reviews Cancer* 4, 891-899 (2004).

³³ Sies et al. Fundamentals of redox regulation in biology. *Nat. Rev. Mol. Cell. Biol.* 25, 701 (2024). Goncalves et al. Chemiexcitation: Mammalian Photochemistry in the Dark. *Photochem. Photobiol.* 99, 251-276 (2023).

³⁴ Lu et al. *Nat. Energy* 6, 582 (2021).

³⁵ Schmickler. The Pre-exponential Factor in Electrochemistry. *Angew. Chem. Int. Ed.* 57, 7948 (2018).

³⁶ This is a crucial quantity for catalytic efficiency. Zeradjanin. *Nat. Energy* 9, 514 (2024).

Is all this a **sufficiently general topic of interest for the broad readership of *Nature***? For comparison, excellent works with a narrower scope, such as Li-O₂ batteries (*Nature* 529, 377 (2016)) or O-redox battery cathodes (*Nature* 594, 213 (2021)), may be quoted. Equally, we are delighted to share the advices of the other Reviewers, who state “*Without further ado, the results reported by the authors are groundbreaking and fall within the scope of the Nature journal.*” or “*My opinion is that, given the relevance of the subject and the importance of such a systematic study, the paper is suitable for Nature.*”

We summarise this discussion very briefly in the Conclusions and in more detail in a new supplementary discussion section “*Wider relevance for life sciences and energy*”. We sincerely hope that the Reviewer will find the above considerations and the revised manuscript, in response to their comments, convincing enough to support publication.

1. The manuscript provides several experimental points about superoxide disproportionation into the triplet molecule products in Fig. 1a, but it lacks the data points correlated with the formation of singlet oxygen molecule products.

Please note that these data points represent measured rate constants for KO₂ oxidation with mediators, which cover up to moderate driving forces of ~1.2 eV (~3.75 V vs K/K⁺) using tetraglyme (TEGDME) as the electrolyte. Higher driving forces could not be utilised in the TEGDME electrolyte due to its limited oxidative stability. Due to this instability, the values at the highest driving forces already exhibited significant errors. These values were from a previous publication as cited in the figure legend³⁷. In response to another Reviewer’s comment, we moved these data points, the fitted parabola, and the associated ¹O₂ yields to Extended Data Fig. 1 (Fig. R3.2 below).

These data points suggested two major conclusions in the previous publication:

- 3) Superoxide oxidation kinetics as a function of driving force follow one single Marcus-type kinetics parabola, and many commonly used mediators act already in the inverted region.
- 4) Driving forces above 0.97 eV (redox potentials $>E_{\text{O}_2/\text{KO}_2}^\circ + 0.97\text{V}$) do produce ¹O₂.

The second conclusion tied in with the views of previous works³⁸ on ¹O₂ evolution from alkali peroxide oxidation, which considered a “*threshold*” for ¹O₂ evolution at $E_{\text{O}_2/\text{superoxide}}^\circ = E_{\text{O}_2/\text{superoxide}}^\circ + 0.97\text{V}$. The frequently used connotation of a “*threshold*” arises from the electrochemiluminescence literature, which considers “*energy-sufficient*” processes³⁹.

Note that none of these previous works³⁸, despite referring to electrochemiluminescent reactions, considered the relative kinetics of the formation of ground/excited state molecules. Remarkably, many reviews on electrochemiluminescence do not even mention the word “*kinetics*” or “*rate*” and only refer to “*energy-deficient*” or “*energy-sufficient*” processes.

³⁷ Petit, et al. *Nat. Chem.* 13, 465 (2021).

³⁸ *Angew. Chem. Int. Ed.* 50, 2999 (2011); *J. Phys. Chem. Lett.* 3, 3043 (2012); *Angew. Chem. Int. Ed.* 55, 6892 (2016).

³⁹ Bard & Faulkner, *Electrochemical Methods: Fundamentals and Applications*; Miao, *Chem. Rev.* 108, 2506 (2008); Richter, *Chem. Rev.* 104, 3003 (2004). Knight & Greenway. *Analyst* 119, 879 (1994).

Following Marcus' classical work on the electrogeneration of electronically excited species⁴⁰, in Fig. 1, we used the kinetic parabola as derived from measurements at low to moderate driving forces (now in Extended Data Fig. 1) and hypothesised that a second parabola for excited-state formation should occur. To access the extensive range of driving forces, we chose acetonitrile as the solvent and an extensive range of mediators. The results are shown in Fig. 2.

Fig. R3.2 (revised Extended Data Fig. 1) | Kinetics and ¹O₂ yields during mediated superoxide oxidation in TEGDME electrolyte. Data from Ref. 18. a, We measured the kinetic constants *k* for mediated KO₂ oxidation with mediators up to moderate redox potentials. Plot of ln(*k*) versus mediator potential ($E_{\text{RM}^{\text{ox/red}}}^\circ$, top axis) and driving force ($-\Delta G^\circ$, bottom axis). $-\Delta G^\circ = (E_{\text{RM}^{\text{ox/red}}}^\circ - E_{\text{O}_2/\text{KO}_2}^\circ)F$, where $E_{\text{O}_2/\text{KO}_2}^\circ = 2.48 \text{ V vs K/K}^+$. The blue curve is a fit to Marcus kinetics in equation (1), which gives $Z_{\text{el}} = 6.4 \times 10^{-3} \text{ cm s}^{-1}$ and $\lambda = 0.61 \text{ eV}$. The red dashed line indicates the driving force at $E_{\text{O}_2/\text{O}_2^-}^\circ = E_{\text{O}_2/\text{KO}_2}^\circ + 0.97 \text{ V}$, the commonly expected “threshold” for ¹O₂ formation. **b**, The formed ¹O₂ per mole of RM^{ox}. ¹O₂ yields were measured using 9,10-dimethylanthracene (DMA) conversion to DMA-O₂ using HPLC.

⁴⁰ Marcus, R. A. On the Theory of Chemiluminescent Electron-Transfer Reactions. *J. Chem. Phys.* 43, 2654 (1965).

2. *The triplet and singlet oxygen molecule yields depending on the mediators suggested in Fig. 2b-c does not follow the expected values. Despite the possible reasons the manuscript suggests including the technical issues and overlooked decay pathways, the authors then should provide more accurate consensus by adopting other theoretical models beyond the renowned Marcus theory since the huge deviations of data significantly undermines the credibility of the manuscript.*

Thank you for this request, which makes the data presentation a lot clearer. Comparing measured $^3\text{O}_2/\text{RM}^{\text{ox}}$ yields and NIR intensities with simulated values certainly allows for a better connection.

We now simulate $^3\text{O}_2/\text{RM}^{\text{ox}}$ yields and NIR emission intensities as a function of mediator redox potential. We considered the following processes:

- 1) $^3\text{O}_2$ and $^1\text{O}_2$ formation rates (k_3 and k_1) as given by the two kinetic parabolas in Fig. 2a.
- 2) Physical and reactive $^1\text{O}_2$ quenching by the solvent.
- 3) Physical and reactive $^1\text{O}_2$ quenching by the reduced mediators RM^{red} . Note that the same processes with RM^{ox} are typically negligible in comparison because of the electron demand of these processes⁴¹.
- 4) $^3\text{O}_2$ losses ($^3\text{O}_2/\text{RM}^{\text{ox}} < 1$) result from reactive quenching of $^1\text{O}_2$ with solvent or mediator.

Details about the model are given in the Methods Section “From oxidation rates to $^3\text{O}_2$ yields and NIR emission intensities” (line 395) and Extended Data Fig. 3. In short, the measurable NIR intensity will be proportional to the $^1\text{O}_2$ concentration, $I_{1270\text{nm}} \propto c_{^1\text{O}_2} \cdot c_{^1\text{O}_2}$ results from the balance of $^1\text{O}_2$ formation rate (proportional to k_1) and decay rate (considering the physical and reactive $^1\text{O}_2$ quenching processes)⁴². Total $^1\text{O}_2$ quenching rate constants by solvent and RM were determined by the $^1\text{O}_2$ lifetime. The fraction of reactive quenching with solvent and RM was determined by fitting the model to measured $^3\text{O}_2$ losses⁴³. Figure R3.3 below shows the revised Fig. 2 and legend with the simulated $^3\text{O}_2/\text{RM}^{\text{ox}}$ yields and NIR emission added. In the simulation, we used both the trendline and the individually measured values of the mediators for the mediator quenching rate constant k_Q (Extended Data Fig. 3c). The simulation results are presented by the dashed line (for the k_Q trendline) and the circular markers (individual k_Q values), respectively.

The measured values follow the simulated ones, which provides evidence for assigning the two kinetic parabolas in Fig. 2a to $^3\text{O}_2$ and $^1\text{O}_2$ formation. Please note that deviations between simulated and measured values result from simplifications in the model, which we intentionally restricted to capture the most important processes. We detailed the simplifications and their effects in the Methods section. For example, deviations in $^3\text{O}_2/\text{RM}^{\text{ox}}$ yields for TDPA, F4BQ, and Fc result from their reactivities with O_2 , which are absent for the other RMs. Moreover, a common reactive fraction of k_Q is a simplification given the diverse chemical nature of the RMs.

⁴¹ Kwak et al. *Nat. Commun.* 10, 1380 (2019).

⁴² This approach is in accord with standard literature: Wilkinson, et al. Rate Constants for the Decay and Reactions of the Lowest Electronically Excited Singlet-State of Molecular-Oxygen in Solution. An Expanded and Revised Compilation. *J. Phys. Chem. Ref. Data*, 24, 663 (1995).

⁴³ This is the commonly used way: Schweitzer et al. Physical Mechanisms of Generation and Deactivation of Singlet Oxygen. *Chem. Rev.* 103, 1685 (2003).

Fig. R3.3 (revised Fig. 2) | Free energy dependence of superoxide oxidation kinetics to 3O_2 and 1O_2 and their yields.

a, We measured the kinetic constants k for mediated KO_2 oxidation in MeCN electrolyte with mediators covering a large range of redox potentials. Plot of $\ln(k)$ versus mediator potential ($E_{RM^{ox/red}}^\circ$, top axis) and driving force ($-\Delta G^\circ$, bottom axis). $-\Delta G^\circ = (E_{RM^{ox/red}}^\circ - E_{^3O_2/KO_2}^\circ)F$, where $E_{^3O_2/KO_2}^\circ = 2.48$ V vs K/K⁺. The mediators are shown in Extended Data Fig. 2. The full line best fits equation (2); the broken line parabolas represent the first and second term in equation (2). Fitted values are $Z_{el,3} = 1.10 \cdot 10^{-2}$ cm·s⁻¹, $Z_{el,1} = 7.00 \cdot 10^{-2}$ cm·s⁻¹, $\lambda_3 = \lambda_1 = 0.95$ eV, $\Delta G_{1\leftarrow 3}^\circ = 0.84$ eV, $R^2 = 0.998$. Based on these fits, the standard potential E_{1O_2/KO_2}° and associated driving force $\Delta G_{1\leftarrow 3}^\circ$ are marked. They are linked via $E_{1O_2/KO_2}^\circ = E_{^3O_2/KO_2}^\circ + \Delta G_{1\leftarrow 3}^\circ/F$. The blue and red shaded area indicates the transition from $k_3/(k_{1+3}) = 0.99$ to $k_1/(k_{1+3}) = 0.99$. **b**, 3O_2 yield per mole of RM^{ox} (bars) during KO_2 oxidation as measured by mass spectrometry.

The dashed line and the circular markers show simulated $^3\text{O}_2$ yields considering $^1\text{O}_2$ quenching by solvent and redox mediator (see Methods and Extended Data Fig. 3). The dashed line used the trend line for the mediator quenching rate constant k_Q while markers use the individually measured values (Extended Data Fig. 3c). **c**, Normalized 1270 nm NIR emission (bars) during KO_2 oxidation. The dashed line and the circular markers show the simulated NIR emission considering $^1\text{O}_2$ formation with the kinetics k_1 (the right parabola in **a**) and $^1\text{O}_2$ quenching by solvent and redox mediator (see Methods and Extended Data Fig. 3). Data are presented as mean \pm SD ($n \geq 3$).

Overall, the simulated values reproduce the trend of the measured values of $^3\text{O}_2/\text{RM}^{\text{ox}}$ and NIR emission very well, giving evidence for assigning the two kinetic parabolas to $^3\text{O}_2$ and $^1\text{O}_2$ formation.

We modified the text associated with Fig. 2 from line 118. Thank you again for this very valuable request.

3. The figures in the manuscript are not sufficiently self-descriptive and do not adequately explain the content. For instance, in Figure 1b, it is challenging to discern what each parabola represents and how elements (i), (ii), and (iii) influence them. Similarly, Figure 4 fails to intuitively convey the correlation between the Pourbaix diagram and singlet oxygen formation. Enhancing the figures with clearer labels and explanatory legends is necessary to improve comprehension.

Thank you for pointing out places where we could improve the figures and legends.

We thoroughly revised all figures and legends in response to this comment, several of your other comments, as well as comments of other Reviewers.

Specifically, concerning this comment, we simplified Fig. 4a and added a new panel, Fig. 4b, which shows the driving force as a function of pH.

4. The manuscript requires overall clarity improvements. For instance, the authors frequently interchange terms such as potassium superoxide, lithium superoxide, and superoxide ion without clear distinction. It is essential to consistently and clearly differentiate these species within both the figures and the text to avoid confusion.

Thank you for pointing out places where we could improve clarity. We reviewed all occurrences and ensured that it is clear what is meant.

Ambiguities appear to have arisen from the use of the term “ O_2^- ”, which we used as a shorthand for “superoxide” generally, without restriction to a particular species. We figured that this would become clear from the context and avoid the text from becoming unnecessarily clunky.

For example, when we wrote $2 \text{O}_2^- \rightarrow \text{O}_2 + \text{O}_2^{2-}$ in the introduction we did not mean free superoxide to disproportionate to free peroxide and O_2 . The coordination of superoxide and peroxide would, of course, depend on the available cations. Accordingly, we also used “ O_2^- ” in cases where superoxide assumed a range of possible coordinations. For example, we wrote $E_{^3\text{O}_2/\text{O}_2^-}$ in the context of mixed Li^+/TBA^+ electrolytes, where the superoxide could be in species anywhere between $\text{LiO}_{2(\text{s})}$, $(\text{Li}^+\text{O}_2^-)_{(\text{sln})}$ ion pairs and clusters, or $(\text{TBA}^+\text{O}_2^-)_{(\text{sln})}$.

To avoid this possible confusion, we revised the text such that at places where a range of superoxide species is considered, we write “superoxide” rather than “ O_2^- ”. For example, we now write $E_{\text{O}_2/\text{superoxide}}^\circ$ rather than $E_{\text{O}_2/\text{O}_2^-}^\circ$. The range of species meant is mentioned in the text. If particular species are meant, they are clearly denoted, e.g., $E_{\text{O}_2/(\text{Li}^+\text{O}_2^-)}(\text{sln})$.

Of note, in accord with the IUPAC recommendation⁴⁴, we now denote solvated species with (sln), which is recommended for the state of aggregation rather than (sol).

5. *The manuscript would benefit from a thorough revision to enhance the quality of English writing and adopt a more academic tone throughout the text. For example, on page 8, line 203, the authors state: “They also enhance kinetics. However, these findings lacked a conclusive explanation, which the insights from Fig. 2 can provide.” It is unclear what “they” refers to and what specific insights Figure 2 offers. The authors should provide explicit explanations of such references to improve clarity.*

We apologise if the English was unclear in places. We thoroughly revised the manuscript and also had it checked by a native speaker.

As for the particular example given, “they” referred to the addition of weak Lewis acids such as TBA^+ . The findings from a previous paper⁴⁵, where we could not provide conclusive explanations, were that a pure Li^+ electrolyte did produce $^1\text{O}_2$ and that the $^1\text{O}_2$ fraction increased from 1% with pure Li^+ electrolyte to 20% with mixed Li^+/TBA^+ electrolyte. We revised the text and refer to this particular result more comprehensively in response to your following comment.

6. *On page 9, line 214, the authors claim that dissolved LiO_2 is less stable compared to solid LiO_2 , leading to energy inversion phenomena in pure Li^+ electrolytes. However, it is generally observed that when solid species dissolve and become solvated, they are typically more stabilized. Furthermore, considering the discharge pathway of lithium-air batteries, intermediate dissolved LiO_2 states precede the formation of Li_2O_2 without precipitation of solid LiO_2 . This suggests that dissolved LiO_2 may, in fact, be more stable than its solid counterpart. The authors should reconcile these statements with existing evidence to ensure consistency and accuracy.*

In short, there is broad literature support that dissolved (solvated) LiO_2 and Li_2O_2 are less stable (in terms of Gibbs free energy) than their bulk solids; we are not aware of a paper showing the opposite. We agree that existing evidence supports LiO_2 to be soluble to an extent that, during discharge of lithium-air cells, it acts as a solution species. Li_2O_2 is agreed on in the literature to be very insoluble.

Please allow us to respond separately and more comprehensively to the two parts of this comment.

⁴⁴ Brett, C. M. A. et al. *Quantities, Units, and Symbols in Physical Chemistry*. (RSC Publishing, 2023), DOI: 10.1039/9781839163180, Page 48.

⁴⁵ *Energy Environm Sci*, 2019, 12, 2559.

Stability of dissolved vs. solid species:

It is right that the solvation of a species will stabilise it relative to the isolated, non-solvated species in the gas phase. However, dissolving a solid into a solution requires overcoming the lattice energy, which must be compensated for by solvation. Solubility may be related to the energies of sublimation (solid→gas), solvation (gas→solution), and dissolution (solid→solution) by $\Delta G_{\text{dissolution}} = \Delta G_{\text{sublimation}} + \Delta G_{\text{solvation}} = -R \cdot T \cdot \ln(S \cdot V)$, where S and V are solubility and molar volume of the solute⁴⁶. For low solubilities, $\Delta G_{\text{dissolution}}$, the energy difference between the bulk solid and the solvated species is positive; hence, the solvated species is less stable than the solid.

In support of this argument, we compiled in Fig. R3.4 literature data on the energy difference between bulk solid $\text{LiO}_{2(\text{s})}$ and $\text{Li}_2\text{O}_{2(\text{s})}$ and the solvated species. The papers employ different levels of DFT calculations and consistently arrive at the same conclusion that the *solvated species are less stable than the solids*. ~2 eV for Li_2O_2 and ~0.5 eV for LiO_2 are in accord with lower solubility of Li_2O_2 than LiO_2 . Dissociated species, e.g. ($\text{Li}^+_{(\text{solvated})} + \text{O}_2^-_{(\text{solvated})}$) are even less stable than the associated ones. A brief explanation of the four subplots:

- Solvation energy denotes here the energy of the solvated species ($\text{LiO}_{2(\text{solvated})}$ and $\text{Li}_2\text{O}_{2(\text{solvated})}$) relative to the bulk solid $\text{LiO}_{2(\text{s})}$ and $\text{Li}_2\text{O}_{2(\text{s})}$. Values are given for the molecule as well as for one cation dissociated ($\text{Li}^+_{(\text{solvated})} + \text{O}_2^-_{(\text{solvated})}$ and $\text{Li}^+_{(\text{solvated})} + \text{LiO}_2^-_{(\text{solvated})}$). Solvents with ϵ ranging from 7 to 30 (resembling dimethoxyethane to acetonitrile) were considered. The significantly higher values for the dissociated species show that dissociation is unfavourable, particularly for the peroxides.
- The free energy profile of the superoxide disproportionation in Li^+ electrolyte, where (sol) and (s) denote solvated and solid species, respectively. The steps $\text{Li}_2\text{O}_{2(\text{sol})} + \text{O}_{2(\text{sol})} \rightarrow \frac{1}{2} (\text{Li}_2\text{O}_2)_{2(\text{sol})} + \text{O}_{2(\text{sol})} \rightarrow \text{Li}_2\text{O}_{2(\text{s})} + \text{O}_{2(\text{sol})}$ show the significant stabilisation from the single molecule via the dimer to the solid.
- Energy difference between solvated and solid Li, Na, and K peroxides in glyme.
- Dissolution energy of LiO_2 and Li_2O_2 into organic solvents with ϵ ranging from 7 to 46 (resembling dimethoxyethane to DMSO). Data are calculated via $\Delta G_{\text{dissolution}} = \Delta G_{\text{sublimation}} + \Delta G_{\text{solvation}}$, where $\Delta G_{\text{solvation}}$ is the energy of solvation of the gas phase species. $\Delta G_{\text{dissolution}}$ is the dissolution energy, which is the energy difference between dissolved, solvated species and their solid.

⁴⁶ Curtiss et al. *J. Electrochem. Soc.* 164, E3696 (2017). Fedorov et al. *J. Chem. Theory Comput.* 8, 3322 (2012)

REDACTED

Fig. R3.4a: Figure 4 in Kang et al. Nat. Commun. 7, 10670 (2016).

Fig. R3.4b: Figure 4 in Peng et al. J. Phys. Chem. C 120, 3690-3698 (2016).

Fig. R3.4c: Table S3 in Freunberger et al. Energy Environ. Sci. 12, 2559-2568 (2019).

Fig. R3.4d: Table III in Curtiss et al. J. Electrochem. Soc. 164, E3696-E3701 (2017).

Fig. R3.4 |

Energy inversion phenomena in pure Li⁺ electrolytes:

We fully agree with the Reviewer mentioning “*intermediate dissolved LiO₂ states precede the formation of Li₂O₂*”. Furthermore, there is evidence that solid Li₂O_{2(s)} forms via disproportionation of LiO_{2(sln)} intermediates, and not via a 2nd electroreduction or precipitation of initially formed dissolved Li₂O_{2(sln)} (as it is insoluble). In other words, Li₂O_{2(s)} forms most facily on existing solid seeds from dissolved LiO_{2(sln)}.

Evidence for this comes from in-situ SAXS/WAXS, RRDE and SEM measurements⁴⁷. LiO₂ is soluble and mobile even in poorly solvating MeCN electrolytes. From these statements, it follows that the relevant reaction, for which thermodynamics needs to be considered, is the following:

Dissolved LiO_{2(sln)} will typically aggregate into (Li⁺O₂⁻)_{n,(sln)} clusters⁴⁸.

The effect of insoluble Li₂O₂ is that $E_{3\text{O}_2/\text{Li}_2\text{O}_{2(\text{s})}}^\circ$ is pinned to the value determined from the formation energy of solid Li₂O_{2(s)} using $\Delta_f G^\circ = -nFE^\circ$. The $E_{3\text{O}_2/\text{superoxide}}$ needs further considerations as detailed in the new Methods section “Thermodynamics in mixed alkali metal/TBA⁺ electrolytes”. Fig. R3.5a shows CVs for O₂ reduction in glyme electrolyte containing either TBA⁺ or Li⁺ salt. (TBA⁺O₂⁻)_(sln) is clearly the most solvated superoxide species, and its potential $E_{3\text{O}_2/(\text{TBA}^+\text{O}_2^-)_{(\text{sln})}}$ the lowest measured in this system. $E_{3\text{O}_2/\text{LiO}_{2(\text{s})}}$ may be estimated from a slow CV in poorly solvating electrolytes (red line in Fig. R3.5a), where LiO₂ aggregates into large (Li⁺O₂⁻)_{n,(sln)} clusters, which approach the thermodynamics of LiO_{2(s)}. $E_{3\text{O}_2/(\text{Li}^+\text{O}_2^-)_{(\text{sln})}}$ is the lower limit of potentials in pure Li⁺ electrolyte and can be estimated from the shift of the ORR potential in pure Li⁺ versus pure TBA⁺ electrolyte with the highly solvating solvent 1-methylimidazole. Fig. R3.5b shows these CVs from Bruce et al.⁴⁹, where a shift of 33 mV is seen. Fig. R3.5c (new Extended Data Fig. 5c) gives a more detailed scheme of these solvation-dependent potentials. For a pure Li⁺ electrolyte, the potentials are within the limits $E_{3\text{O}_2/\text{LiO}_{2(\text{s})}} > E_{3\text{O}_2/(\text{Li}^+\text{O}_2^-)_{n>1(\text{sln})}} > E_{3\text{O}_2/(\text{Li}^+\text{O}_2^-)_{(\text{sln})}}$. If TBA⁺ is present, the even weaker association in (TBA⁺O₂⁻)_(sln) extends the lower potential limit to $E_{3\text{O}_2/(\text{TBA}^+\text{O}_2^-)_{(\text{sln})}}$.

We revised Fig. 3a (R3.5d) to make it clearer. The ³O₂/superoxide potentials (light blue) are shown in the boundaries as just mentioned. The ¹O₂/superoxide potentials (red) result from an upshift of 0.84 V. The superoxide/Li₂O_{2(s)} couple is the oxidant for superoxide during disproportionation. The oxidant potential is $E_{\text{superoxide}/\text{Li}_2\text{O}_{2(\text{s})}} = 2E_{3\text{O}_2/\text{Li}_2\text{O}_{2(\text{s})}} - E_{3\text{O}_2/\text{superoxide}}$ and therefore depending on the thermodynamics of the superoxide species. For solvated LiO₂ species, $E_{\text{superoxide}/\text{M}_2\text{O}_{2(\text{s})}}$ exceeds $E_{1\text{O}_2/\text{superoxide}}$, which explains ¹O₂ formation.

⁴⁷ Prehal et al. *Proc. Natl. Acad. Sci. USA* 118, e2021893118 (2021); Prehal et al. *ACS Energy Lett.* 7, 3112 (2022); Lu et al. *Joule* 2, 2364-2380 (2018).

⁴⁸ Shao-Horn et al. *Angew. Chem. Int. Ed.* 55, 3129 (2016).

⁴⁹ Johnson et al. *Nat. Chem.* 6, 1091 (2014).

REDACTED

Fig. R3.5b: Figure 2a in Johnson et al. Nat. Chem. 6, 1091 (2014).

Fig. R3.5 | Potential shift as a function of LiO₂ solvation. **a**, Cyclic voltammograms in O₂-saturated tetraglyme containing 100 mM of either TBAClO₄, LiClO₄.

c, Detailed thermodynamics of the ³O₂/superoxide couple as a function of the fractions of Li⁺ and TBA⁺ salt. The gradient box and the arrow indicate increasing superoxide solvation with potentials between values relevant for solid LiO_{2(s)} (dark colour), solvated (Li⁺O₂⁻)_{n≥1,(sln)} clusters, and solvated (Li⁺O₂⁻)_(sln) (faded colour). **d**, (revised Fig. 3a) Based on **c**. Thermodynamics of relevant redox couples for Li⁺-induced superoxide disproportionation as a function of the fractions of Li⁺ and TBA⁺ salt. LiO₂ is in all relevant electrolytes at least somewhat soluble as (Li⁺O₂⁻)_{n≥1,(sln)} clusters. The nature of superoxide shifts from (Li⁺O₂⁻)_{n≥1,(sln)} towards (TBA⁺O₂⁻)_(sln) as the cation changes from pure Li⁺ towards pure TBA⁺. The inclined lines indicate the associated shift of the potentials. $E_{1O_2/superoxide} = E_{3O_2/superoxide} + 0.84$ V. The driving force $\Delta G = -(E_{superoxide/Li_2O_2} - E_{3O_2/superoxide})F$ grows with the shift of the O₂/superoxide and superoxide/Li₂O_{2(s)} potentials. Note that the O₂ reduction potential changes nonlinearly with the Li⁺:TBA⁺ ratio¹. **a,c** are from Extended Data Fig. 9, **d** is the revised Fig. 3a.

7. *Although the authors address factors such as pH and solvent properties, more concrete solutions for achieving stable oxygen redox behavior would strengthen the manuscript's impact.*

We expanded the Conclusions section to be more concrete about implications for life sciences and energy applications. Given that this had to be very concise, we added more detailed discussion in a new Methods section “*Wider relevance for life sciences and energy*”, see below. With respect to solvent properties and salts, we added a discussion in the section “*Thermodynamics in mixed alkali metal/TBA⁺ electrolytes*”.

Please note that we – and very likely anyone – will not be able to present a magic bullet that solves all problems that ¹O₂ may cause in the diverse systems we touched. The core of the paper is to experimentally demonstrate how the driving force for superoxide oxidation controls the formation of ¹O₂ and ³O₂. We examined this for three ways to impact the driving force: chemical oxidants and superoxide disproportionation in non-aqueous and aqueous systems. Furthermore, the reorganisation energy determines at which driving force ¹O₂ evolution will become significant. In manmade systems, there is some limited freedom to manipulate.

We further describe how the work expands the current knowledge on the electrogeneration of excited species more generally, beyond ¹O₂, which is significant for biological systems and energy storage.

Wider relevance for life sciences and energy

Our paper contributes to understanding how pH impacts the link between the four important reactive oxygen species (ROS): superoxide, peroxide, ³O₂, and ¹O₂. Disproportionation is notably the pathway to maintain a low superoxide concentration, but detoxification from superoxide produces the harmful ¹O₂. Superoxide occurs in cells in several organelles with different pH levels between 4.7 and 8, but the superoxide degrading enzyme superoxide dismutase (SOD) only occurs in neutral to basic organelles²⁵. In pathological situations, the pH balance is known to be impacted (typically towards lower pH) and therewith signalling, redox regulation, and defence^{64,65}. Our paper contributes to the understanding of how the redox chemistry of superoxide, pH, and ¹O₂ formation are linked.

Concerning energy applications, further relevance and future research directions emerge: (A) For suppressing ¹O₂, generally, the driving force should be decreased, and the reorganisation energy for the superoxide oxidation reaction should be increased. The classical equation by Marcus⁶⁰, which connects the reorganisation energy with the effective dielectric properties of the electrolyte and the separation of the redox centres, applies well to aqueous anionic redox couples⁶². For non-aqueous, low dielectric constant media, where strong ion-pairing occurs, particularly with superoxide, the change in reorganisation energy among different cations may not be predicted straightforwardly and would merit further investigation. (B) Oxygen evolution catalysis from water: metal-superoxo species have been identified to precede the O₂ evolution on, for example, the extensively studied Ni(Fe)OOH or CoOOH catalysts⁶⁶. The metal M^{n-1/n} redox couple is considered to oxidise the superoxo moiety to O₂⁶⁷. Some of the most active M^{n-1/n} metal redox couples are typically a few 100 mV above the ¹O₂/superoxide potential shown in Fig. 4a and provide, in principle, enough driving force for ¹O₂ evolution. Further investigations, specifically on ¹O₂ evolution in oxygen evolution catalysis, are therefore warranted.

The results expand the current knowledge on the electrogeneration of excited species more generally and pose open questions about the origin. Specifically, the process is more effective than assumed, given that $\Delta G_{1\leftarrow 3}^{\circ} < \Delta H_{1\leftarrow 3}^{\circ}$ and that $Z_{el,1} \gg Z_{el,3}$. Electrogeneration of excited reactive species has significance ranging from biological systems to energy storage. Reactive excited species in life are very broadly associated with pathogenic events^{1,68}. Recombination reactions in redox flow batteries are recognised to cause self-discharge, but this has so far not been recognised to potentially form extremely energetic excited species. Soluble parasitic oxidised and reduced species at the cathode and anode of Li- and Na-ion batteries may recombine to form energetic excited species.

Response to reviewers on the paper entitled

“Individual Marcus-type kinetics controls singlet and triplet oxygen evolution from superoxide oxygen formation”

Manuscript: 2024-09-18358B

We thank the Reviewer for their helpful comment, to which we respond below. The Reviewer's comment are reproduced in italics. In the manuscript, we highlighted the changes in yellow. The comments helped us in improving the manuscript, and we are confident that we have satisfactorily responded to all of them.

Reviewer #3:

We appreciate the authors' detailed explanations and the revised manuscript that thoughtfully addresses our previous comments. Our primary concern pertained to the novelty of the main arguments, particularly regarding the kinetics and thermodynamics of superoxide disproportionation into singlet and triplet oxygen species in the presence of peroxide. The authors have now provided sufficient clarification concerning the novelty of their work in relation to their prior publications and other relevant studies. In particular, the strength of this manuscript lies in its rigorous elucidation of phenomena that have been experimentally observed yet previously lacked clear mechanistic interpretation. Moreover, the authors convincingly demonstrate the broad applicability of their findings across diverse disciplines, including the life sciences and energy storage, as articulated in the Conclusion and Methods sections. We find this interdisciplinary scope to be well aligned with Nature's broad readership, extending the relevance of the work beyond the battery field.

To further improve the clarity and perceived novelty of the manuscript prior to final publication in Nature, we offer the following suggestion:

As noted in the Introduction, oxygen redox chemistry plays a critical role in battery materials. In addition to metal–air batteries, layered transition metal oxides have recently gained attention as cathode materials that exploit oxygen redox reactions (see reference 2 in the manuscript). While the discussion of oxygen spin states in this context remains limited, the evolution and reactivity of various oxygen species are highly pertinent. We believe it would greatly benefit the community if the authors could briefly share their perspective on the relevance of their findings to the oxygen-redox cathode field as well.

Thank you very much for your favourable assessment of our previous revision. Thank you for raising this important point. We fully agree that the spin state of the evolved oxygen from transition metal intercalation materials deserves much deeper investigation as it likely holds the key to understanding an important unrecognized aspect of surface stability and reactivity. We expanded the Methods Section “Wider relevance for life sciences and energy” by a brief perspective on how the findings presented here will be relevant for the oxygen-redox cathode field. The added text including the two new references are reproduced below.

Wider relevance for life sciences and energy

Concerning energy applications, further relevance and future research directions emerge: (A) (B)

(C) Both Li-stoichiometric⁶ and Li-rich transition metal (TM, e.g., Ni, Mn, Co) oxide^{2,7,61,62} intercalation materials used for positive electrodes in Li- or Na-ion batteries are known to undergo parasitic lattice oxygen loss at high states of charge. Both the intercalation material and the electrolyte degrade, hampering long-term cyclability. However, non-matching patterns of O₂ and CO₂/CO release from electrolyte decomposition (all containing lattice O as shown by isotopic labelling^{7,62}) and enhanced lattice O loss with surface carbonates present⁷ remain unexplained and highlight the need for deeper understanding of the prevailing reactive O species and decomposition pathways. For LiNiO₂, ¹O₂ evolution has been suggested to result from the disproportionation of oxide radicals⁶. More generally, ¹O₂ may evolve from superoxo species (at the lattice surface, in (su)peroxocarbonates^{4,30}) at the available driving forces. The oxidants could be a combination of (su)peroxides (e.g., coordinated by TMs or carbonates) that get stabilized by further reduction and TM redox such as Co^{III/IV} or Ni^{III/IV}. Further investigations into the involvement of ¹O₂ evolution in TM oxide outgassing and surface reactions are therefore warranted.

61. Zhang, M. et al. Pushing the limit of 3d transition metal-based layered oxides that use both cation and anion redox for energy storage. *Nat. Rev. Mater.* **7**, 522-540 (2022).

62. Luo, K. et al. Charge-compensation in 3d-transition-metal-oxide intercalation cathodes through the generation of localized electron holes on oxygen. *Nat. Chem.* **8**, 684-691 (2016).